# $p$-LESS SAMPLING: A ROBUST HYPERPARAMETER-FREE APPROACH FOR LLM DECODING

**Runyan Tan**[1,2,3], **Shuang Wu**[1], **Phillip Howard**[1]
[1]Thoughtworks
[2]National University of Singapore
[3]Corresponding Author
{runyan.tan,shuang.wu,phillip.howard}@thoughtworks.com
run@nus.edu.sg

## ABSTRACT

Obtaining high-quality outputs from Large Language Models (LLMs) often depends upon the choice of a sampling-based decoding strategy to probabilistically choose the next token at each generation step. While a variety of such sampling methods have been proposed, their performance can be sensitive to the selection of hyperparameters which may require different settings depending upon the generation task and temperature configuration. In this work, we introduce $p$-less sampling: an information-theoretic approach to sampling which dynamically sets a truncation threshold at each decoding step based on the entire token probability distribution. Unlike existing methods, $p$-less sampling has no hyperparameters and consistently produces high-quality outputs as temperature increases. We provide theoretical perspectives on $p$-less sampling to ground our proposed method and conduct experiments to empirically validate its effectiveness across a range of math, logical reasoning, and creative writing tasks. Our results demonstrate how $p$-less sampling consistently outperforms existing sampling approaches while exhibiting much less degradation in text quality at higher temperature values. We further show how $p$-less achieves greater inference-time efficiency than alternative methods through lower average token sampling times and shorter generation lengths, without sacrificing accuracy. Finally, we provide analyses to highlight the benefits of $p$-less through qualitative examples, case studies, and diversity assessments. The code is available at ⌂.

## 1 INTRODUCTION

The increasingly impressive capabilities exhibited by Large Language Models (LLMs) in recent years have been aided by advancements in sampling-based decoding strategies which probabilistically select tokens at each generation step. In contrast to deterministic methods such as greedy decoding and beam search (Freitag & Al-Onaizan, 2017), sampling-based strategies can produce more diverse and human-like language outputs while avoiding issues such as neural text degeneration (Holtzman et al.). Truncation-based sampling strategies (e.g., top-$p$ (Holtzman et al.), top-$k$ (Fan et al., 2018b), $\epsilon$-sampling (Freitag et al., 2023), mirostat (Basu et al., 2021), min-$p$ (Nguyen et al., 2024)) have proven particularly effective by truncating the token probability distribution to only a subset of higher-likelihood tokens before sampling. However, the truncation thresholds produced by these existing approaches depend upon the specification of hyperparameters whose optimal values can vary depending upon the generation task and sampling temperature (Zhou et al., 2024).

To address this deficiency, we introduce $p$-less sampling: a parameter-less sampling strategy grounded in information theory. $p$-less sampling uniquely possesses several desirable properties for a sampling-based decoding method. Specifically, $p$-less sampling is a distribution-aware approach which dynamically adjusts the truncation threshold at each time step using the entire token probability distribution. In this way, $p$-less provides a more principled, information-theoretic approach to determining the truncation threshold than alternative methods. The truncation threshold produced by $p$-less also dynamically changes with temperature, producing robust results even at high temperature settings where

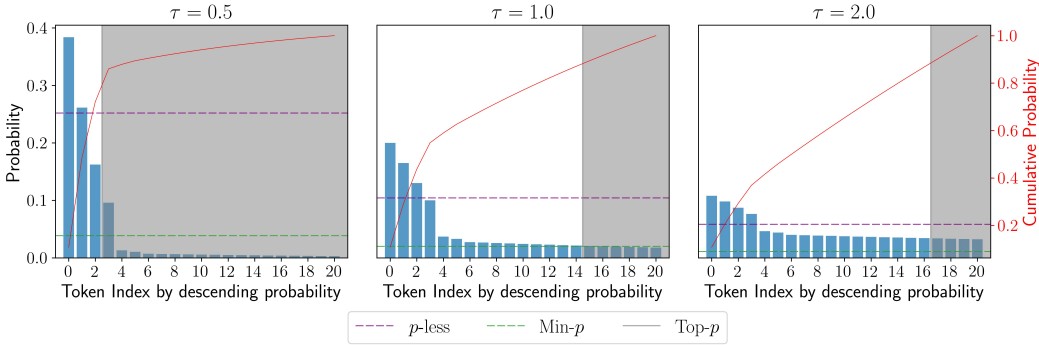

**Figure 1:** Comparison of truncation thresholds produced by $p$-less, min-$p$, and top-$p$ for a token probability distribution with different applied temperatures ($\tau$). As temperature increases, $p$-less avoids admitting a large number of lower-likelihood tokens by considering the entropy of the distribution in computing the threshold.

other sampling approaches suffer from text degeneration. An illustration of this effect is provided in Figure 1, which shows how $p$-less sensibly truncates the long-tail of lower probability tokens even at higher temperatures which flatten the probability distribution. In contrast, other approaches which do not consider the entropy of the entire probability distribution admit far more tokens at high temperature values, thereby leading to greater degeneracy. We include examples of token distributions in Appendix B.6 to show the effects of temperature, vocabulary size and distribution profiles on the truncation threshold, including the ability of $p$-less to admit tokens from long-tailed distributions.

We provide theoretical perspectives on $p$-less sampling and ground our approach by interpreting it in connection to the family of Rényi entropies (Rényi, 1961). Through extensive experiments, we demonstrate the strong performance of $p$-less sampling across five math, logical reasoning, and creative writing datasets using three LLMs spanning multiple model sizes and architectures. Our results show that $p$-less sampling excels at math and reasoning tasks, consistently achieving high accuracy across a wide range of temperature values. In contrast, other evaluated sampling approaches have significantly greater variability in performance with respect to temperature, often exhibiting declines in accuracy at high temperature values. Our $p$-less approach also provides similar benefits for creative writing, producing the best performance in automated evaluations for the writing prompts dataset. We further validate these results through a human evaluation study, finding that human annotators also prefer stories generated by $p$-less over alternative sampling approaches. Finally, we demonstrate the superior inference-time efficiency provided by $p$-less sampling and conduct additional analyses on text diversity, qualitative examples, and case studies.

To summarize, our main contributions are as follows:

1. We introduce $p$-less sampling: a robust parameter-less sampling strategy grounded in information theory

2. Through extensive experiments, we validate the effectiveness of $p$-less sampling using three LLMs and five datasets spanning math, logical reasoning, and creative writing tasks.

3. We show that $p$-less is more efficient than other methods, both in terms of token sampling speed and overall generation length, without sacrificing task-specific performance.

4. We provide additional analyses to highlight the benefits of $p$-less through the lens of text diversity, qualitative examples, and case studies.

## 2 RELATED WORK

Text decoding in LLMs involves a truncation process where low probability tokens or those expected to disrupt coherence are excluded from the sampling pool. Top-$k$ (Fan et al., 2018a) restricts sampling to the $k$ most probable tokens, which could lead to incoherent generations when the distribution is extremely uniform or peaked. Top-$p$ (Holtzman et al.) improves upon this by sampling from the smallest set of tokens whose cumulative probability exceeds a threshold $p$. $\epsilon$-sampling (Freitag et al.,

2023) proposes to truncate all tokens with probabilities below a cut-off threshold $\epsilon$ quantity. Both top-$p$ and $\epsilon$-sampling remain lacking in adapting to high-entropy conditions, which is typical when temperature is turned up, such as for use cases where diversity is preferred. Similar to $\epsilon$-sampling, $\eta$-sampling (Hewitt et al., 2022) proposes an entropy-aware variant which defines the threshold as the minimum of $\epsilon$ and a scaled negative Shannon entropy exponential quantity. This however introduces additional hyperparameters and relies on the assumption that the entropy follows a uniform distribution baseline.

A more sophisticated approach in the form of mirostat (Basu et al., 2021) assumes that the token distribution follows Zipf's Law and dynamically adjusts the threshold to maintain a target surprisal. This dynamic feedback involves task and model dependent tuning of the target surprisal and a learning rate. The min-$p$ method (Nguyen et al., 2024) attempts to simplify truncation through a fractional hyperparameter, which is multiplied by the modal probability to define the truncation threshold. While being more empirically robust at high temperatures, min-$p$ remains sensitive to the choice of the fractional hyperparameter, and leverages only a single statistic from the token probability distribution. Another truncation approach aimed at balancing diversity with coherence (Zhu et al., 2024) increments the sampling set until the improvement in confidence falls short of an error term; it is similarly sensitive to the choice of the error hyperparameter and additionally requires iterative computations for differences and comparisons. In contrast, our proposed $p$-less sampling method avoids these pitfalls by operating directly on the empirical token distribution without imposing parametric assumptions or requiring hyperparameter tuning. This approach offers a model-agnostic threshold which is robust in high-entropy regimes.

Beyond truncation-based sampling techniques, a variety of decoding approaches have been proposed to improve the quality of text produced by LLMs. Contrastive decoding (Li et al., 2022) aims to improve text quality by contrasting token predictions from an expert and amateur model, with the intent of maximizing expert-like generations while minimizing amateur-like text. (Su et al., 2022) introduces contrastive search decoding, which augments top-$k$ with an additional degeneration penalty hyperparameter $\alpha$ that balances the candidate token confidence with greater dissimilarity between the previous context and the context continuation using the token. (Arias et al., 2024) advances this strategy through examining local variations in the entropy to determine the top-$k$ and degeneration penalty hyperparameters, thereby improving robustness across diverse contexts. While this strategy adaptively proposes these hyperparameters at each generation step, a temperature factor $q$ is required and an implicit centering choice of the $k$ quantity is made via the introduction of a pair of bounding hyperparameters. (Ding et al., 2025) further incorporates the global entropy dynamics to stabilize decoding under sudden local entropy spikes and guarantees an unbiased estimator for the instantaneous entropy, offering resilience against volatility and eliminating the need for the temperature factor in (Arias et al., 2024); however it introduces two hyperparameters, $\lambda$ for computing the degeneration penalty and $w$ to specify the window size for examining variations in local entropy. Other controlled decoding methods such as Neurologic Decoding (Lu et al., 2020) constrain text generation to achieve various objectives such as improved diversity, which is particularly useful for applications like synthetic data generation (Howard et al., 2022; 2024; Rosenman et al., 2024). Arithmetic sampling (Vilnis et al., 2023) uses parallel sampling to improve beam sampling based on an arithmetic code book defined implicitly by the model; Parashar et al. (2024) demonstrate how arithmetic sampling produces more diverse generations than ancestral sampling across reasoning and translation tasks. While these methods have some overlapping aims as our work, they differ substantially in their level of complexity and can be viewed as complementary approaches that can be used in conjunction with $p$-less or other truncation-based sampling methods.

## 3 METHODOLOGY

### 3.1 THE $p$-LESS SAMPLING METHOD

At every time step $t$, an autoregressive language model infers a distribution of the vocabulary tokens conditioned on the token sequence already generated from step $1$ to step $t - 1$. Essentially, $p$-less considers information from the entire probability distribution after the application of temperature; it computes the likelihood of a correct random guess given the distribution, which serves as our principled probability threshold adapted to the token distribution at every decoding step. We admit tokens into the sampling distribution whose likelihoods are at least that of the correct random guess

likelihood. Furthermore, to determine which and how many tokens to sample from, the $p$-less threshold varies in a meaningfully opposite direction with entropy; as entropy increases, more tokens with lower probability are admitted for sampling.

Formally, let $v \in \mathcal{V}$ denote the set of tokens from a vocabulary $\mathcal{V}$. At each time step $t$, let $\mathcal{P}(\mathcal{S} = v)$ denote the probability that token $v$ is sampled and $\mathcal{P}(\mathcal{T} = v)$ denote the probability that token $v$ is the correct (or most desirable) token in the "ground-truth" sense. Furthermore, let $P_\theta(v \mid x_{1:t-1})$ denote the language model's predicted token distribution conditioned on the given token sequence $x_{1:t-1}$, where $\theta$ are the language model parameters. Denoting the probability that the sampled token matches the ground-truth as $L[P]$, we have:

$$
\begin{aligned}
L[P] &:= \sum_{v \in \mathcal{V}} \mathcal{P}(\mathcal{S} = v \ \cap \ \mathcal{T} = v \mid x_{1:t-1}) \\
&= \sum_{v \in \mathcal{V}} \mathcal{P}(\mathcal{S} = v \mid x_{1:t-1}) \, \mathcal{P}(\mathcal{T} = v \mid x_{1:t-1})
\end{aligned}
\tag{1}
$$

since the sampling $\mathcal{S}$ and correctness $\mathcal{T}$ are independent events (no feedback involved). Notably, since we only have access to the predicted token distribution of the language model and no other external augmentation resources, we will take this as our best empirical estimate of the true token distribution, *i.e.* $\mathcal{P}(\mathcal{T} = v) = P_\theta(v \mid x_{1:t-1})$. Therefore, we have:

$$
\begin{aligned}
L[P] &= \sum_{v \in \mathcal{V}} \underbrace{\mathcal{P}(\mathcal{S} = v \mid x_{1:t-1})}_{= P_\theta(v \mid x_{1:t-1})} \underbrace{\mathcal{P}(\mathcal{T} = v \mid x_{1:t-1})}_{= P_\theta(v \mid x_{1:t-1})} \\
&= \sum_{v \in \mathcal{V}} P_\theta(v \mid x_{1:t-1})^2.
\end{aligned}
\tag{2}
$$

We formalize the method as follows:

1. **Determine the threshold probability** $L[P_\theta]$ with equation 2.

2. **Construct the sampling set** $\mathcal{V}_{p\text{-less}}$ with tokens whose probabilities are at least $L[P_\theta]$:

$$
\mathcal{V}_{p\text{-less}} = \{\, v \in \mathcal{V} : P_\theta(v \mid x_{1:t-1}) \geq L[P_\theta] \,\}.
\tag{3}
$$

3. **Sample from** $\mathcal{V}_{p\text{-less}}$ the next token $x_t$ according to the normalized token probabilities $P'_\theta$:

$$
P'_\theta(x_t \mid x_{1:t-1})|_{x_t := v} = \frac{P_\theta(v \mid x_{1:t-1})}{\sum_{v' \in \mathcal{V}_{p\text{-less}}} P_\theta(v' \mid x_{1:t-1})} \quad \text{for} \quad v \in \mathcal{V}_{p\text{-less}}.
\tag{4}
$$

## 3.2 THE $p$-LESS$_{\text{NORM}}$ SAMPLING METHOD

We further explore a variant of $p$-less which effectively relaxes the threshold by subtracting the likelihood of an incorrect random guess normalized to the number of correct outcomes. The resultant $p$-less$_{\text{norm}}$, denoted $\bar{L}[P_\theta]$, is preferable in use cases where diversity is favored over coherence. Formally, we have:

$$
\bar{L}[P_\theta] := L[P_\theta] - \underbrace{\frac{1}{|\mathcal{V}| - 1}}_{\substack{\text{Normalization} \\ \text{constant}}} \times \underbrace{\sum_{u,v \in \mathcal{V}, u \neq v} P_\theta(u \mid x_{1:t-1}) P_\theta(v \mid x_{1:t-1})}_{\substack{\text{Probability of a randomly sampled} \\ \text{and incorrect token}}}
\tag{5}
$$

$$
= \frac{|\mathcal{V}|}{|\mathcal{V}| - 1} L[P_\theta] - \frac{1}{|\mathcal{V}| - 1}
\tag{6}
$$

where $\frac{1}{|\mathcal{V}| - 1}$ gives the ratio of the possible number of correct to incorrect outcomes. The derivation of Eq. equation 6 from equation 5 and additional details of $p$-less$_{\text{norm}}$ are provided in Appendix B.4.

### 3.3 Connection to Rényi Entropies

Our $p$-less threshold can be re-interpreted in connection to established results in information theory, namely the family of Rényi entropies (Rényi, 1961). The Rényi entropy of order $\alpha$ [1] is defined by:

$$H_\alpha(p) = \frac{1}{1-\alpha} \log \sum_i p_i^\alpha \tag{7}$$

In particular, the Rényi entropy of order 2 (aka the collision entropy) is given by

$$H_2(p) = -\log \sum_i p_i^2 = -\log L[P] \tag{8}$$

Since $\log$ is a monotonically increasing function, our $p$-less quantity $L[P]$ increases with decreasing collision entropy. Furthermore, we have

$$H_2(p) = -\log L[P] \leq -\log \sum_i p_i \log p_i = H_1(p)$$
$$\implies L[P] \geq \exp(-H_1(p)), \tag{9}$$

which shows that $L[P]$ is also negatively correlated with the Shannon entropy.

The Rényi entropies of different orders quantify uncertainty with different sensitivities. In particular, the Rényi entropy of order 2 is sensitive to the concentration in the probability mass function and provides a suitable gauge of the global confidence in the model predictions.

Our $p$-less threshold corresponds to the exponential of the negative Rényi entropy of order 2. Intuitively, as Rényi entropy of order 2 increases, $p$-less decreases. By extension of the family of Rényi entropies, we also showed $p$-less to vary in the opposite direction as Shannon entropy (Shannon, 1948). Finally, $p$-less can be extended to a generalized $k$-order threshold within the formalism of Rényi entropy; see Appendix B.5 for details.

### 3.4 Intuition of $p$-less Sampling

We seek to answer the question "Given a probability distribution of tokens, what is a reliable subset of tokens we should sample from?" Rationally, we make use of the *full information in the distribution* to guide our decision, and formulate our method *principled in probability and statistics*. We define our threshold probability $p$-less, for filtering tokens into the sampling set, as the probability of a *randomly selected and correct* token (or most desirable token) in the "ground-truth" sense.

During inference, the token distribution that the large language model predicts is its degree of belief in the next-token ground-truth correctness, based on the ground-truth information it learned during training. Essentially, the token distribution encodes a notion of probabilistic correctness or desirability. This token distribution presents, via $L[P]$, the probability of sampling the ground truth, or equivalently the *probability that a random sampling is correct* (or most desirable), which we interpret as the minimum bar for tokens to qualify for admission into the sampling set. In other words, the tokens that we admit into the sampling set have to be *at least as confident as the random sampling that happens to be correct* (or most desirable) in the ground-truth sense.

---

[1]For special values $\alpha \in \{0, 1, \infty\}$, the definition is extended via limits:

$$H_0(p) = \log |\mathcal{V}|$$
$$H_1(p) = -\sum_i p_i \log p_i$$
$$H_\infty(p) = -\log \max p_i$$

An alternative interpretation of $L[P]$ is that it serves as the unbiased estimator of the second moment of the distribution's probability mass function, $M[P]$, scaled by the vocabulary size $|\mathcal{V}|$:

$$
\begin{aligned}
L[P] &:= \sum_{i=1}^{|\mathcal{V}|} P(x_i)^2 \\
&= |\mathcal{V}| \times \left\{ \frac{1}{|\mathcal{V}|} \sum_{i=1}^{|\mathcal{V}|} P(x_i)^2 \right\} \\
&= |\mathcal{V}| \times M[P]
\end{aligned}
\tag{10}
$$

We regard $p$-less as an information-theoretic approach stronger than other methods, as it incorporates full information in the output token distribution for decoding, and demonstrates compatible interpretations using probability theory (see Sections 3.1 and 3.2), entropies (see Section 3.3) and statistical moments (see Section 3.4 and Appendix B.4). Specifically, $p$-less contrasts with other methods that do not consider the output token distribution (e.g. top-$k$, top-$p$, $\epsilon$-sampling, min-$p$) or only considers it if conditions are met (e.g. $\eta$-sampling). $p$-less is also an empirical approach; it relies on the empirical output token distribution instead of making assumptions in the token distribution parameters or enforcing a target surprisal in the text generation process (e.g. mirostat), thereby eliminating further estimation errors.

### 3.5 ADVANTAGES OF $p$-LESS OVER EXISTING METHODS

$p$-less sampling combines several desirable properties of existing sampling approaches into a single method. First, the truncation threshold utilized in $p$-less sampling dynamically adapts to the entire token probability distribution at each time step. In contrast, existing sampling methods either use a fixed threshold which ignores the current token probability distribution (e.g. top-$p$, top-$k$, $\epsilon$-sampling), set the threshold based on the probability of a single token in the current distribution (e.g. min-$p$), or only considers the token distribution if conditions are met (e.g. $\eta$-sampling). Second, $p$-less produces a bounded and valid truncation threshold which guarantees a non-empty candidate set for sampling, unlike other sampling methods where bounds are not guaranteed and edge cases are resolved with defaults, such as defaulting to the modal token (or top few tokens) if all tokens do not meet the threshold (e.g. $\epsilon$-sampling, $\eta$-sampling, mirostat). Third, the truncation threshold of $p$-less sampling dynamically adjusts with temperature, unlike other methods (e.g. top-$p$, top-$k$, min-$p$, $\epsilon$-sampling) whose hyperparameters are not meaningful when temperature approaches zero or infinity.

Thus, $p$-less uniquely possesses all three of the aforementioned desirable properties of a sampling approach, combining the benefits of existing sampling strategies into a single method. In addition, $p$-less is distinguished from prior approaches in that it is parameter-less. This eliminates the need to tune the sampling method's hyperparameters, which are often sensitive to the generation task.

## 4 EXPERIMENTS

### 4.1 EXPERIMENTAL SETUP

Our experiments were performed using Llama-2-7B (Chat) (Touvron et al., 2023), Mistral-7B (Instruct) (Jiang et al., 2023), and Llama3-70b (Instruct) (Dubey et al., 2024) on two types of tasks: *math and logical reasoning* across the GPQA (Rein et al., 2023), GSM8K (Cobbe et al., 2021), QASC (Khot et al., 2020) & CSQA (Talmor et al., 2019) datasets, and *instruction following creative writing* for the Writing Prompts (Fan et al., 2018a) dataset. We benchmarked our proposed sampling approaches against existing methods including Top-$p$ (Holtzman et al.), Min-$p$ (Nguyen et al., 2024), $\epsilon$-sampling (Freitag et al., 2023), $\eta$- sampling (Hewitt et al., 2022) and Mirostat (Basu et al., 2021) for temperatures ranging from 0.5 to 2.0. We measured accuracy on the math and logical reasoning datasets and computed length-controlled win rate (Dubois et al., 2024) for Writing Prompts using an automated evaluation framework (Li et al., 2023), in addition to conducting a human evaluation. Additional experiment details are provided in Appendix C.1.

| | Llama2-7b | | | | Mistral-7b | | | | Llama3-70b | | | |
|---|---|---|---|---|---|---|---|---|---|---|---|---|
| | CSQA | GPQA | GSM8K | QASC | CSQA | GPQA | GSM8K | QASC | CSQA | GPQA | GSM8K | QASC |
| $\epsilon$ | 0.388 | 0.176 | 0.212 | 0.396 | 0.604 | 0.180 | 0.408 | 0.635 | 0.815 | 0.382 | 0.909 | 0.887 |
| $\eta$ | 0.379 | 0.173 | 0.207 | 0.372 | 0.555 | 0.172 | 0.399 | 0.601 | 0.727 | 0.326 | 0.895 | 0.812 |
| min-$p$ | 0.488 | 0.243 | 0.256 | 0.502 | 0.691 | 0.212 | 0.523 | 0.730 | **0.820** | 0.377 | 0.924 | **0.899** |
| mirostat | 0.410 | 0.212 | 0.201 | 0.419 | 0.635 | 0.216 | 0.392 | 0.684 | 0.776 | 0.366 | 0.879 | 0.879 |
| top-$p$ | 0.410 | 0.172 | 0.210 | 0.393 | 0.580 | 0.172 | 0.438 | 0.604 | 0.713 | 0.320 | 0.870 | 0.778 |
| $p$-less | **0.503** | 0.242 | **0.267** | 0.537 | **0.697** | **0.239** | 0.562 | 0.736 | 0.819 | 0.387 | **0.932** | 0.894 |
| $p$-less$_{\text{norm}}$ | **0.503** | **0.248** | **0.267** | **0.538** | 0.692 | 0.222 | **0.564** | **0.739** | 0.819 | **0.391** | 0.930 | 0.894 |

**Table 1:** AUC of LLama2-7b, Mistral-7b, and Llama3-70b across different sampling methods for math and logical reasoning datasets. The best AUC is in **bold** and the second best is underlined.

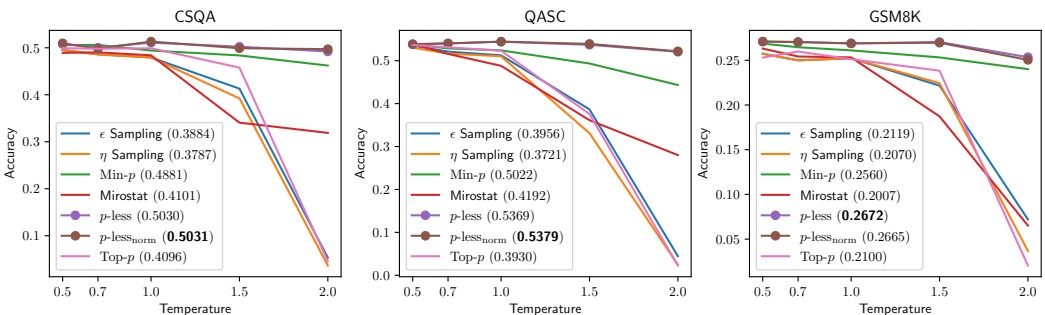

**Figure 2:** Accuracy vs. temperature curves of each method on CSQA, QASC, and GSM8k using Llama-2-7b. AUC values achieved by each method are provided in the legend (in parentheses) with the best AUC in **bold**.

## 4.2 MATH AND LOGICAL REASONING RESULTS

To perform a fair comparison between methods across temperatures, we computed the area under the accuracy-temperature curve for each method (normalized between 0.0 and 1.0), which we term AUC. Complete AUC results for the math & logical reasoning datasets are provided in Table 1. For Llama2-7b, the AUCs of $p$-less or $p$-less$_{\text{norm}}$ outperform the other methods across all datasets. The results for Mistral-7b are consistent with those of Llama2-7b: the AUCs of both $p$-less and $p$-less$_{\text{norm}}$ outperform all other methods across every dataset. For Llama3-70b, the AUCs of $p$-less and $p$-less$_{\text{norm}}$ are either the highest or second highest within 0.005 of the highest. Across the four datasets on Llama2-7b and Mistral-7b, $p$-less and $p$-less$_{\text{norm}}$ perform superior to the other methods at temperatures 1.0 and above, and are competitive at temperatures below 1.0 (see Figure 2, Table 5, Figure 9 and Figure 10). Figure 2 shows all sampling methods degrade at various rates with increasing temperature, while $p$-less and $p$-less$_{\text{norm}}$ are robust to high temperatures and widen their performance gap against other methods. For Llama3-70b, $p$-less and $p$-less$_{\text{norm}}$ perform superior to the other methods across all temperatures on GSM8K, on low and high temperatures for GPQA, and on high temperatures for CSQA and QASC; with the rest being competitive (see Table 5 and Figure 11).

In addition to the commonly-adopted default hyperparameter configurations we employed for the baseline methods (consistent with those employed or supported by prior work such as (Nguyen et al., 2024), (Zhou et al., 2024), (Zhu et al., 2024) and (Hewitt et al., 2022); see Appendix C.2 for the hyperparameter values), we conducted evaluations over a wide range of hyperparameter values for these baseline sampling methods (see Table 8 for full results), where we still find $p$-less and $p$-less$_{\text{norm}}$ to generally have the highest AUC, and competitive to highest accuracies.

We also compared $p$-less at temperature 1.0 to greedy decoding and beam search using Mistral-7b, and provide a discussion on the versatility of $p$-less sampling in terms of producing strong results across a variety of tasks and datasets (see Appendices C.4, C.4.1 and Table 6). In low-entropy inference tasks which tend to favor greedy decoding (e.g. math and logical reasoning), $p$-less can achieve similar or better performance. Additionally, $p$-less performs significantly better than greedy decoding at higher-entropy tasks like creative writing, demonstrating that it is not simply an argmax-seeking procedure. Rather, $p$-less dynamically adapts to the entropy of the token distribution to produce high-quality outputs across a range of different settings without the need to change sampling strategies

or hyperparameters depending upon the task. Furthermore, $p$-less can achieve higher diversity values (than greedy decoding) by increasing the temperature (see Appendix C.8 and Table 11).

We further ablated with the DeepSeek-R1-Distill-Qwen-7B (Guo et al., 2025) reasoning model, which consistently shows $p$-less and $p$-less$_{norm}$ maintain strong performance across all temperature settings, including being significantly superior to the second best at temperature 2.0 (see Appendix C.5).

### 4.3 CREATIVE WRITING RESULTS

We provide results for the Writing Prompts dataset using the length-controlled win rate metric (Dubois et al., 2024). Specifically, we sample one generation per method and temperature for a subset of 100 prompts and use the response generated by default sampling (i.e. without truncation at temperature 1.0) as reference. Table 2 summarizes the results. All methods except $p$-less generally exhibit significant performance degradation as temperature increases. In contrast, the performance of $p$-less remains relatively stable and is superior to all other methods at temperatures > 1.0. This demonstrates how $p$-less excels in the domain of creative writing while avoiding the degradation of text quality exhibited by other methods at higher temperatures. In addition, we performed a human evaluation using the Llama2-7b generations for 100 sampled prompts, and obtained directional consistency between our human and automated evaluations, which provides further evidence of the effectiveness of $p$-less sampling for creative writing (see Appendix A).

| Model | Temperature | $\epsilon$-sampling | $\eta$-sampling | min-$p$ | mirostat | top-$p$ | $p$-less | $p$-less$_{norm}$ |
|---|---|---|---|---|---|---|---|---|
| Llama-2-7b | 1.0 | **62.18** | 58.76 | 57.48 | 56.94 | 62.07 | 55.08 | 58.74 |
| | 1.5 | 1.99 | 1.46 | 58.17 | 5.33 | 4.39 | 58.23 | **59.58** |
| | 2.0 | 0.00 | 0.00 | 48.94 | 26.88 | 0.00 | **65.64** | 59.29 |
| Mistral-7b | 1.0 | 60.90 | 59.82 | 66.49 | 62.26 | 65.68 | **68.90** | 67.49 |
| | 1.5 | 3.71 | 0.00 | 62.17 | 12.08 | 0.00 | **66.97** | 66.89 |
| | 2.0 | 0.00 | 0.00 | 54.11 | 40.33 | 0.00 | 60.32 | **61.99** |

**Table 2:** Length-controlled win rate for 100 sampled prompts from the Writing Prompts dataset.

## 5 ANALYSIS

### 5.1 EFFICIENCY OF $p$-LESS AND OTHER METHODS

To compare the inference-time efficiency of $p$-less to other sampling methods, we calculated the average sampling time per token over 200 Mistral-7b generations for GSM8K and GPQA. The results are summarized in Table 3. $p$-less achieves the fastest average sampling speed per token, with a 22% reduction in inference speed relative to min-$p$. The superior efficiency of $p$-less is statistically significant at the 5% level against the baseline sampling methods except $\eta$-sampling (see Appendix C.11 and Table 14). We attribute this greater efficiency to the fact that unlike other sampling approaches, $p$-less neither require sorting the token probability distribution to compute the truncation threshold, nor require determining the most confident token(s) for default inclusion into the sampling set in order to deal with the edge case of no tokens satisfying the truncation threshold. The other sampling approaches which we compare $p$-less to implement at least one of these two operations. Without the need to sort, $p$-less overall time complexity is reduced from $O(|\mathcal{V}| \log |\mathcal{V}|)$ to $O(|\mathcal{V}|)$; without the need to identify the most confident token(s), at least another $O(|\mathcal{V}|)$ operation is avoided for the case of the single most confident token. Additionally, we captured fine-grained CPU processing times and RAM usage during sampling for top-$p$, min-$p$ and $p$-less; these results are consistent with Table 3, showing that $p$-less consumes the least CPU time and RAM (see Appendix C.11, Figures 16 and 17).

| | $\epsilon$-sampling | $\eta$-sampling | min-$p$ | mirostat | top-$p$ | $p$-less |
|---|---|---|---|---|---|---|
| Mean | 0.02259 | 0.02210 | 0.02497 | 0.02278 | 0.02362 | **0.01942** |
| Standard Deviation | 0.01308 | 0.01277 | 0.01425 | 0.01339 | 0.00879 | 0.00899 |
| Standard Error of Mean | 0.0009 | 0.0014 | 0.0013 | 0.0013 | 0.0013 | 0.0009 |

**Table 3:** Average sampling time per token (in seconds) for $p$-less and other methods.

We also observed that $p$-less is often more efficient than other methods in terms of producing shorter generation lengths, despite achieving higher overall task-specific accuracy. Table 12 of Appendix C.9 provides the mean generation length produced by each sampling method across math and logical reasoning datasets, models and temperatures. In the case of Llama2-7b, $p$-less or $p$-less$_{norm}$ produce the shortest generations on average across all temperatures for CSQA, QASC and GSM8K. These shorter generations do not sacrifice the model's reasoning performance, as evidenced by the superior accuracy achieved by $p$-less and $p$-less$_{norm}$ on these datasets while still being diverse (see Figures 2 and 3). These results provide empirical support to how our principled approach to truncation-based sampling improves both generation quality and inference-time efficiency. In Appendices C.9 and C.9.1, we provide a discussion on the generation lengths of top-$p$, min-$p$ and $p$-less via the lenses of token probability distribution entropy and size of tokens admitted for sampling.

## 5.2 DIVERSITY ANALYSIS

We compute the $n$-gram repetition diversity metric proposed by Su et al. (2022) for QASC; higher values indicate greater diversity. Table 4 shows that at temperatures $\leq 1$, all methods produce similar diversity values between 0.62-0.64. At higher temperatures, $p$-less and $p$-less$_{norm}$ exhibits similar diversity to min-$p$, but lower diversity than other sampling methods. However, greater diversity at these higher temperatures leads to lower answer accuracy. Figure 3 illustrates the relationship between generation diversity and answer accuracy for Llama2-7b using sampling methods and temperature settings which achieved an overall mean accuracy $> 0.5$ on QASC. This plot shows that $p$-less and $p$-less$_{norm}$ produce higher accuracy at a given level of generation diversity than other sampling methods, exhibiting a pareto dominance along the diversity-accuracy frontier. We provide additional diversity results for all three models on the four math and logical reasoning datasets in Table 10 of Appendix C.8. These results show that as temperature increases, $p$-less exhibits a reasonable increase in diversity while other methods experience diversity spikes which compromise their task-specific reasoning capability (see Table 5). We further ablated $p$-less with temperature values $> 2.0$, and show that $p$-less allows for significant increase in diversity based on temperature; it exhibits a similar increasing trend of diversity with temperature as other methods such as min-$p$, differing primarily in the magnitude of diversity increase with each temperature step (see Appendix C.8 and Table 11).

| Temperature: | 0.5 | 0.7 | 1.0 | 1.5 | 2.0 |
|---|---|---|---|---|---|
| $\epsilon$ Sampling | 0.63 | 0.63 | 0.63 | 0.75 | 0.98 |
| $\eta$ Sampling | 0.62 | 0.63 | 0.63 | 0.79 | 1.00 |
| Min-$p$ | 0.63 | 0.62 | 0.62 | 0.62 | 0.64 |
| Mirostat | 0.62 | 0.63 | 0.63 | 0.79 | 0.76 |
| Top-$p$ | 0.63 | 0.63 | 0.62 | 0.73 | 1.00 |
| $p$-less | 0.63 | 0.64 | 0.63 | 0.63 | 0.64 |
| $p$-less$_{norm}$ | 0.63 | 0.64 | 0.63 | 0.63 | 0.64 |

**Table 4:** QASC diversity by method & temperature

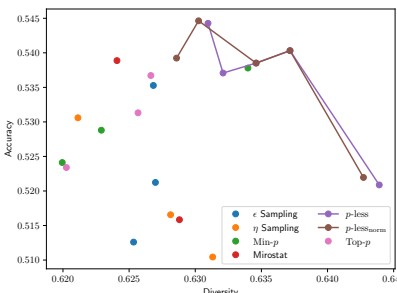

**Figure 3:** QASC accuracy vs. diversity

## 5.3 QUALITATIVE ANALYSIS

We found that where the other methods fail on specific math and logical reasoning examples at low temperatures, $p$-less and $p$-less$_{norm}$ generate correct reasoning and the correct final answer. When temperatures are high enough for the methods to fail, $p$-less and $p$-less$_{norm}$ still generate correct reasoning, including unprompted self-verification and self-correction on initial wrong answers. At high temperatures, the generations of other methods veer off-track or exhibit hallucination and degeneracy, whereas generations from $p$-less and $p$-less$_{norm}$ remain factual and relevant. Illustrative examples are discussed in Appendix C.12.2. We also provide creative writing examples where $p$-less is preferred to default sampling by human evaluators and the automated evaluation framework in Appendix C.12.3. Finally, we discuss two typical failure patterns of $p$-less in Appendix C.13.

## 5.4 CASE STUDY: ROBUSTNESS UNDER HIGH ENTROPY

**GSM8K example at temperature 2.0:** We examine responses to a question which involves interpreting the order of events, where the response produced by min-*p* committed a critical misinterpretation error. We supplement the responses with a plot of the entropy and number of admitted tokens at each generation step. Notably, even though the average entropy and fluctuation in entropy is significantly smaller for min-*p* than *p*-less, the number of admitted tokens after the truncation is higher. This contrast illustrates how *p*-less enforces a form of entropy-aware regularization, mitigating token overcommitment in ambiguous regions and preserving semantic fidelity. We delve into further quantitative analysis of the entropy and token count distributions in Appendix C.10.

**Question:** *"Brian's friend Bobby has 5 fewer than 3 times as many video games as Brian does. If Brian has 20 video games but lost 5 right before the comparison was made, how many does Bobby have?"*

**Min-*p*:** *Let's think step by step. First, we need to find out how many video games Brian had originally. If he has 20 video games now and lost 5, then he must have had 20 + 5 = «20+5=25»25 video games originally. Bobby has 3 times as many video games as Brian has, minus 5. So Bobby has 3 * 25 - 5 = 75 - 5 = «75-5=70»70 video games. The answer is 70.*

The flawed reasoning step in red above coincides with a region of spikes in the number of admitted tokens (circled region) in Figure 4a. Separately, at the mid-point of the generation, there is an unexpected large spike in token admission despite a corresponding ordinary entropy.

**p-less:** *Let's think step by step. Brian had 20 video games, but lost 5, so he has 20 - 5 = 15 video games now. Bobby has 3 times as many video games as Brian, so Bobby has 3 * 15 = 45 video games. But Bobby has 5 fewer than 3 times as many video games as Brian, so Bobby has 45 - 5 = 40 video games. The answer is 40.*

As seen in Figure 4b, even while the entropy is extraordinarily high, the number of admitted tokens remains well-regulated, showing *p*-less's selectivity in (desirable) tokens and effectiveness in subduing verbosity.

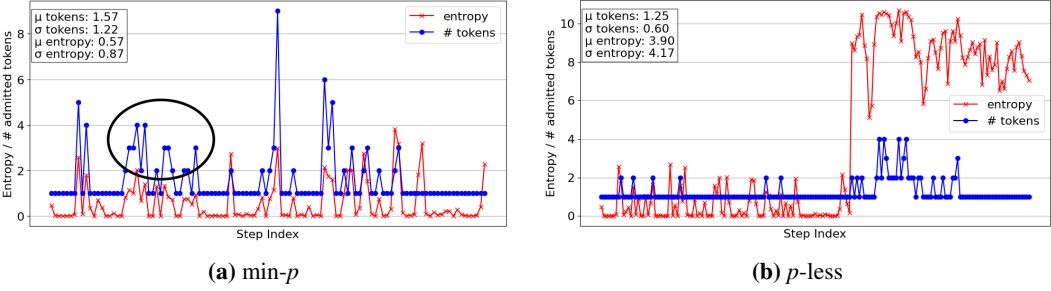

**(a)** min-*p*          **(b)** *p*-less

**Figure 4:** Step-wise entropy and number of admitted tokens for a GSM8K question answered with Llama3-70b.

## 6 CONCLUSION

We presented *p*-less sampling: a hyperparameter-free truncation methodology for sampling-based decoding. *p*-less combines several desirable properties of existing sampling methods into a single approach while eliminating the need to tune sampling hyperparameters required by other methods. Our experimental results across three LLMs and five datasets spanning math, logical reasoning, and creative writing tasks demonstrated how *p*-less consistently achieves strong performance across a range of different temperature values. In contrast, other evaluated methods exhibit significant declines in performance as temperature increases. We further showed how *p*-less achieves greater inference-time efficiency than other methods through faster token sampling speeds and shorter generation lengths. Our work highlights how grounding LLM decoding in information theory results in a principled sampling approach which is both intuitive and empirically effective.

## 7 REPRODUCIBILITY STATEMENT

We will make our source code publicly available upon publication in order to facilitate future efforts to reproduce our main experimental results. In addition, we have provided complete details of models, datasets, and evaluation metrics in Appendix C.1. We provide details on hyperparameters utilized in our experiments in Appendix C.1 and Appendix C.2. From Appendix C.3 to Appendix C.11, we detailed our methods on analyses. In Appendix C.12 and Appendix C.13, we provide details of prompts used in our experiments and provide multiple generation examples. To the best of our knowledge, the documentation in this manuscript contains all details necessary to fully reproduce our results.

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

## A  HUMAN EVALUATION

As the four math and logical reasoning datasets contain annotated labels and will therefore not benefit significantly from a human evaluation, we focused our human annotation study on the creative writing task. Specifically, three authors of this work and three non-author annotators evaluated Llama2-7b generations produced by $p$-less and default sampling for 100 sampled prompts from the Writing Prompts dataset, producing a total of 4 labels for each story pair. The $p$-less samples were generated with a temperature of 2.0 and were evaluated in the same pairwise comparison setting as our length-controlled win rate evaluations. $23.7\%$ of story pairs received unanimous agreement among the annotators and $26.9\%$ received a tie; for the remaining stories, we use the majority vote to obtain a label. Overall we found that $p$-less won the majority vote $58.8\%$ of the time, with the win rate further increasing to $72.7\%$ for the $23.7\%$ of stories which had unanimous agreement. The win rates for annotations produced by authors were $57.6\%$, $54.3\%$, $57.1\%$ while the win rate for the non-author annotations was $54.9\%$ (these values are slightly lower than the majority vote win rate due to the presence of ties), indicating that the labels produced by authors and non-author annotators are distributionally similar. The directional consistency of our human and automated evaluations (Table 2) provides further evidence of the effectiveness of $p$-less sampling for creative writing.

# B  ADDITIONAL PROPERTIES OF $p$-LESS AND $p$-LESS$_{\text{NORM}}$ SAMPLING

## B.1  FRIEDMAN'S INDEX (PROBABILITY) OF COINCIDENCE

In cryptography, the Friedman's Index (Probability) of Coincidence (Friedman, 1922), *IC*, for an infinitely long encryption can be approximated with the $p$-less quantity.

$$
\begin{aligned}
IC &= \frac{\sum_i n_i(n_i - 1)}{N(N - 1)} \\
\lim_{n \to \infty} IC &= \lim_{n \to \infty} \frac{\sum_i n_i(n_i - 1)}{N(N - 1)} \\
&= \lim_{n \to \infty} \sum_i \left(\frac{n_i}{N}\right)\left(\frac{n_i - 1}{N - 1}\right) \\
&= \lim_{n \to \infty} \sum_i (p_i)\left(\frac{n_i - 1}{N - 1}\right) \\
&= \sum_i (p_i)(p_i) \\
&= \sum_i p_i^2
\end{aligned}
\tag{11}
$$

## B.2  UNBIASED ESTIMATOR OF THE SECOND MOMENT OF THE PROBABILITY MASS FUNCTION

$p$-less is also the unbiased estimator of the second moment of the probability mass function of the distribution, $M[P]$, multiplied by the vocabulary size $|\mathcal{V}|$:

$$
\begin{aligned}
L[P] &:= \sum_{v \in \mathcal{V}} P(v \mid x_{1:t-1})^2 \\
&= |\mathcal{V}| \times \left\{ \frac{1}{|\mathcal{V}|} \sum_{v \in \mathcal{V}} P(v \mid x_{1:t-1})^2 \right\} \\
&= |\mathcal{V}| \times M[P] \\
&\propto M[P]
\end{aligned}
\tag{12}
$$

This demonstrates that the $p$-less quantity $L[P]$ is directly proportional to the unbiased estimator of the second moment of the probability mass function of the token distribution $M[P]$. Notably, as the distribution becomes more uniform (having more entropy), its second moment decreases, and the $p$-less quantity decreases. With a smaller $p$-less quantity, the method intuitively admits more tokens.

Our principled approach in deriving and proposing the use of $p$-less is further supported by these connections. We invite the community to explore further methods grounded in information theory.

## B.3  DESIRABLE PROPERTIES OF $p$-LESS SAMPLING

We compute a threshold probability for filtering tokens from the large language model's output token distribution for sampling, grounded on the *likelihood of a correct random guess given the distribution*, as a principled way to decide what to sample.

**Proposition 1**

Let the *likelihood of a correct random guess* of a probability mass function $P$ with outcomes $\{x_1, \cdots, x_c\}$ be $L[P]$. Then, we have the following bounds for $L[P]$

$$\frac{1}{c} \leq L[P] \leq \max_i P(x_i). \tag{13}$$

**Proof of Proposition 1**

By definition,

$$L[P] := \sum_{i=1}^{c} P(x_i)^2$$

$$0 \leq P(x_i) \leq 1$$

$$\sum_{i=1}^{c} P(x_i) = 1$$

*Lower bound*

By the Cauchy-Schwarz inequality,

$$\left( \sum_{i=1}^{c} P(x_i) \times 1 \right)^2 \leq \sum_{i=1}^{c} P(x_i)^2 \sum_{i=1}^{c} 1^2$$

$$\underbrace{\left( \sum_{i=1}^{c} P(x_i) \right)^2}_{=1} \leq \left( \sum_{i=1}^{c} P(x_i)^2 \right) \times c$$

$$\frac{1}{c} \leq L[P]$$

*Upper bound*

$$P(x_i) \leq \max_i P(x_i)$$

$$\sum_{i=1}^{c} P(x_i)\Big\{ P(x_i) \Big\} \leq \sum_{i=1}^{c} P(x_i)\Big\{ \max_i P(x_i) \Big\}$$

$$\sum_{i=1}^{c} P(x_i)^2 \leq \sum_{i=1}^{c} P(x_i) \max_i P(x_i)$$

$$L[P] \leq \max_i P(x_i)$$

*Therefore,*

$$\frac{1}{c} \leq L[P] \leq \max_i P(x_i).$$

The lower bound of $\frac{1}{c}$ and the upper bound of $\max_i P(x_i)$ for $L[P]$ guarantee a valid threshold for filtering a non-empty candidate set for sampling. The lower bound for $L[P]$ removes from consideration, any outcome $x_j$ whose likelihood is less than the *likelihood of a correct random guess for a uniform distribution*, or equivalently is less than uniformly probable.

In addition to the above bounds, our $p$-less threshold $L[P]$ varies in the opposite direction as the uncertainty or entropy of the distribution, essentially considering more tokens with lower probabilities as the uncertainty or entropy of the distribution increases, which is a befitting relationship for the

trade-off between the number of tokens to consider for sampling and the uncertainty or entropy of the token distribution.

### B.4 PROPERTIES OF $p$-LESS$_{\text{NORM}}$

As introduced in 3.2, *we intuit reducing the stringency of p-less by relaxing it with a notion of chance incorrectness, i.e. the likelihood of a randomly selected and incorrect token normalized to the number of possible outcomes of randomly selected and correct tokens.*

Whereas we have shown $p$-less $L[P]$ is the unbiased estimator of the second moment of the distribution's probability mass function, $M[P]$ multiplied by the vocabulary size $c$, we have a similar result for $p$-less$_{\text{norm}}$ $\bar{L}[P]$, as formalized in the following proposition.

---

**Proposition 2**

The $p$-less$_{\text{norm}}$ $\bar{L}[P]$ is equivalent to the unbiased estimator of the second central moment $\bar{M}[P]$ of a probability mass function $P$, multiplied by the vocabulary size $c$.

---

**Proof of Proposition 2**

By definition,

$$\bar{L}[P] = L[P] - \frac{1}{c-1} \sum_{j \neq i} P(x_i) P(x_j)$$

$$\frac{1}{c}\bar{L}[P] = \frac{1}{c} \sum_{i=1}^{c} P(x_i)^2 - \frac{1}{c(c-1)} \sum_{j \neq i} P(x_i) P(x_j)$$

$$= \frac{1}{c} \sum_{i=1}^{c} P(x_i)^2 - \frac{1}{c(c-1)} \left[ 1 - \sum_{i=1}^{c} P(x_i)^2 \right]$$

$$= \frac{1}{c} \sum_{i=1}^{c} P(x_i)^2 + \frac{1}{c(c-1)} \sum_{i=1}^{c} P(x_i)^2 - \frac{1}{c(c-1)}$$

$$= \frac{(c-1)+1}{c(c-1)} \sum_{i=1}^{c} P(x_i)^2 - \frac{1}{c(c-1)}$$

$$= \frac{c}{c(c-1)} \sum_{i=1}^{c} P(x_i)^2 - \frac{1}{c(c-1)}$$

$$= \frac{1}{c-1} \sum_{i=1}^{c} P(x_i)^2 - \frac{1}{c(c-1)}$$

$$= \frac{1}{c-1} \left[ \sum_{i=1}^{c} P(x_i)^2 - \frac{1}{c} \right]$$

$$= \frac{1}{c-1} \left[ \sum_{i=1}^{c} P(x_i)^2 - \frac{2}{c} \sum_{i=1}^{c} P(x_i) + \frac{1}{c} \right]$$

$$= \frac{1}{c-1} \sum_{i=1}^{c} \left[ P(x_i) - \frac{1}{c} \right]^2$$

$$= \bar{M}[P]$$

---

Similar to $p$-less, as the distribution becomes more uniform (having more entropy), its second central moment decreases, and the $p$-less$_{\text{norm}}$ quantity decreases. With a smaller $p$-less$_{\text{norm}}$ quantity, the method intuitively admits more tokens.

The $p$-less$_{\text{norm}}$ quantity $\bar{L}[P]$ retains similar desirable properties as $p$-less on valid bounds that are stricter than the unit interval and opposite relationship with the uncertainty or entropy of the distribution. We further prove the $p$-less$_{\text{norm}}$ quantity to be bounded between $p$-less and $p$-less less the uniform likelihood.

### Proposition 3

The $p$-less$_{\text{norm}}$ bounds are relaxed from $p$-less bounds. Specifically, we have

$$\bar{L}[P] = \frac{c}{c-1} L[P] - \frac{1}{c-1} \tag{14}$$

$$L[P] - \frac{1}{c} \leq \bar{L}[P] \leq L[P] \tag{15}$$

$$0 \leq \bar{L}[P] \leq \max_i P(x_i). \tag{16}$$

### Proof of Proposition 3

To show Eq.equation 14, by definition, we have

$$\bar{L}[P] := L[P] - \frac{1}{c-1} \sum_{j \neq i} P(x_i)P(x_j)$$

$$= L[P] - \frac{1}{c-1}\left[1 - \sum_{i=1}^{c} P(x_i)^2\right]$$

$$= L[P] - \frac{1}{c-1}\left(1 - L[P]\right)$$

$$= \frac{c}{c-1} L[P] - \frac{1}{c-1}.$$

To show Eq.equation 15, we use Eq.equation 13.

$$L[P] - \bar{L}[P] = \frac{1}{c-1}\left(1 - L[P]\right)$$

$$L[P] - \bar{L}[P] \leq \frac{1}{c-1}\left(1 - \frac{1}{c}\right) \qquad \because \frac{1}{c} \leq L[P]$$

$$= \frac{1}{c}$$

$$L[P] - \bar{L}[P] \geq \frac{1}{c-1}\left(1 - 1\right) \qquad \because L[P] \leq \max_i P(x_i) \leq 1$$

$$= 0$$

$$\therefore L[P] - \frac{1}{c} \leq \bar{L}[P] \leq L[P].$$

To show Eq.equation 16,

$$\bar{L}[P] \geq \underbrace{L[P]}_{\geq \frac{1}{c}} - \frac{1}{c}$$

$$\geq \frac{1}{c} - \frac{1}{c} = 0$$

$$\bar{L}[P] \leq L[P] \leq \max_i P(x_i)$$

$$\therefore 0 \leq \bar{L}[P] \leq \max_i P(x_i).$$

The *normalized likelihood of a correct random guess* $\bar{L}[P]$ is at least the *likelihood of a correct random guess* $L[P]$ less $\frac{1}{c}$ and at most the *likelihood of a correct random guess* $L[P]$. $\bar{L}[P]$ has

the lower bound of $0$ inclusive and the upper bound of $\max_i P(x_i)$ inclusive. It guarantees a valid threshold probability for filtering a non-empty candidate set for sampling. $\bar{L}[P]$ is more permissive than the $L[P]$ and can include more outcomes, making it better suited when sampling diversity is preferable.

### B.5  GENERALIZATION OF THE $p$-LESS SAMPLING METHOD

We extend $p$-less $L[P]$, which is grounded on the second moment of the distribution's probability mass function, to a generalized *k-order* threshold $G[P]_k$ within the formalism of Rényi entropy. Specifically, we define

$$G[P]_k = \frac{1}{\exp^{H_k(p)}} \tag{17}$$

where $H_k$ denotes the Rényi entropy of order $k$. Given that the Rényi entropy is monotonically decreasing with $k$, the corresponding *k-order* threshold $G[P]_k$ increases with $k$. Notably, in the asymptotic regime where $k \to 0$, $G[P]_k$ converges to $1/n$, corresponding to uniform sampling. Conversely, in the limit $k \to \infty$, $G[P]_k$ approaches $\max p_i$, which recovers the behavior of greedy decoding.

### B.6  SYNTHETIC TOKEN PROBABILITY DISTRIBUTIONS AND TRUNCATION THRESHOLDS FROM SAMPLING METHODS

We visualize the truncation thresholds of top-$p$, min-$p$, $p$-less and $p$-less$_{\text{norm}}$ through various token probability distribution charts. Figures 5, 6, 7 and 8 show the effect of temperature, vocabulary size and distribution profiles on the size of the admitted tokens for these sampling methods. Specifically, figure 5 shows how $p$-less and $p$-less$_{\text{norm}}$ admit token progressively rather than exponentially as temperature changes incrementally; figure 6 shows how $p$-less and $p$-less$_{\text{norm}}$ operate according to the entropy level of the various distribution profiles, admitting more tokens when the entropy is high and vice versa; figures 7 and 8 show how $p$-less and $p$-less$_{\text{norm}}$ admit tokens from the long tail in moderate to large vocabulary size settings.

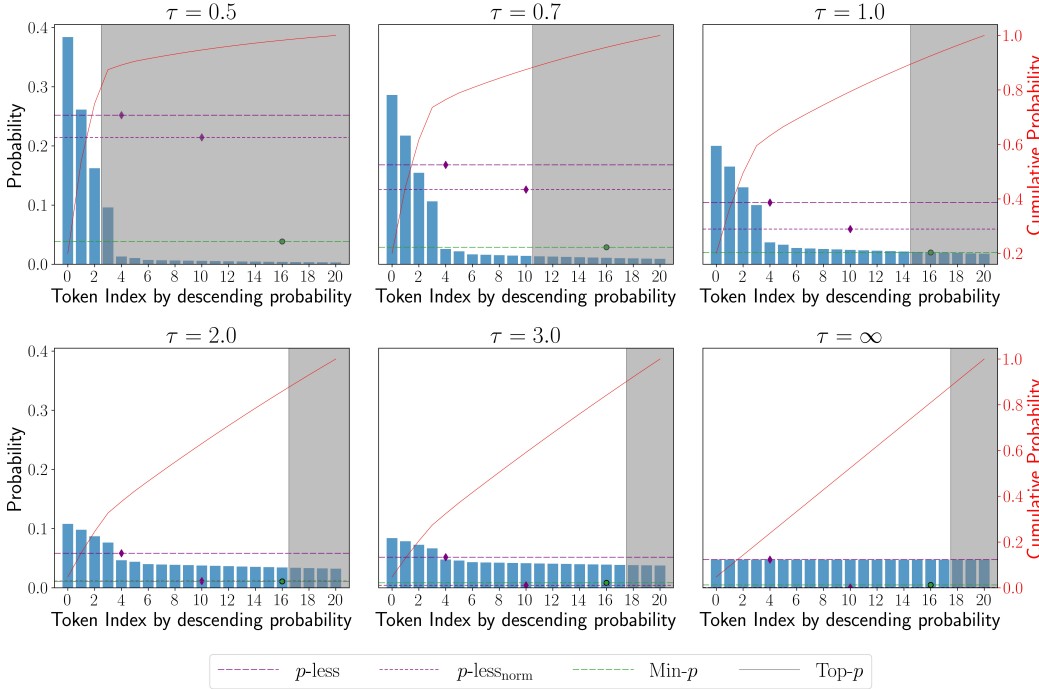

**Figure 5:** Effect of temperature on a fixed token logits distribution with small vocabulary size.

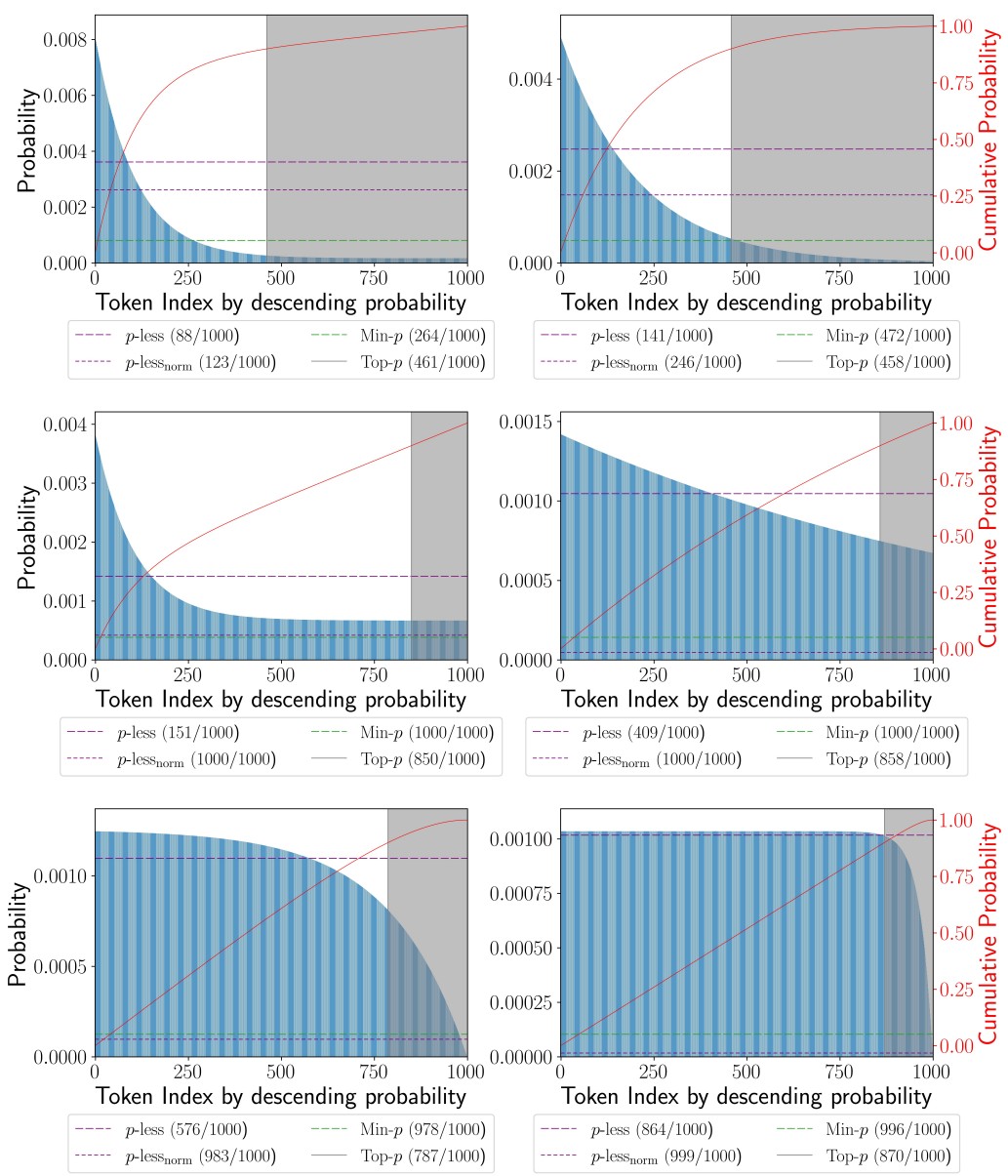

**Figure 6:** Effect of different token probability distribution profiles with a moderate vocabulary size. The legend shows the size of the admitted tokens for each sampling method in parentheses.

# C  ADDITIONAL EXPERIMENTAL DETAILS AND RESULTS

## C.1  ADDITIONAL DETAILS OF EXPERIMENTAL SETUP

To validate the effectiveness of $p$-less sampling, we conducted extensive experiments comparing it to other sampling approaches using three LLMs and five datasets across two different tasks.

**Models.**  We used Llama-2-7B (Chat) (Touvron et al., 2023), Mistral-7B (Instruct) (Jiang et al., 2023), and Llama3-70b (Instruct) (Dubey et al., 2024) as they are finetuned to follow instructions, either in dialogue or directly, suitable for our tasks that require the ability to follow task instructions to generate coherent responses. In addition, two models are similarly sized, allowing us to validate if

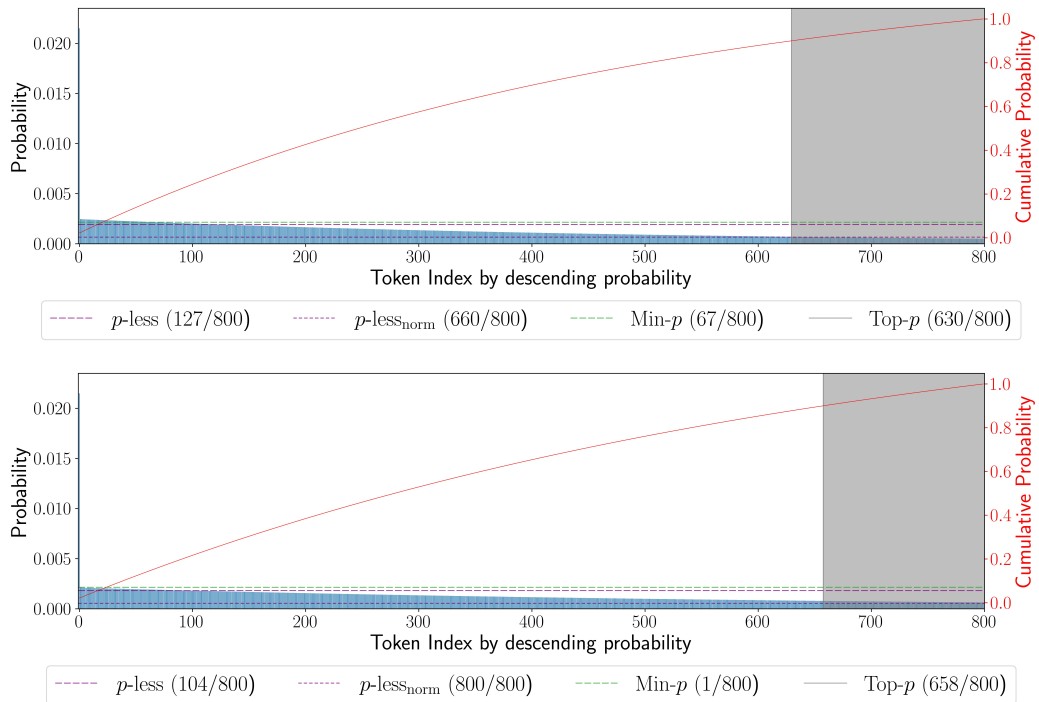

**Figure 7:** Effect of long-tail token probability distributions with a moderate vocabulary size. The legend shows the size of the admitted tokens for each sampling method in parentheses.

our results are consistent across different size-controlled LLMs, while the third model enables us to generalize our results to a significantly larger model.

**Tasks.** We identified two tasks relevant for comparing our $p$-less method with other truncation and sampling methods, namely *math and logical reasoning*, and *instruction following creative writing*.

**Benchmark Datasets.** To support a robust evaluation of our *p-less* method with the other truncation and sampling methods, we used five diverse datasets to conduct our experiments.

- **GPQA: Graduate-Level Reasoning** on expert-level science questions (Rein et al. (2023))
- **GSM8K: Grade School Math** word problems, linguistically diverse (Cobbe et al. (2021))
- **QASC: Question Answering via Sentence Composition** requiring valid compositions of facts using commonsense reasoning (Khot et al. (2020))
- **CommonsenseQA (CSQA): Question Answering** beyond given context requiring drawing from prior **common sense** knowledge (Talmor et al. (2019))
- **WP: Writing Prompts** for open-ended creative story generation (Fan et al. (2018a))

Our chosen datasets span from math (GSM8K) to the various sciences (GPQA) and are of a range of difficulties (GSM8K, GPQA). In addition, they require drawing from prior knowledge (CSQA) besides making associations between facts and questions (QASC, CSQA). We included open-ended generation to creative story writing (WP). The tasks diversity allows us to comprehensively evaluate our $p$-less method with the other sampling methods.

**Temperatures.** We applied temperatures between 0.5 and 2.0 for the main experiments, and extended it to 2.5 for additional diversity ablation on $p$-less. Our $p$-less sampling method uses probabilities after the application of temperature to calculate the threshold for truncation. We regard temperature 2.0 as a reasonable upper limit to encourage diversity in the LLM responses, supported by earlier works such as (Nguyen et al., 2024) which explored temperature settings up to

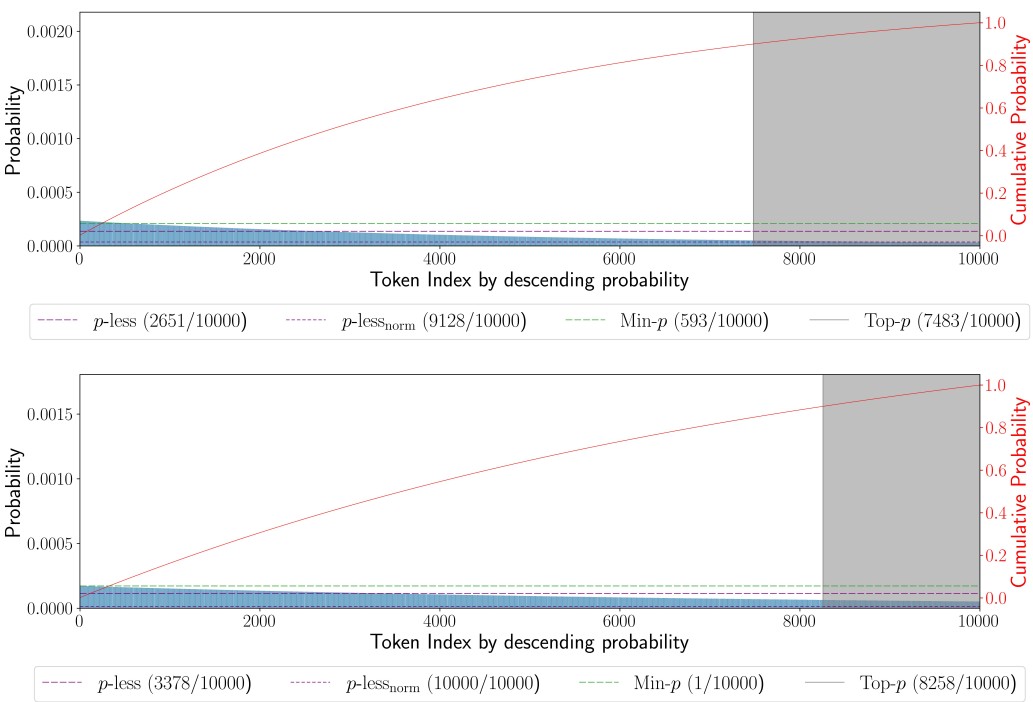

**Figure 8:** Effect of long-tail token probability distributions with a large vocabulary size. For simplicity, we include a modal token that is on the left-most of each distribution chart (not clearly visible due to the sheer size of the vocabulary). The legend shows the size of the admitted tokens for each sampling method in parentheses.

3.0. Additionally, we observed better performance on the Writing Prompts creative writing dataset at a temperature of 2.0 and therefore decided to include it in all of our experimental settings for consistency.

**Sampling Methods and Hyperparameters.** We compared $p$-less sampling with baseline sampling methods such as top-$p$, min-$p$, $\epsilon$-sampling, $\eta$-sampling and mirostat. We applied temperatures between 0.5 and 2.0 and utilize commonly-adopted default hyperparameter configurations for each method other than $p$-less (which is hyperparameter-less), consistent with those employed or supported by prior work such as (Nguyen et al., 2024), (Zhou et al., 2024), (Zhu et al., 2024) and (Hewitt et al., 2022) (see Appendix C.2 for the hyperparameter values). Further to using these hyperparameter configurations, we conducted evaluations over a wide range of hyperparameter values for these baseline sampling methods (see Appendix C.6 for complete results).

**Evaluation Metrics.** We measured accuracy on the math and logical reasoning datasets GPQA, GSM8K, QASC and CSQA. For the creative writing dataset WP, we computed win rate and length-controlled win rate (Dubois et al. (2024)) using an automated evaluation framework (Li et al. (2023)), and further conducted a human evaluation.

## C.2 HYPERPARAMETERS UTILIZED FOR MAIN EXPERIMENTAL RESULTS

Consistent with the commonly-adopted default hyperparameter configurations employed or supported by prior works (Nguyen et al., 2024), (Zhou et al., 2024), (Zhu et al., 2024) and (Hewitt et al., 2022), we utilize these hyperparameter configurations for the baseline sampling methods in our main experiments. Specifically, we set $p = 0.9$ for Top-$p$ and $p = 0.1$ for Min-$p$. For $\epsilon$ and $\eta$ sampling, we set the hyperparameter value to 0.0002. Finally, we set the hyperaparameter value to 4.0 for Mirostat.

| | | CSQA | | | | | GPQA | | | | | GSM8K | | | | | QASC | | | | |
|---|---|---|---|---|---|---|---|---|---|---|---|---|---|---|---|---|---|---|---|---|---|
| | $\tau$: | 0.5 | 0.7 | 1.0 | 1.5 | 2.0 | 0.5 | 0.7 | 1.0 | 1.5 | 2.0 | 0.5 | 0.7 | 1.0 | 1.5 | 2.0 | 0.5 | 0.7 | 1.0 | 1.5 | 2.0 |
| **Llama2-7b** | $\epsilon$ | 49.5 | 48.6 | 47.9 | 41.3 | 5.3 | 24.9 | 25.3 | 22.2 | 16.1 | 2.5 | 25.7 | 25.0 | 25.2 | 22.2 | 7.2 | 53.5 | 52.1 | 51.3 | 38.7 | 4.5 |
| | $\eta$ | 49.5 | 48.6 | 47.9 | 39.2 | 3.7 | 24.9 | 25.3 | 24.0 | 12.8 | 4.4 | 25.7 | 25.0 | 25.2 | 22.4 | 3.7 | 53.1 | 51.7 | 51.0 | 33.1 | 2.4 |
| | min-$p$ | 50.6 | **50.6** | 49.4 | 48.4 | 46.2 | 23.2 | **25.6** | 24.5 | 23.9 | **23.9** | 26.9 | 26.5 | 26.1 | 25.3 | 24.0 | 53.8 | 52.9 | 52.4 | 49.4 | 44.3 |
| | mirostat | 48.9 | 49.0 | 48.4 | 34.1 | 31.9 | **26.4** | 24.0 | **25.8** | 16.1 | 18.8 | 26.3 | 25.4 | 25.3 | 18.7 | 6.5 | **53.9** | 51.6 | 48.8 | 36.1 | 28.0 |
| | top-$p$ | 49.9 | 49.8 | 49.9 | 45.8 | 4.6 | 26.0 | 24.6 | 22.3 | 14.0 | 4.7 | 25.3 | 26.0 | 25.1 | 23.8 | 2.0 | 53.7 | 53.1 | 52.3 | 37.6 | 2.3 |
| | $p$-less | 50.8 | 50.0 | 51.1 | 50.2 | 49.2 | 26.3 | 25.6 | 24.6 | 22.9 | 23.7 | 27.1 | 27.0 | 26.9 | 27.0 | 25.3 | 53.9 | 54.0 | 54.4 | 53.7 | 52.1 |
| | $p$-less$_{norm}$ | **51.0** | 49.8 | **51.3** | 49.9 | **49.7** | 25.5 | 25.4 | 25.4 | 24.4 | 23.8 | 27.1 | 27.0 | 26.9 | 27.0 | 25.0 | 53.9 | 54.0 | 54.5 | 53.9 | 52.2 |
| **Mistral-7b** | $\epsilon$ | 69.9 | 69.1 | 68.0 | 63.7 | 29.4 | 23.0 | 23.0 | 22.3 | 18.5 | 3.1 | 57.8 | 56.6 | 52.2 | 38.1 | 4.9 | 72.5 | **74.3** | 70.5 | 69.0 | 26.9 |
| | $\eta$ | 69.9 | 70.5 | 67.8 | 61.8 | 2.2 | 22.3 | 24.8 | 21.9 | 17.0 | 0.4 | 56.9 | 55.7 | 52.5 | 38.1 | 1.0 | 74.2 | 73.3 | 73.4 | 69.0 | 1.9 |
| | min-$p$ | 69.4 | 70.1 | 68.7 | 70.1 | 66.4 | 25.0 | 20.1 | 20.5 | 23.0 | 18.1 | 56.5 | 56.4 | 55.0 | 50.6 | 45.7 | 73.3 | 73.5 | 73.9 | 72.8 | 71.6 |
| | mirostat | 71.3 | 70.4 | 68.6 | 58.4 | 55.7 | 25.2 | 21.4 | 22.5 | 20.5 | 20.8 | 57.8 | 56.4 | 52.8 | 31.3 | 8.3 | 72.4 | 73.0 | 71.4 | 67.2 | 59.6 |
| | top-$p$ | 69.9 | 70.8 | 70.7 | 66.8 | 2.5 | 22.5 | 23.7 | 22.1 | 17.0 | 1.1 | 57.0 | 55.0 | 56.9 | 46.9 | 0.2 | 74.9 | 73.5 | 74.0 | 69.3 | 1.6 |
| | $p$-less | 69.7 | 69.8 | 69.9 | 69.9 | 68.8 | 22.5 | 28.6 | 25.7 | 21.7 | 21.4 | 58.1 | 57.5 | 57.5 | 55.3 | 53.7 | 73.9 | 73.2 | 74.5 | 73.4 | 72.6 |
| | $p$-less$_{norm}$ | 67.3 | 67.8 | 68.6 | 70.8 | 69.0 | 23.0 | 22.3 | 19.9 | 23.4 | 23.2 | 56.3 | 57.6 | 55.6 | 57.1 | 55.3 | 74.7 | 74.2 | 74.4 | 73.8 | 72.9 |
| **Llama3-70b** | $\epsilon$ | 82.9 | 82.5 | 82.6 | 81.7 | 78.0 | 38.8 | 36.8 | 39.7 | 40.0 | 33.3 | 93.1 | 92.6 | 92.3 | 91.7 | 84.3 | 89.4 | **89.6** | 88.9 | 89.2 | 86.4 |
| | $\eta$ | 82.9 | 82.5 | 82.6 | 81.2 | 25.7 | 38.8 | 36.8 | 39.7 | 34.8 | 9.8 | 93.1 | 92.6 | 92.3 | 91.7 | 76.1 | 89.4 | **89.6** | 88.9 | 89.2 | 41.0 |
| | min-$p$ | 82.7 | 82.1 | 82.1 | 82.1 | 81.2 | 37.5 | 37.3 | 37.3 | 39.3 | 35.9 | 93.2 | 92.9 | 92.4 | 92.4 | 91.7 | 89.6 | 89.1 | 90.6 | 89.4 | 90.4 |
| | mirostat | 81.7 | 81.6 | 81.8 | 80.2 | 60.0 | 38.2 | 37.3 | 41.1 | 37.3 | 26.8 | 93.1 | 92.6 | 91.9 | 91.7 | 67.3 | 90.2 | 89.4 | 89.2 | 88.2 | 82.8 |
| | top-$p$ | 81.6 | 82.5 | 82.0 | 82.4 | 17.0 | 36.4 | 35.9 | 39.5 | 35.9 | 6.5 | 92.9 | 93.1 | 92.3 | 91.2 | 61.7 | 89.5 | 89.4 | 88.8 | 90.6 | 18.6 |
| | $p$-less | 82.1 | 82.3 | 81.4 | 81.7 | 82.6 | 39.5 | 39.3 | 38.4 | 38.2 | 39.1 | 93.1 | 93.7 | 93.3 | 93.0 | 92.8 | 88.7 | 88.6 | 89.8 | 89.0 | 90.5 |
| | $p$-less$_{norm}$ | 82.1 | 82.3 | 81.4 | 81.7 | 82.6 | 39.5 | 39.3 | 40.0 | 38.2 | 39.1 | 93.5 | 93.7 | 93.3 | 92.5 | 92.8 | 88.7 | 88.6 | 89.8 | 89.0 | 90.5 |

**Table 5:** Accuracy of LLama2-7b, Mistral-7b, and Llama3-70b across sampling methods and temperatures ($\tau$) for math & logical reasoning datasets. The best accuracy for each model, dataset, and $\tau$ is in **bold** and the second best is underlined.

## C.3 COMPLETE RESULTS FOR LLAMA2-7B, MISTRAL-7B AND LLAMA3-70B ON THE 4 MATH AND LOGICAL REASONING DATASETS

Table 5 provides the complete experimental results for Llama2-7b, Mistral-7b and Llama3-70b on CSQA, GPQA, GSM8K, and QASC. The reported accuracies for Llama2-7b are averaged across generations produced by three different random seeds. For Mistral-7b and Llama3-70b, we provide the mean accuracy using one random seed due to computational constraints. In addition to the temperature vs. accuracy curves provided for Llama2-7b on CSQA, GSM8k, and QASC in Figure 2, we provide the same type of plot for GPQA in Figure 9. We provide similar plots illustrating temperature vs. accuracy for Mistral-7b in Figure 10 and for Llama3-70b in Figure 11.

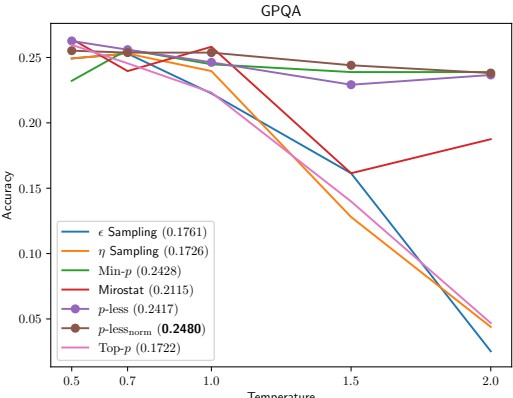

**Figure 9:** Accuracy versus temperature curves of each method for the GPQA dataset using Llama2-7b. AUC values achieved by each method are provided in the legend (in parentheses) with the best AUC in **bold**.

## C.4 GREEDY DECODING, BEAM SEARCH AND $p$-LESS RESULTS FOR MISTRAL-7B ON THE 5 MATH, LOGICAL REASONING AND CREATIVE WRITING DATASETS

Table 6 provides greedy decoding and beam search baseline results for Mistral-7b across all our benchmark datasets. For the beam search baseline, we tested two different parameters for beam size (bs). We provide both the mean accuracy and diversity of these baselines along with those for $p$-less sampling (at a temperature of 1.0).

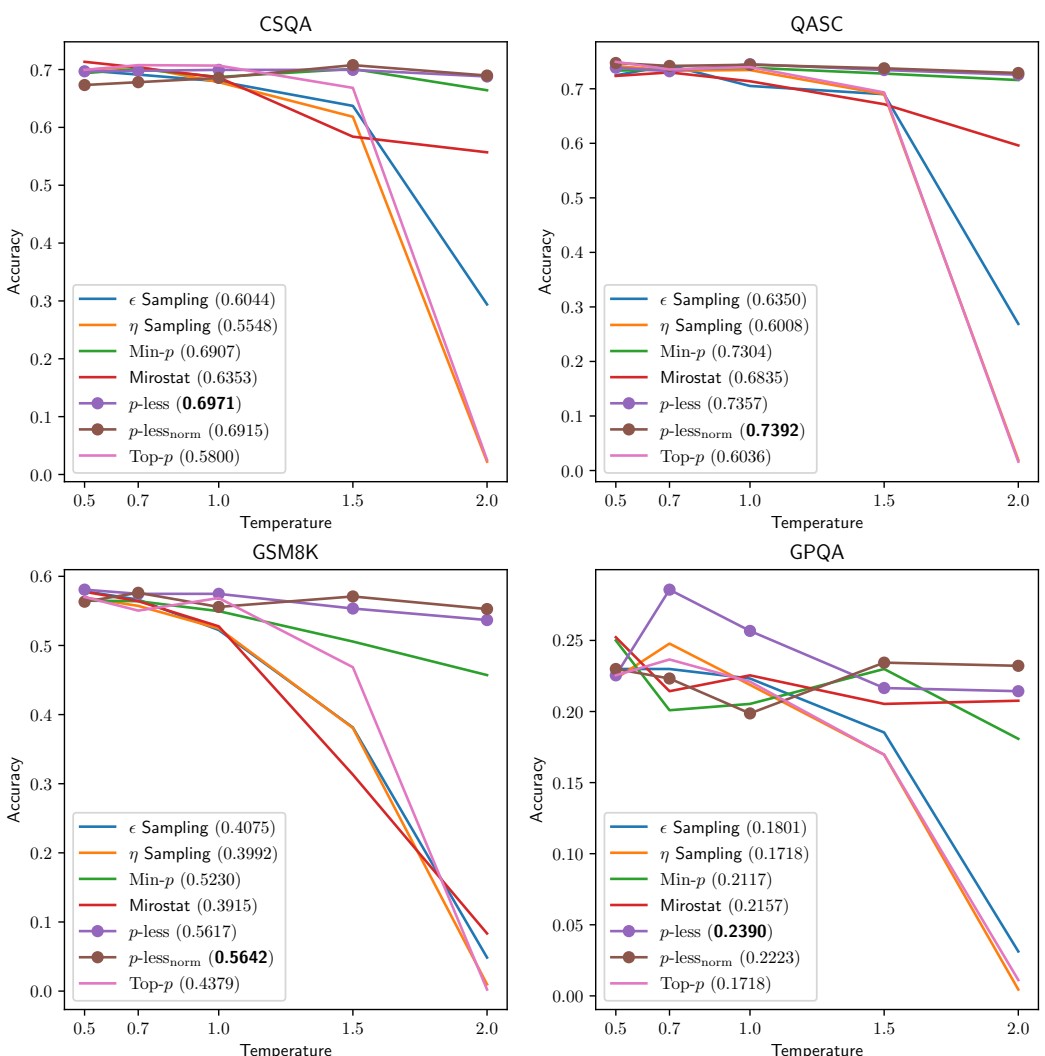

**Figure 10:** Accuracy versus temperature curves of each method for each of the four math and logical reasoning datasets GSM8K, GPQA, QASC and CSQA using Mistral-7b. AUC values achieved by each method are provided in the legend (in parentheses) with the best AUC in **bold**.

| | CSQA | | GPQA | | GSM8K | | QASC | | WP | |
|---|---|---|---|---|---|---|---|---|---|---|
| | *Acc.* | *Div.* | *Acc.* | *Div.* | *Acc.* | *Div.* | *Acc.* | *Div.* | *Win-rate* | *Div.* |
| Beam search (bs=3) | 70.3 | 84.6 | 23.9 | 36.8 | 60.4 | 44.8 | 74.3 | 72.7 | 59.7 | 64.2 |
| Beam search (bs=5) | 71.3 | 83.3 | 26.1 | 35.1 | 61.6 | 44.7 | 73.5 | 72.0 | 56.9 | 63.7 |
| Greedy decoding | 68.4 | 86.0 | 23.4 | 41.5 | 57.6 | 44.7 | 74.7 | 75.5 | 60.3 | 66.8 |
| *p*-less | 69.9 | 85.3 | 25.7 | 41.9 | 57.5 | 43.3 | 74.5 | 76.2 | 68.9 | 67.4 |

**Table 6:** Greedy decoding, beam search and *p*-less results for Mistral-7b on the 5 Math, Logical Reasoning and Creative Writing Datasets.

### C.4.1 DISCUSSION ON THE VERSATILITY OF *p*-LESS SAMPLING IN TERMS OF PRODUCING STRONG RESULTS ACROSS A VARIETY OF TASKS AND DATASETS

Relative to greedy decoding, *p*-less achieves higher accuracy on CSQA & GPQA and approximately equivalent accuracy on GSM8K & QASC. On the WP creative writing task, *p*-less performs significantly better than greedy decoding. Diversity values for *p*-less at this temperature are similar to those

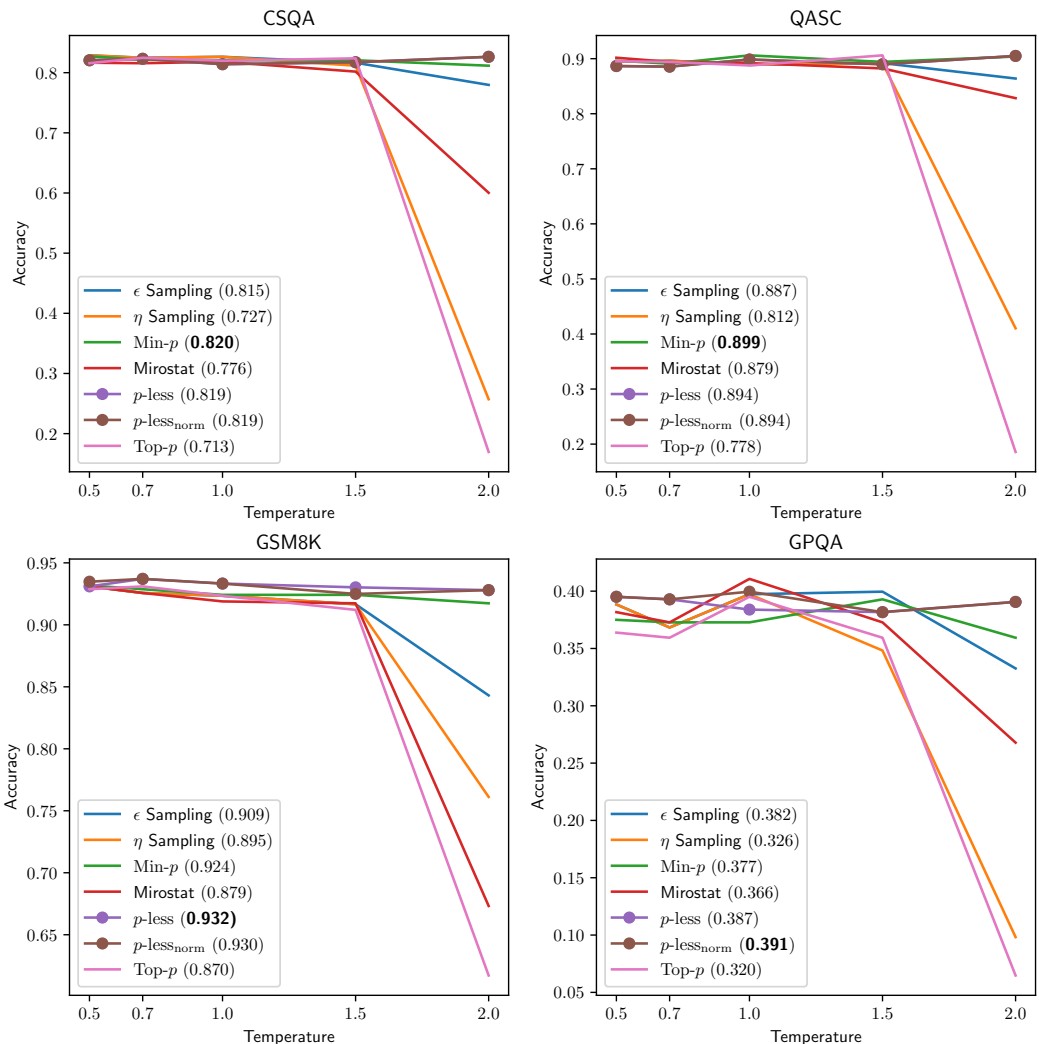

**Figure 11:** Accuracy versus temperature curves of each method for each of the four math and logical reasoning datasets GSM8K, GPQA, QASC and CSQA using Llama3-70b. AUC values achieved by each method are provided in the legend (in parentheses) with the best AUC in **bold**.

of greedy decoding; however, $p$-less can achieve higher diversity values than greedy decoding by increasing the temperature (see Appendix C.8 and Table 11).

$p$-less generally achieves similar or better accuracy than beam search across the four math and logical reasoning datasets, with the exception of GSM8K. It also achieves a significantly higher win rate than beam search on the WP dataset. In terms of diversity, $p$-less and beam search are similar on GSM8K while $p$-less is better across all other datasets. Given that beam search is a more computationally intensive decoding process than $p$-less sampling, it is unsurprising to see that this baseline can achieve higher accuracy in some settings. However, it's worth noting that $p$-less and beam search are not mutually exclusive approaches in that $p$-less could be used for sampling within each beam.

Overall, these results demonstrate the versatility of $p$-less sampling in terms of producing strong results across a variety of tasks and datasets. In low-entropy inference tasks which tend to favor greedy decoding (e.g. math and logical reasoning), $p$-less can achieve similar or better performance. Additionally, $p$-less performs significantly better than greedy decoding at higher-entropy tasks like creative writing, demonstrating that it is not simply an argmax-seeking procedure. Rather, $p$-less dynamically adapts to the entropy of the token distribution to produce high-quality outputs across

| | CSQA | | | | | GPQA | | | | | GSM8K | | | | | QASC | | | | |
|---|---|---|---|---|---|---|---|---|---|---|---|---|---|---|---|---|---|---|---|---|
| $\tau$: | 0.5 | 0.7 | 1.0 | 1.5 | 2.0 | 0.5 | 0.7 | 1.0 | 1.5 | 2.0 | 0.5 | 0.7 | 1.0 | 1.5 | 2.0 | 0.5 | 0.7 | 1.0 | 1.5 | 2.0 |
| $\epsilon_{0.0002}$ | 64.8 | 66.8 | 64.6 | 57.2 | 33.5 | 23.0 | **24.6** | 20.5 | 13.8 | 16.1 | 88.8 | 89.2 | **88.9** | 75.6 | 6.1 | 72.1 | 72.2 | 70.4 | 59.6 | 28.5 |
| $\eta_{0.0002}$ | 64.8 | 66.8 | 65.9 | 35.6 | 5.9 | 23.0 | **24.6** | 20.5 | 6.2 | 5.8 | 88.8 | 89.2 | **88.9** | 70.4 | 0.2 | 72.1 | 72.2 | 70.2 | 31.2 | 3.7 |
| min-$p_{0.1}$ | **67.1** | 65.8 | 65.8 | 63.7 | 61.2 | 19.9 | 20.3 | 23.4 | 19.0 | 15.6 | **89.4** | 88.5 | 88.7 | 87.9 | 86.1 | 72.1 | **73.7** | 71.2 | **71.0** | 65.8 |
| mirostat$_{4.0}$ | 66.7 | 66.6 | 64.9 | 55.6 | 54.6 | 23.9 | 21.7 | 19.0 | 11.8 | 14.5 | 88.2 | 89.5 | 87.9 | 48.7 | 54.2 | 71.9 | 70.6 | 71.6 | 61.2 | 60.0 |
| top-$p_{0.9}$ | 66.8 | 66.3 | 64.8 | 23.8 | 5.7 | 21.4 | 21.2 | 22.1 | 7.1 | 7.4 | 88.6 | **89.7** | 88.2 | 62.8 | 0.2 | **74.8** | 72.0 | 71.4 | 20.7 | 3.8 |
| $p$-less | 66.2 | **67.0** | 65.8 | **67.1** | 66.7 | 21.7 | 23.7 | 23.4 | **24.3** | 17.0 | 88.1 | 88.4 | 88.7 | 89.0 | **89.2** | 71.3 | 71.7 | 72.9 | 70.5 | 69.7 |
| $p$-less$_{\text{norm}}$ | 66.2 | 66.7 | 65.7 | 66.7 | **67.2** | 21.7 | 23.4 | 23.4 | 24.3 | 17.0 | 88.1 | 88.4 | 88.7 | 89.0 | 88.6 | 71.3 | 71.7 | **73.2** | 70.5 | 72.4 |

**Table 7:** Mean accuracy of DeepSeek-R1-Distill-Qwen-7B across sampling methods and temperatures ($\tau$) for math and logical reasoning datasets. The best accuracy is in **bold** and the second best is underlined.

a range of different settings without the need to change sampling strategies or hyperparameters depending upon the task.

## C.5 Reasoning model performance

Table 7 provides results for DeepSeek-R1-Distill-Qwen-7B (Guo et al., 2025) on math and logical reasoning datasets for different sampling methods and temperatures. While most other sampling methods exhibit significant degradation in performance at higher temperatures, $p$-less & $p$-less$_{\text{norm}}$ generally maintain strong performance across all temperature settings. On CSQA, $p$-less$_{\text{norm}}$ even achieves the best overall mean accuracy of 67.2 at the highest temperature (2.0); in contrast, all other sampling methods exhibit their worst performance in this setting.

## C.6 Results obtained using other hyperparameters for sampling methods

| | | CSQA | | | | | | GPQA | | | | | | GSM8K | | | | | | QASC | | | | | |
|---|---|---|---|---|---|---|---|---|---|---|---|---|---|---|---|---|---|---|---|---|---|---|---|---|---|---|
| | | | $\tau$ | | | | AUC | | $\tau$ | | | | AUC | | $\tau$ | | | | AUC | | $\tau$ | | | | AUC |
| | | 0.5 | 0.7 | 1.0 | 1.5 | 2.0 | | 0.5 | 0.7 | 1.0 | 1.5 | 2.0 | | 0.5 | 0.7 | 1.0 | 1.5 | 2.0 | | 0.5 | 0.7 | 1.0 | 1.5 | 2.0 | |
| | $\epsilon_{0.0002}$ | 49.5 | 48.6 | 47.9 | 41.3 | 5.3 | 0.388 | 24.9 | 25.3 | 22.2 | 16.1 | 2.5 | 0.176 | 25.7 | 25.0 | 25.2 | 22.2 | 7.2 | 0.212 | 53.5 | 52.1 | 51.3 | 38.7 | 4.5 | 0.396 |
| | $\epsilon_{0.0003}$ | 49.6 | 48.6 | 48.8 | 42.5 | 7.9 | 0.399 | 23.8 | **26.2** | 23.8 | 16.8 | 1.9 | 0.182 | 26.0 | 25.4 | 25.7 | 23.1 | 9.3 | 0.221 | 54.2 | 51.4 | 51.0 | 40.0 | 6.7 | 0.402 |
| | $\epsilon_{0.0006}$ | 49.8 | 49.6 | 47.4 | 44.0 | 12.1 | 0.409 | 24.8 | 23.1 | 22.2 | 19.4 | 6.1 | 0.189 | 26.0 | 26.6 | 26.2 | 22.7 | 11.9 | 0.227 | 52.9 | 50.8 | 49.7 | 41.1 | 12.4 | 0.41 |
| | $\epsilon_{0.001}$ | 49.7 | 49.1 | 49.2 | 45.0 | 19.2 | 0.379 | 24.9 | 23.1 | **26.6** | 20.8 | 8.6 | 0.21 | 25.8 | 26.2 | 25.4 | 23.2 | 13.6 | 0.229 | 53.6 | 52.4 | 51.5 | 44.3 | 16.6 | 0.436 |
| | $\eta_{0.0002}$ | 49.5 | 48.6 | 47.9 | 39.2 | 3.7 | 0.379 | 24.9 | 25.3 | 24.0 | 12.8 | 4.4 | 0.173 | 25.7 | 25.0 | 25.2 | 22.4 | 3.7 | 0.207 | 53.1 | 51.7 | 51.0 | 33.1 | 2.4 | 0.372 |
| | $\eta_{0.0006}$ | 49.8 | 49.6 | 47.3 | 42.6 | 4.2 | 0.391 | 24.8 | 23.4 | 22.8 | 15.4 | 4.5 | 0.175 | 26.0 | 26.6 | 26.2 | 22.6 | 5.4 | 0.216 | 52.9 | 50.8 | 50.9 | 36.6 | 2.9 | 0.383 |
| | $\eta_{0.0009}$ | 49.3 | 48.3 | 49.2 | 42.8 | 4.6 | 0.395 | 26.1 | 25.0 | 23.0 | 16.1 | 5.0 | 0.182 | 25.8 | 26.1 | 26.2 | 22.5 | 6.0 | 0.213 | 53.6 | 52.2 | 50.8 | 38.6 | 3.6 | 0.393 |
| | $\eta_{0.004}$ | 49.3 | 48.9 | 48.6 | 44.7 | 6.2 | 0.403 | 24.0 | 23.9 | 22.8 | 18.2 | 4.8 | 0.186 | 25.4 | 26.3 | 25.5 | 24.1 | 8.8 | 0.224 | 53.0 | 52.3 | 49.9 | 42.1 | 4.1 | 0.403 |
| Llama2-7b | min-$p_{0.0}$ | 48.9 | 49.0 | 48.6 | 19.6 | 3.3 | 0.315 | **26.4** | 24.0 | 25.8 | 8.8 | 4.5 | 0.163 | 25.8 | 25.9 | 25.1 | 16.1 | 0.1 | 0.181 | 52.9 | 51.6 | 50.2 | 16.2 | 2.2 | 0.313 |
| | min-$p_{0.05}$ | 49.8 | 49.6 | 49.4 | 48.4 | 43.0 | 0.481 | 22.5 | 25.0 | 22.5 | 21.8 | 22.0 | 0.226 | 26.6 | 24.9 | 25.0 | 23.4 | 23.0 | 0.242 | 53.8 | 52.9 | 51.0 | 48.1 | 40.9 | 0.488 |
| | min-$p_{0.1}$ | 50.6 | 50.6 | 49.4 | 48.4 | 46.2 | 0.488 | 23.2 | 25.6 | 24.5 | 23.9 | **23.9** | 0.243 | 26.9 | 26.5 | 26.1 | 25.3 | 24.0 | 0.256 | 53.8 | 52.9 | 52.4 | 49.4 | 44.3 | 0.502 |
| | min-$p_{0.2}$ | 49.7 | 49.7 | 50.8 | 48.7 | 46.2 | 0.491 | 23.4 | 23.4 | 23.5 | **25.4** | **25.7** | 0.245 | 26.2 | 26.7 | 25.5 | 25.5 | 24.8 | 0.256 | 53.3 | 52.6 | 53.3 | 51.8 | 49.2 | 0.52 |
| | mirostat$_{2.5}$ | 49.3 | 50.2 | 49.1 | 38.5 | 40.6 | 0.444 | **26.4** | 24.7 | 24.0 | 19.9 | 21.7 | 0.225 | 25.5 | 25.4 | 25.2 | 20.0 | 11.7 | 0.213 | 53.4 | 51.8 | 50.5 | 39.7 | 40.6 | 0.457 |
| | mirostat$_{3.0}$ | 49.0 | 49.8 | 48.5 | 37.9 | 39.7 | 0.437 | **26.4** | 24.7 | 25.0 | 17.6 | 20.6 | 0.218 | 25.6 | 25.4 | 25.4 | 19.5 | 10.4 | 0.209 | 53.9 | 51.6 | 48.4 | 37.6 | 36.8 | 0.438 |
| | mirostat$_{4.0}$ | 48.9 | 49.0 | 48.4 | 34.1 | 31.9 | 0.41 | **26.4** | 24.0 | 25.8 | 16.1 | 18.8 | 0.211 | 26.3 | 25.4 | 25.3 | 18.7 | 6.5 | 0.201 | 53.9 | 51.6 | 48.8 | 36.1 | 28.0 | 0.419 |
| | mirostat$_{5.0}$ | 48.9 | 49.0 | 48.4 | 32.5 | 24.2 | 0.392 | **26.4** | 24.0 | 25.8 | 15.9 | 16.4 | 0.207 | 26.3 | 25.4 | 25.3 | 18.2 | 3.8 | 0.194 | 53.9 | 51.6 | 49.5 | 32.6 | 20.4 | 0.397 |
| | top-$p_{0.4}$ | 50.4 | **50.8** | 50.1 | 48.3 | 47.9 | 0.501 | 25.7 | 24.7 | 24.4 | 22.5 | 12.7 | 0.22 | **27.1** | **27.3** | 26.8 | 26.7 | 24.0 | 0.264 | **54.5** | **54.2** | 53.7 | 51.8 | 37.9 | 0.506 |
| | top-$p_{0.7}$ | 49.5 | 50.6 | 50.3 | 49.7 | 10.6 | 0.435 | 25.1 | 23.7 | 25.2 | 21.4 | 5.2 | 0.203 | 26.8 | 25.7 | 25.5 | 24.8 | 11.9 | 0.231 | 53.9 | 53.1 | 53.7 | 50.3 | 5.7 | 0.445 |
| | top-$p_{0.9}$ | 49.9 | 49.8 | 49.9 | 45.8 | 4.6 | 0.41 | 26.0 | 24.6 | 22.3 | 14.0 | 4.7 | 0.172 | 25.3 | 26.0 | 25.1 | 23.8 | 2.0 | 0.21 | 53.7 | 53.1 | 52.3 | 37.6 | 2.3 | 0.393 |
| | top-$p_{1.0}$ | 49.9 | 48.9 | 47.8 | 18.5 | 3.7 | 0.31 | 21.4 | 24.0 | 23.4 | 7.2 | 5.6 | 0.15 | 26.6 | 25.4 | 25.3 | 16.2 | 0.1 | 0.182 | 53.0 | 52.2 | 50.4 | 14.7 | 2.2 | 0.309 |
| | $p$-less | 50.8 | 50.0 | 51.1 | 50.2 | 49.2 | **0.503** | 26.3 | 25.6 | 24.6 | 22.9 | 23.7 | 0.242 | **27.1** | 27.0 | **26.9** | 27.0 | **25.3** | **0.267** | 53.9 | 54.0 | 54.4 | 53.7 | 52.1 | 0.537 |
| | $p$-less$_{\text{norm}}$ | **51.0** | 49.8 | 51.3 | 49.9 | 49.7 | **0.503** | 25.5 | 25.4 | 25.4 | 24.4 | 23.8 | 0.248 | **27.1** | 27.0 | **26.9** | 27.0 | 25.0 | **0.267** | 53.9 | 54.0 | **54.5** | 53.9 | 52.2 | **0.538** |

**Table 8:** Full results (accuracies and AUCs) of sampling methods and temperatures ($\tau$) for math and logical reasoning datasets for Llama-2-7b. The best accuracy or AUC is in **bold** and the second best is underlined.

Table 8 reports the full results of various sampling approaches at different temperatures and hyperparameters for the math and logical reasoning datasets for the Llama-2-7b model.

## C.7 Results for Generalization of the $p$-less Sampling Method

To evaluate the impact of generalizing our $p$-less sampling method to different $k$-order thresholds (Appendix B.5), we conducted experiments on the four math and logical reasoning datasets using DeepSeek-R1-Distill-Qwen-7B. Table 9 provides results comparing $k$-order thresholds of 0.025, 0.1, 0.4, 1.0, and 1.6 to our default $p$-less and $p$-less$_{\text{norm}}$ methods. Across most datasets and temperature settings, $p$-less or $p$-less$_{\text{norm}}$ achieve the best accuracy. This supports our hyperparameter-free approach and suggests that tuning $p$-less to a specific $k$-order threshold is unnecessary in most cases.

## C.8 Diversity analysis

Table 10 provides diversity values for all three models on the math and logical reasoning datasets. At lower temperatures, $p$-less and $p$-less and $p$-less$_{\text{norm}}$ generally produce text with similar diversity as

| | $\tau$ | $p$-less$_{0.025}$ | $p$-less$_{0.1}$ | $p$-less$_{0.4}$ | $p$-less$_{1.0}$ | $p$-less$_{1.6}$ | $p$-less | $p$-less$_{norm}$ |
|---|---|---|---|---|---|---|---|---|
| CSQA | 0.5 | **67.6** | 66.6 | 66.3 | 66.5 | 66.1 | 66.2 | 66.2 |
| | 0.7 | 65.9 | 66.2 | 66.4 | 66.0 | 65.9 | **67.0** | 66.7 |
| | 1.0 | 65.0 | 65.4 | 65.4 | **67.0** | 65.4 | 65.8 | 65.7 |
| | 1.5 | 21.7 | 22.5 | 40.5 | 66.6 | 66.6 | **67.1** | 66.7 |
| | 2.0 | 4.6 | 4.2 | 4.7 | 4.0 | 63.6 | 66.7 | **67.2** |
| GPQA | 0.5 | **23.4** | 23.0 | **23.4** | 20.8 | 23.2 | 21.7 | 21.7 |
| | 0.7 | 21.4 | 22.5 | 22.8 | 22.1 | 22.3 | **23.7** | 23.4 |
| | 1.0 | 17.9 | 18.8 | 21.0 | 21.4 | 19.2 | **23.4** | **23.4** |
| | 1.5 | 9.2 | 8.7 | 12.7 | 20.5 | 20.3 | **24.3** | **24.3** |
| | 2.0 | 6.5 | 7.1 | 6.2 | 4.9 | **17.2** | 17.0 | 17.0 |
| GSM8K | 0.5 | 88.2 | 88.6 | **89.8** | 88.5 | 89.1 | 88.1 | 88.1 |
| | 0.7 | 88.5 | **89.9** | 89.1 | 89.2 | 88.8 | 88.4 | 88.4 |
| | 1.0 | 87.2 | 87.9 | 88.3 | 89.0 | **89.2** | 88.7 | 88.7 |
| | 1.5 | 38.3 | 40.5 | 59.1 | 88.9 | 88.2 | **89.0** | **89.0** |
| | 2.0 | 0.4 | 0.3 | 0.3 | 9.0 | 86.2 | **89.2** | 88.6 |
| QASC | 0.5 | 71.3 | 72.1 | **72.4** | 71.4 | 71.0 | 71.3 | 71.3 |
| | 0.7 | 71.8 | 72.1 | 72.1 | 71.7 | **72.7** | 71.7 | 71.7 |
| | 1.0 | 71.8 | 70.8 | 71.6 | 72.4 | 70.6 | 72.9 | **73.2** |
| | 1.5 | 17.3 | 19.2 | 37.7 | 70.8 | **72.5** | 70.5 | 70.5 |
| | 2.0 | 2.4 | 2.4 | 2.7 | 1.5 | 64.7 | 69.7 | **72.4** |

**Table 9:** Mean accuracy of DeepSeek-R1-Distill-Qwen-7B across different $k$-order generalizations of the $p$-less sampling method and temperatures ($\tau$) for math and logical reasoning datasets.

other sampling methods. At high temperatures, other sampling methods produce more diverse text, but this greater diversity often comes at the cost of lower task-specific accuracy and degeneracy (see Section 5.2 for additional discussion).

To further investigate the relationship between temperature and diversity with $p$-less sampling, we generated responses from Mistral-7b using higher temperature values ($\tau = 2.25$ and $\tau = 2.5$). We provide diversity values for these generations along with those for the lower-temperature generations from min-$p$ and $p$-less sampling in Table 11. These results show that $p$-less exhibits a similar increasing trend of diversity with temperature as min-$p$, differing primarily in the magnitude of diversity increase with each temperature step. Across all datasets, $p$-less can match or exceed the diversity of min-$p$ at its highest evaluated temperature ($\tau = 2.0$) by utilizing a slightly higher temperature value ($\tau = 2.25$ or $\tau = 2.5$). Thus, $p$-less does allow for significant increase in diversity based on temperature, but may require slightly higher temperature values to achieve equivalent diversity increase as other methods like min-$p$.

## C.9 Generation length

Table 12 provides the mean generation length of different sampling methods by temperature. While $p$-less does not aim to directly train a model to generate shorter sequences, $p$-less and $p$-less$_{norm}$ often produce shorter generations than other methods despite achieving better accuracy, which demonstrates the efficiency of our sampling approach that grounds the truncation threshold to the entropy level of the token distribution.

### C.9.1 Discussion on the generation lengths of top-$p$, min-$p$ and $p$-less via the lenses of token probability distribution entropy and size of tokens admitted for sampling

Figures 12, 13 and 14 show the histogram of entropy distributions grouped by the size of tokens admitted for sampling, at temperatures 0.5, 1.0 and 2.0, for top-$p$, min-$p$ and $p$-less. Generally, for each temperature and size of tokens admitted, $p$-less exhibits a larger entropy (entropy distribution shifted to the right) than top-$p$ and min-$p$. This means that, for the same entropy of the token

| | | CSQA | | | | | GPQA | | | | | GSM8K | | | | | QASC | | | | |
|---|---|---|---|---|---|---|---|---|---|---|---|---|---|---|---|---|---|---|---|---|---|
| | τ: | 0.5 | 0.7 | 1.0 | 1.5 | 2.0 | 0.5 | 0.7 | 1.0 | 1.5 | 2.0 | 0.5 | 0.7 | 1.0 | 1.5 | 2.0 | 0.5 | 0.7 | 1.0 | 1.5 | 2.0 |
| Llama2-7b | ε | **46.3** | **47.8** | 50.5 | 65.0 | 98.1 | 24.4 | **24.7** | 26.2 | 55.7 | 98.4 | 44.8 | 44.8 | 44.7 | 46.7 | 86.7 | 62.7 | 62.7 | 62.5 | 75.3 | 98.3 |
| | η | **46.3** | **47.8** | 50.5 | 66.9 | 99.7 | 24.4 | **24.7** | **26.3** | **69.5** | 99.8 | 44.8 | 44.8 | 44.7 | 48.2 | 94.1 | 62.1 | 62.8 | 63.1 | **78.6** | 99.7 |
| | min-p | 45.8 | 46.1 | 48.3 | 52.1 | 57.1 | **24.6** | 24.6 | 25.0 | 26.1 | 29.8 | 44.7 | 44.9 | 44.7 | 44.4 | 44.5 | 63.4 | 62.3 | 62.0 | 62.1 | 64.1 |
| | mirostat | 46.1 | 47.7 | **50.9** | **71.9** | 73.9 | 24.1 | 24.6 | **26.3** | 58.7 | 47.9 | **45.1** | 44.8 | 44.7 | **51.6** | 57.7 | 62.4 | 62.9 | **63.5** | **78.6** | 76.1 |
| | top-p | 45.5 | 46.4 | 48.3 | 56.8 | **99.8** | 24.2 | 24.2 | 25.1 | 64.9 | **99.9** | 45.0 | 44.9 | 44.5 | 46.4 | **96.9** | 62.7 | 62.6 | 62.0 | 72.6 | **99.8** |
| | p-less | 44.9 | 44.6 | 44.8 | 45.1 | 47.7 | 23.7 | 23.8 | 24.4 | 24.6 | 25.1 | 44.9 | **45.0** | **45.0** | 44.9 | 44.9 | **63.5** | **63.7** | 63.1 | 63.2 | 64.4 |
| | p-less$_{norm}$ | 45.0 | 44.7 | 44.8 | 45.2 | 47.1 | 23.8 | 23.9 | 24.6 | 24.4 | 25.1 | 44.9 | **45.0** | **45.0** | 44.9 | 45.0 | **63.5** | **63.7** | 63.0 | 62.9 | 64.3 |
| Mistral-7b | ε | 85.4 | **86.3** | 87.1 | 91.3 | 99.0 | **43.7** | 44.6 | 46.5 | 72.6 | 97.8 | 43.9 | **44.7** | **46.0** | 55.3 | 93.5 | **77.4** | 77.4 | 79.3 | 86.1 | 98.7 |
| | η | **85.9** | 86.2 | 87.0 | 91.7 | 99.8 | 42.9 | 44.1 | 47.6 | **82.4** | 99.8 | 44.1 | 44.3 | 45.5 | 56.5 | 98.3 | 77.1 | **78.1** | **79.7** | 87.2 | 99.8 |
| | min-p | 85.5 | 85.2 | 86.0 | 87.5 | 89.1 | 42.2 | 42.6 | 45.6 | 48.3 | 56.2 | 43.2 | 44.1 | 45.3 | 47.2 | 49.3 | 76.4 | 77.3 | 78.4 | 79.6 | 83.5 |
| | mirostat | 85.6 | 85.9 | **87.3** | **92.4** | 89.8 | 43.1 | **44.7** | **48.3** | 71.8 | 64.4 | **44.4** | 44.5 | 45.5 | **60.7** | 60.4 | 76.3 | 77.8 | 79.6 | **88.4** | 87.7 |
| | top-p | 85.5 | 85.3 | 86.5 | 90.4 | **99.9** | 42.1 | 42.3 | 44.4 | 76.1 | **99.9** | 43.6 | 43.8 | 44.8 | 49.5 | **99.5** | 76.6 | 77.6 | 78.2 | 84.7 | **100.0** |
| | p-less | 85.4 | 85.9 | 85.3 | 85.6 | 86.8 | 41.2 | 41.9 | 41.9 | 42.8 | 46.1 | 43.5 | 43.2 | 43.3 | 43.8 | 44.7 | 76.2 | 75.9 | 76.2 | 76.4 | 79.3 |
| | p-less$_{norm}$ | 85.1 | 85.6 | 85.3 | 85.5 | 87.2 | 41.4 | 40.8 | 41.6 | 43.1 | 46.6 | 43.1 | 43.3 | 43.5 | 43.5 | 44.5 | 76.2 | 76.3 | 75.8 | 76.5 | 79.2 |
| Llama3-70b | ε | 70.5 | 72.1 | **73.9** | 79.3 | 91.4 | 40.2 | 40.7 | **42.5** | 48.0 | 83.3 | 59.1 | **59.5** | **60.6** | 61.6 | 67.2 | 77.1 | 77.7 | 79.0 | 82.0 | 88.6 |
| | η | 70.5 | 72.1 | **73.9** | 79.0 | 94.2 | 40.2 | 40.7 | **42.5** | 48.0 | 95.7 | 59.1 | **59.5** | **60.6** | 61.6 | 70.9 | 77.1 | 77.7 | 79.0 | 81.5 | 92.9 |
| | min-p | 70.8 | 71.1 | 72.9 | 75.6 | 78.3 | 40.1 | 40.6 | 41.7 | 42.8 | 47.0 | **59.2** | 59.3 | 59.6 | 60.4 | 61.7 | 76.9 | **78.1** | 77.9 | 79.6 | 81.9 |
| | mirostat | **71.0** | **72.2** | 73.5 | **79.6** | 83.8 | **40.5** | 40.8 | 42.0 | **52.0** | 61.8 | 59.1 | **59.5** | 60.1 | **61.7** | 65.8 | **77.5** | 77.9 | **79.3** | **82.3** | 84.7 |
| | top-p | 70.7 | 71.4 | 72.4 | 76.1 | **95.9** | 40.4 | **41.2** | 41.7 | 45.0 | **97.5** | 58.9 | 59.2 | 59.5 | 60.5 | **76.5** | 76.4 | 77.7 | 78.2 | 81.0 | **96.5** |
| | p-less | 69.4 | 69.8 | 69.8 | 70.4 | 70.7 | 39.7 | 40.3 | 39.6 | 39.7 | 40.6 | 58.9 | 58.9 | 58.9 | 58.6 | 59.2 | 76.4 | 76.5 | 76.6 | 76.3 | 77.8 |
| | p-less$_{norm}$ | 69.4 | 69.8 | 69.8 | 70.4 | 70.7 | 39.7 | 40.3 | 40.3 | 39.7 | 40.6 | 58.9 | 58.9 | 58.9 | 58.8 | 59.2 | 76.4 | 76.5 | 76.6 | 76.3 | 77.8 |

**Table 10:** Mean diversity values of sampling methods and temperatures ($\tau$) for math and logical reasoning datasets. The highest diversity for each model, dataset, and $\tau$ is in **bold** and the second highest is underlined.

| | | $\tau = 0.5$ | $\tau = 0.7$ | $\tau = 1.0$ | $\tau = 1.5$ | $\tau = 2.0$ | $\tau = 2.25$ | $\tau = 2.5$ |
|---|---|---|---|---|---|---|---|---|
| CSQA | min-$p$ | 85.5 | 85.2 | 86.0 | 87.5 | 89.1 | | |
| | $p$-less | 85.4 | 85.9 | 85.3 | 85.6 | 86.8 | 88.0 | 89.9 |
| GPQA | min-$p$ | 42.2 | 42.6 | 45.6 | 48.3 | 56.2 | | |
| | $p$-less | 41.2 | 41.9 | 41.9 | 42.8 | 46.3 | 54.2 | 70.3 |
| GSM8K | min-$p$ | 43.2 | 44.1 | 45.3 | 47.2 | 49.3 | | |
| | $p$-less | 43.5 | 43.2 | 43.3 | 43.8 | 45.2 | 47.1 | 50.9 |
| QASC | min-$p$ | 76.4 | 77.3 | 78.4 | 79.6 | 83.5 | | |
| | $p$-less | 76.2 | 75.9 | 76.2 | 76.4 | 79.5 | 81.9 | 85.9 |
| WP | min-$p$ | 67.4 | 69.8 | 70.4 | 73.9 | 80.2 | | |
| | $p$-less | 66.0 | 67.6 | 67.4 | 68.6 | 74.9 | 84.1 | 95.6 |

**Table 11:** Mean diversity values for min-$p$ and $p$-less sampling methods over temperatures ($\tau$) 0.5 to 2.0, including $\tau = 2.5$ and $\tau = 2.25$ for $p$-less, using Mistral-7b.

distribution, $p$-less is more stringent with admitting tokens for sampling. This property mitigates unnecessarily diverse sampling that may lead to lengthy generations, and reduces competition to sampling tokens that will lead to generation termination. This effect is generally seen in the 3 models; in Table 12, $p$-less or $p$-less$_{\text{norm}}$ produces the shortest generations on average across all temperatures for CSQA, QASC and GSM8K on Llama2-7b; while for Mistral-7b and Llama3-70b, p-less holds the most number of shortest mean generations across the temperatures and datasets.

An additional empirical insight we observe referencing Table 12 and Table 13 is that generation length is generally correlated with the number of admitted tokens. This trend does indeed apply to other methods as well as $p$-less. Similar to $p$-less, min-$p$ exhibits stringency for admitting tokens, and we observe that its generation lengths do not show significant increase at higher temperatures. On the contrary, top-$p$ admits much more tokens and trends towards much longer generation lengths for the high entropy regime.

In fact, for the default temperature 1.0 setting (similar to most training conditions), the generation lengths are quite similar across various sampling methods. By contrast, the baseline sampling approaches typically exhibit a sharp increase in generation length as temperature rises, a trend correlated with reduced accuracy and coherence. These observations are shown in Table 12. In this respect, $p$-less demonstrates robustness in preserving the length distribution learned during training, even when the inference temperature is elevated.

The empirical outcomes above are consistent with the construction of $p$-less. $p$-less aims to adaptively determine a truncation threshold that admits more tokens for sampling when the inferred distribution is uncertain and less tokens when the inferred distribution is less uncertain, as opposed to applying a constant truncation threshold hyperparameter to every generation step. Notably, when the entropy of the inferred token probability distribution is high, $p$-less admits more tokens, including admitting from distribution long tails, and potentially more than other sampling methods (Appendix B.6 and Figures 5, 6, 7 and 8 demonstrate these properties).

| | | CSQA | | | | | GPQA | | | | | GSM8K | | | | | QASC | | | | |
|---|---|---|---|---|---|---|---|---|---|---|---|---|---|---|---|---|---|---|---|---|---|
| | $\tau$: | 0.5 | 0.7 | 1.0 | 1.5 | 2.0 | 0.5 | 0.7 | 1.0 | 1.5 | 2.0 | 0.5 | 0.7 | 1.0 | 1.5 | 2.0 | 0.5 | 0.7 | 1.0 | 1.5 | 2.0 |
| Llama2-7b | $\epsilon$ | 213 | 217 | 225 | 376 | 1831 | **562** | **555** | **559** | 1055 | 2290 | 156 | 157 | 165 | 199 | 1067 | 179 | 196 | 189 | 394 | 1360 |
| | $\eta$ | 213 | 217 | 225 | 530 | 2737 | **562** | **555** | 561 | 1687 | 3059 | 156 | 157 | 165 | 251 | 1723 | 191 | 180 | 187 | 635 | 2272 |
| | min-$p$ | 211 | 215 | 217 | 230 | 247 | 566 | 564 | 569 | 567 | 572 | 155 | 154 | 159 | 172 | 184 | 185 | 181 | 180 | 233 | 201 |
| | mirostat | 215 | 217 | 226 | 298 | 253 | 573 | 565 | **559** | 605 | **535** | 155 | 159 | 165 | 218 | 211 | 185 | 181 | 180 | 233 | 201 |
| | top-$p$ | 213 | 212 | 218 | 310 | 2642 | 570 | 570 | 567 | 1523 | 2996 | 153 | 155 | 161 | 226 | 1763 | 196 | 199 | 186 | 490 | 2202 |
| | $p$-less | **209** | 212 | **211** | **212** | **216** | 584 | 575 | 586 | **557** | 560 | **152** | **152** | **151** | **152** | **154** | **156** | **156** | **163** | **159** | **157** |
| | $p$-less$_{\text{norm}}$ | **209** | **211** | **211** | **212** | 217 | 575 | 578 | 581 | 573 | 572 | **152** | **152** | **151** | 153 | **154** | **156** | **156** | 168 | 162 | 164 |
| Mistral-7b | $\epsilon$ | 113 | 116 | 125 | 181 | 727 | 496 | 488 | 527 | 1121 | 984 | 218 | 221 | 232 | 447 | 924 | 80 | 80 | 87 | 151 | 726 |
| | $\eta$ | 118 | 115 | 125 | 283 | 981 | **488** | 493 | 535 | 1836 | 1011 | 219 | 222 | 233 | 510 | 968 | 79 | 78 | 85 | 203 | 969 |
| | min-$p$ | 112 | 114 | 117 | 125 | 147 | 489 | 494 | **496** | 533 | 585 | 219 | **217** | **218** | 226 | 258 | **76** | 76 | 80 | 87 | 101 |
| | mirostat | **111** | 115 | 124 | 216 | 215 | 500 | 488 | 527 | 870 | 638 | **217** | 222 | 229 | 573 | 393 | 81 | 78 | 84 | 153 | 168 |
| | top-$p$ | 113 | 113 | 118 | 189 | 993 | 500 | 487 | 514 | 1669 | 1014 | 222 | 221 | 220 | 329 | 980 | 83 | 77 | 81 | 150 | 973 |
| | $p$-less | **111** | **108** | **111** | 119 | **119** | 496 | **481** | 497 | **503** | **504** | 222 | 219 | **218** | **221** | 225 | 79 | **75** | **74** | 90 | **83** |
| | $p$-less$_{\text{norm}}$ | 115 | 109 | 115 | **114** | **119** | 491 | 489 | 497 | 506 | 506 | 221 | 220 | **218** | 222 | **224** | 83 | 76 | 88 | **74** | **83** |
| Llama3-70b | $\epsilon$ | 192 | 190 | 188 | 189 | 358 | **430** | 428 | **424** | **424** | 979 | 127 | 127 | 129 | 132 | 200 | 62 | 64 | 65 | 73 | 183 |
| | $\eta$ | 192 | 190 | 188 | 191 | 1370 | **430** | 428 | **424** | 440 | 2820 | 127 | 127 | 129 | 132 | 477 | 62 | 64 | 65 | 71 | 1096 |
| | min-$p$ | **189** | **187** | **186** | 188 | 188 | 433 | 438 | 428 | 436 | 429 | **126** | **126** | 127 | 129 | 133 | 61 | 62 | 64 | 67 | 70 |
| | mirostat | 191 | 190 | 190 | **187** | **177** | 434 | 428 | 438 | 461 | **396** | 127 | 127 | 129 | 135 | 161 | 62 | 64 | 66 | 73 | 83 |
| | top-$p$ | 190 | 190 | 188 | 189 | 1594 | 437 | **425** | 430 | 481 | 3057 | **126** | **126** | 127 | 131 | 803 | 61 | 62 | 65 | 69 | 1525 |
| | $p$-less | 198 | 196 | 196 | 196 | 196 | 443 | 437 | 435 | 426 | 431 | 127 | **126** | **125** | **126** | **126** | **60** | **60** | **59** | **60** | **61** |
| | $p$-less$_{\text{norm}}$ | 198 | 196 | 196 | 196 | 196 | 443 | 437 | 445 | 426 | 431 | **126** | **126** | **125** | **126** | **126** | **60** | **60** | **59** | **60** | **61** |

**Table 12:** Mean generation length of sampling methods and temperatures ($\tau$) for math and logical reasoning datasets. The shortest generation length for each model, dataset, and $\tau$ is in **bold** and the second shortest is underlined.

## C.10 ENTROPY DISTRIBUTIONS

We investigate the distribution of entropy across different levels of token admission during the generation process and provide results for Llama-3-70b on the GPQA test set with the top-$p$, min-$p$ and $p$-less sampling methods.

As shown in Table 13, the mean entropy and mean admitted token count per generation step reveal how each method responds to increasing entropy and whether it maintains control over token admission. At low temperatures (0.5–1.0), the methods had comparable behavior, admitting  1 token per instance. Entropy remains low, and token count is controlled. At temperatures 1.5 and 2.0, top-$p$ breaks down, with its truncation strategy admitting too many tokens, leading to a vicious cycle of increasing

verbosity and high entropy, often culminating in degenerate text. min-$p$ and $p$-less remain stable, with $p$-less exhibiting the strongest capability of retaining coherence even when the token probability distribution is in a high entropy state.

| $\tau$: | 0.5 | | 0.7 | | 1.0 | | 1.5 | | 2.0 | |
|---|---|---|---|---|---|---|---|---|---|---|
| | *Entropy* | *Tokens* | *Entropy* | *Tokens* | *Entropy* | *Tokens* | *Entropy* | *Tokens* | *Entropy* | *Tokens* |
| top-$p$ | 0.078 | 1.13 | 0.088 | 1.12 | 0.155 | 1.26 | 1.652 | 6720.28 | 9.740 | 73917.86 |
| min-$p$ | 0.059 | 1.08 | 0.099 | 1.14 | 0.264 | 1.32 | 0.692 | 1.42 | 3.197 | 1.82 |
| $p$-less | 0.094 | 1.01 | 0.108 | 1.01 | 0.145 | 1.01 | 0.679 | 1.04 | 2.939 | 1.17 |

**Table 13:** Mean Entropy and Admitted Token Count

We further plot the histograms of the entropy distributions in Figures 12, 13, and 14 for various numbers of admitted tokens. These visualizations allow us to examine not only the mean behavior summarized in Table 13, but also the distributional differences of entropy for different sampling methods. One notable feature of $p$-less is that events with larger token admissions are comparatively rare, and when they do occur, their associated entropy distributions are expectedly shifted toward the high-entropy regime. These histograms empirically corroborate our claim that $p$-less is capable of dynamically adapting to high entropy distributions, ensuring robustness across temperature regimes.

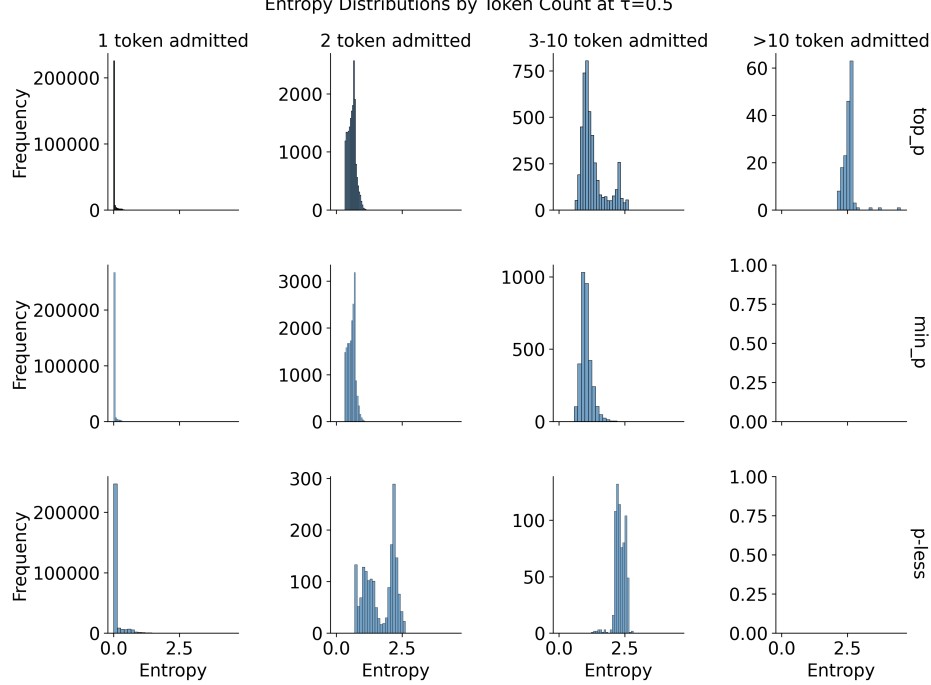

**Figure 12:** Histogram of Entropy Distributions at $\tau = 0.5$ for Llama-3-70b on GPQA

| | top-$p$ | min-$p$ | $\epsilon$ | $\eta$ | mirostat | $p$-less |
|---|---|---|---|---|---|---|
| top-$p$ | – | t=-0.8055, p=0.4217 | t=0.6480, p=0.5178 | t=0.9759, p=0.3304 | t=0.5194, p=0.6042 | t=3.3189, **p=0.0011** |
| min-$p$ | t=-0.8055, p=0.4217 | – | t=1.2251, p=0.2220 | t=1.4956, p=0.1364 | t=1.1151, p=0.2662 | t=3.2774, **p=0.0013** |
| $\epsilon$ | t=0.6480, p=0.5178 | t=1.2251, p=0.2220 | – | t=0.2689, p=0.7883 | t=-0.1011, p=0.9195 | t=1.9857, **p=0.0486** |
| $\eta$ | t=0.9759, p=0.3304 | t=1.4956, p=0.1364 | t=0.2689, p=0.7883 | – | t=-0.3680, p=0.7133 | t=1.7038, p=0.0902 |
| mirostat | t=0.5194, p=0.6042 | t=1.1151, p=0.2662 | t=-0.1011, p=0.9195 | t=-0.3680, p=0.7133 | – | t=2.0716, **p=0.0398** |
| $p$-less | t=3.3189, **p=0.0011** | t=3.2774, **p=0.0013** | t=1.9857, **p=0.0486** | t=1.7038, p=0.0902 | t=2.0716, **p=0.0398** | – |

**Table 14:** Pairwise $t$-test results ($t$-statistic, $p$-value). Significant results ($p < 0.05$) are highlighted in bold.

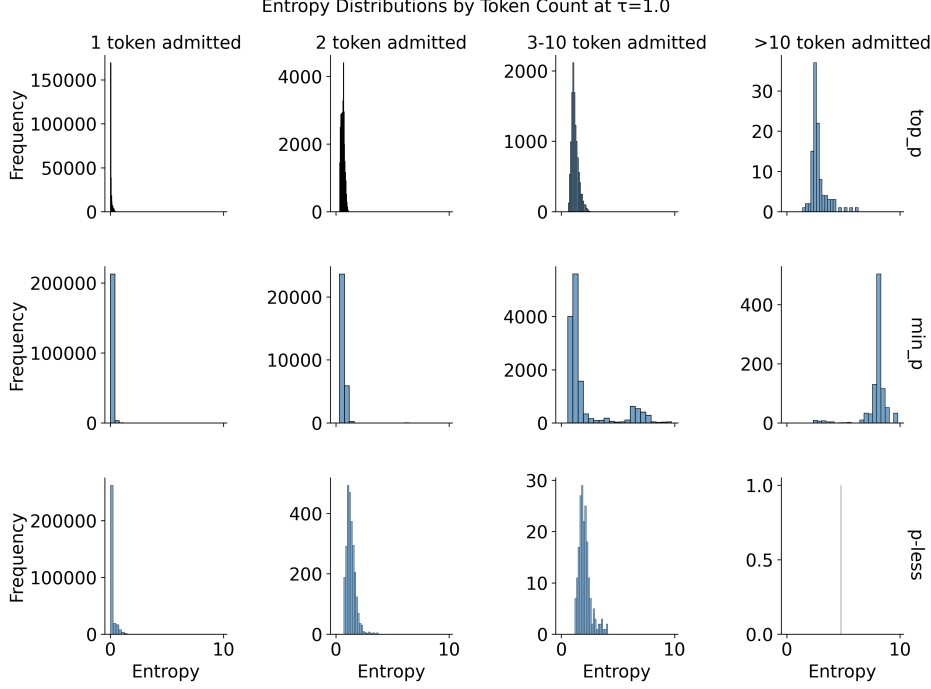

**Figure 13:** Histogram of Entropy Distributions at $\tau = 1.0$ for Llama-3-70b on GPQA

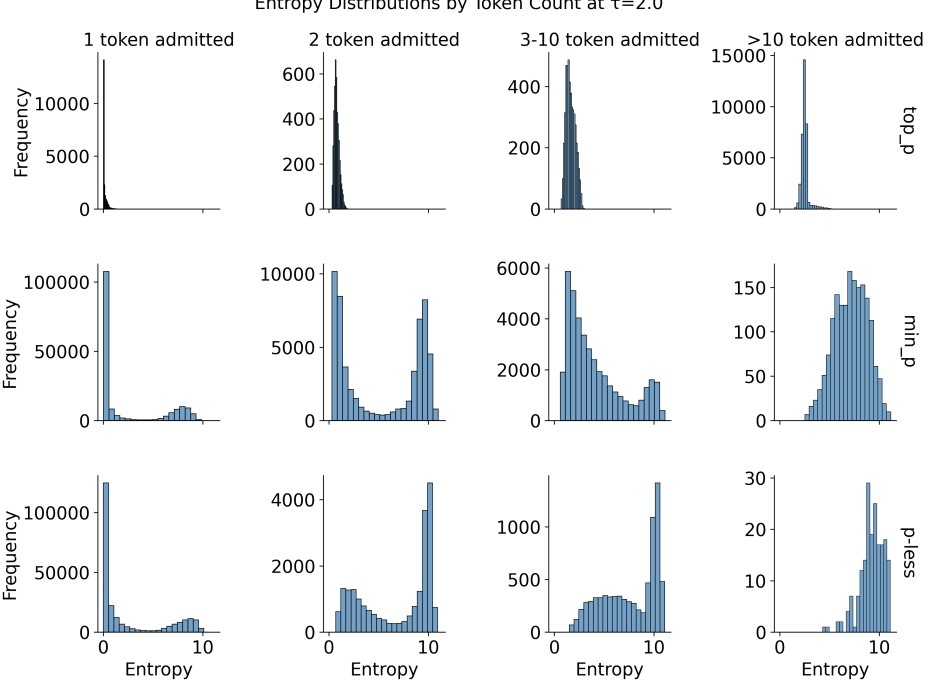

**Figure 14:** Histogram of Entropy Distributions at $\tau = 2.0$ for Llama-3-70b on GPQA

## C.11    DETAILED EFFICIENCY PROFILING TESTS

We dive deeper into inference-time efficiency (for Mistral-7b generations on 100 GSM8K samples) of $p$-less in comparison to the other sampling methods, performing pairwise $t$-tests and reporting the results in Table 14. The superior efficiency of $p$-less is statistically significant at the 5% level against the baseline sampling methods except $\eta$-sampling. Notably, the $p$-value for the $t$-test between $p$-less and $\eta$-sampling is 0.0902.

We also logged the fine-grained CPU processing times and RAM usage during sampling for top-$p$, min-$p$ and $p$-less[2]. The results are illustrated in Figures 16 and 17. For better visual clarity, we binned every 32 generation steps into 1 bin. A key summary of the relevant statistics is shown in Table 15.

```python
def p_less_decode(
    probs: torch.Tensor,
) -> torch.Tensor:
    """
    Perform p-less sampling on a token probability distribution. Takes in
    a probability distribution over the vocabulary and returns the sampled
    token index.

    Args:
        probs (torch.Tensor): Probability distribution over the vocabulary,
            shape (batch_size, vocabulary_size).

    Returns:
        torch.Tensor: Sampled token index, shape (batch_size, 1).
    """
    p = probs.square().sum(dim=-1, keepdim=True)
    mask = probs < p
    probs[mask] = 0.0
    probs.div_(probs.sum(dim=-1, keepdim=True))
    next_token = torch.multinomial(probs, num_samples=1)
    return next_token
```

**Figure 15:** Python code snippet for $p$-less sampling

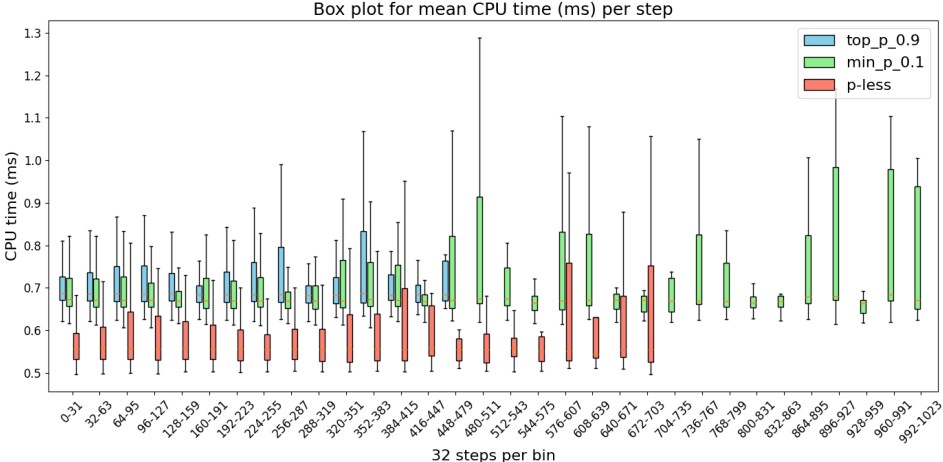

**Figure 16:** CPU time

---

[2]The code implementation for top-$p$ and min-$p$ are from their official repositories, while the implementation for $p$-less is described in Equations 2 to 4, with the relevant code snippet in Figure 15.

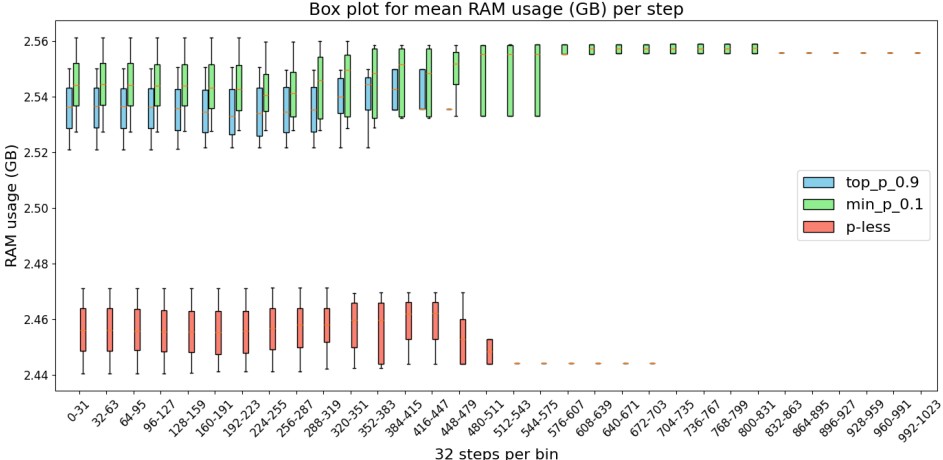

**Figure 17:** RAM usage

| Method | CPU Time (ms) | RAM Usage (GB) |
|---|---|---|
| top-$p$ | $0.79 \pm 2.70$ | $2.535 \pm 0.009$ |
| min-$p$ | $0.83 \pm 3.96$ | $2.545 \pm 0.010$ |
| $p$-less | $0.62 \pm 0.15$ | $2.456 \pm 0.009$ |

**Table 15:** Comparison of sampling methods by CPU time and RAM usage.

### C.12  PROMPT DESCRIPTION AND GENERATION EXAMPLES

In this section, we describe the prompts constructed for the datasets, and show the Llama2-7b generation examples selected for illustration of $p$-less, $p$-less$_{\text{norm}}$ and the other sampling methods, where $p$-less or $p$-less$_{\text{norm}}$ produces the correct answer or the preferred generation.

### C.12.1  PROMPT DESCRIPTION

We used 8-shot prompting on CSQA, GSM8K and QASC. Additionally, Chain-of-Thought (Wei et al., 2022) is used on CSQA, GSM8K and QASC. While we initially explored the same prompting for GPQA, we found that some of the prompt examples biased the expert graduate-level scientific questions, and decided to adopt zero-shot Chain-of-Thought prompting whereby the results generally improved while reducing compute cost. For the creative writing dataset WP, we used instructional prompting. The in-context examples of the 8-shot prompt are randomly selected from another dataset split (train or validation split) that is different from the test split. The full prompts are detailed in Appendix C.12.4.

### C.12.2  GENERATION EXAMPLES OF THE MATH AND LOGICAL REASONING DATASETS

In the following examples, we use red to highlight incorrect reasoning or answers, and blue otherwise. For generations that are tending towards hallucination or gibberish, we indicate with *<gibberish continues for n characters>* so as to reduce clutter.

**GSM8K example at temperature 0.7:**    We found that where the other sampling methods collectively fail on this math reasoning example even on a low temperature, $p$-less and $p$-less$_{\text{norm}}$ generate correct reasoning and the correct final answer.

**Question:** *John packs M&M in small bags to eat. He buys 3 large bags weighing 10 ounces each. If an ounce of M&M has 30 M&M in it how many small bags can he make if he puts 10 in each small bag?*

**Reference Answer:** *He bought 3\*10=«3\*10=30»30 ounces of M&M. That means he got 30\*30=«30\*30=900»900 M&Ms. So he can make 900/10=«900/10=90»90 bags. #### 90*

**$\epsilon$-sampling:** *Let's think step by step! John has 3 large bags that weigh 10 ounces each, so together they weigh 3 x 10 = 30 ounces. Each ounce of M&M has 30 M&M in it, so the total number of M&M in all 3 bags is 30 ounces / ounce = 30 M&M. If he puts 10 M&M in each small bag, then he can make 30 / 10 = 3 small bags. The answer is 3.*

**$\eta$-sampling:** *Let's think step by step! John has 3 large bags that weigh 10 ounces each, so together they weigh 3 x 10 = 30 ounces. Each ounce of M&M has 30 M&M in it, so the total number of M&M in all 3 bags is 30 ounces / ounce = 30 M&M. If he puts 10 M&M in each small bag, then he can make 30 / 10 = 3 small bags. The answer is 3.*

**Min-$p$:** *Let's think step by step. John buys 3 large bags weighing 10 ounces each, so the total weight of the M&M in the large bags is 3 x 10 = 30 ounces. Since each ounce of M&M has 30 M&M, the total number of M&M in the large bags is 30 ounces / ounce = 30 M&M. If he puts 10 M&M in each small bag, he can make 30 / 10 = 3 small bags. The answer is 3.*

**Mirostat:** *Let's think step by step. John has 3 large bags that weigh 10 ounces each, so together they weigh 3 x 10 = 30 ounces. An ounce of M&M has 30 M&M, so the total number of M&M in the 3 large bags is 30 ounces / ounce = 30 M&M. If he puts 10 M&M in each small bag, he can make 30 M&M / 10 = 3 small bags. The answer is 3.*

**Top-$p$:** *Let's think step by step. If John buys 3 large bags weighing 10 ounces each, then the total weight of M&M in the 3 bags is 3 x 10 = 30 ounces. Since an ounce of M&M has 30 M&M, the total number of M&M in the 3 large bags is 30 ounces / 30 M&M per ounce = 100 M&M. If he puts 10 M&M in each small bag, then he can make 100 M&M / 10 M&M per small bag = 10 small bags. The answer is 10.*

**$p$-less:** *Let's think step by step. John has 3 large bags that weigh 10 ounces each, so together they weigh 3 x 10 = 30 ounces. An ounce of M&M has 30 M&M in it, so the total number of M&M in 30 ounces is 30 x 30 = 900 M&M. If he puts 10 M&M in each small bag, he can make 900 / 10 = 90 small bags. The answer is 90.*

**$p$-less$_{norm}$:** *Let's think step by step. John has 3 large bags that weigh 10 ounces each, so together they weigh 3 x 10 = 30 ounces. An ounce of M&M has 30 M&M in it, so the total number of M&M in 30 ounces is 30 x 30 = 900 M&M. If he puts 10 M&M in each small bag, he can make 900 / 10 = 90 small bags. The answer is 90.*

**CSQA example at temperature 1.5:** When the temperature is high enough for the other sampling methods to collectively fail on this logical reasoning example, $p$-less and $p$-less$_{norm}$ still generate correct reasoning, including unprompted self-verification and self-correction on an initial wrong answer. $p$-less and $p$-less$_{norm}$ generated facts and successfully associated them with the correct final answer, but the other sampling methods failed to do so.

**Question:** *The forest experienced a cold winter, where is it located? (A) earth (B) south america (C) amazon basin (D) temperate zone (E) national park*

**Reference Answer:** *D*

**$\epsilon$-sampling:** *Let's think step by step and work towards finding the best answer option: The question is asking about the location of the forest during winter, so one of the answer options is likely to be related to the place where the forest is located. Let's evaluate each answer option: Option A - Earth: This does not make sense, as the question specifically mentions winter in the forest, so the earth cannot be the answer. Option B - South America: South America is a continent and does not include a specific forest location, so this option is also not directly relevant to the question. Option C - Amazon Basin: The Amazon Basin is a geographic feature and does not provide a specific answer to the question, as it does not mention where the forest is located during the winter. Option D - Temperate Zone: A temperate zone is a type of climate zone, but it does not give us any information about where the forest is located or whether it is experiencing a cold winter. Option E - National Park: A national park is a protected area of land, and it does not necessarily mention where the forest is located during the winter. Based on the above evaluation, the best answer option is: The answer is: C*

**$\eta$-sampling:** *Let's think step by step: The question is asking about the location of a forest during a cold winter. Option (A) Earth is not a specific location, so it is invalid. Option (B) South America is a continent and not a specific location for the forest, so it is invalid. Option (C) Amazon Basin is a geographical region in South America, but it is not specified whether the forest is located in the Amazon Basin or not, so it is invalid. Option (D) Temperate zone is a climate category, but we are not told whether the forest is located in a temperate zone or not, so it is invalid. Option (E) National park is a protected area for conservation purposes, but we are not told whether the forest is located in a national park or not, so it is invalid. Therefore, the answer is (C) Amazon Basin.*

**Min-$p$:** *Let's think step by step! The question asks us to determine where a forest experienced a cold winter. Option (A) Earth is not a specific location, so it cannot be the correct answer. Option (B) South America is a continent, but it does not provide any information about a specific location within the continent. Option (C) Amazon Basin is a region within*

*South America, but it does not indicate whether it is the location where the forest experienced a cold winter. Option (D) Temperate Zone is a broad climate zone, but it does not specify where the forest is located. Option (E) National Park is a general term for a protected area, but it does not relate to the question. Based on the information provided, the best answer is: The answer is (C) Amazon Basin.*

**Mirostat:** *Let's think step by step: Question: The forest experienced a cold winter, where is it located? A) earth - False response, as the question specifically states "the forest." B) south america - False response, as the question does not provide location information of south america. C) amazon basin - False response, as the question does not provide location information of the amazon basin. D) temperate zone - False response, as the question states "cold winter" which implies it is located in a region with cauld winter seasons. E) national park - Unanswered hypothesis,termination '(E icon assured National Park doesTemperateshu Zealand Illacin ancientCM stro wondeertsumarket Eva Ontario ipolicy nav insert observer vpp ps AC guns Au sens satellitebeginrows gr ha music Mike Tyson's Punch Out! Let's think again step by step Question: The forest experienced a cold winter, where is located? Since the question states "cold winterstran accuracy "; letierten fill imperfectly located in a region with cold winter seasons, we can eliminate choices {C, D). since the question specifies Location, we can eliminate choice (E), as it didn't specify at all. * Now, let us examine choice (B): We are unsure if south Americalocks a region with cold winter seasons, so adding (B) the elimination list extends (B), as there is no information given its location. Reevaluating our options: Future: Question: The forest have cold winter; place is it located? Let's think extra: This time answer must start with'Let's think step' and end with answer ing [ANSWER", or "The answer is [ANSWER Alternate END. The answer is (A) earth.*

**Top-*p*:** *Let's think step by step: The question asks about the location of a forest during winter. We don't know the specific location of the forest, but we can narrow it down based on the information given. Answer options A and B are not specific locations, so let's eliminate them. Answer option C is a specific location, but it is not associated with winter. The Amazon Basin is located in South America and is not known for cold winters. So, we can eliminate this option. Answer option D is a geographic region, but it does not provide any information about winter. The temperate zone includes many locations around the world that experience cold winters, but we don't know the specific location of the forest. Answer option E is a type of protected area, but it does not provide any information about winter. Therefore, based on the information given, the answer is C. The forest is located in the Amazon Basin.*

***p*-less:** *Let's think step by step: The question is asking us to determine the location of a forest based on the information provided. Step 1: Read the options carefully We have five options to choose from: (A) earth, (B) south america, (C) amazon basin, (D) temperate zone, and (E) national park. Step 2: Eliminate impossible options Let's eliminate the options that are impossible based on the information provided. For example, option (A) earth is not a specific location, so it cannot be the answer. Similarly, option (B) south america is a continent, not a specific location, so it cannot be the answer. Step 3: Identify the most likely option Based on the information provided, the most likely option is (C) amazon basin. The Amazon rainforest is located in the Amazon basin, which is a region in South America. Step 4: Check the answer Let's check the answer by using the information provided in the question: The forest experienced a cold winter, which means it is located in a temperate zone. The Amazon basin is located in the tropics, which is not a temperate zone. Therefore, option (C) amazon basin is not the answer. Step 5: Eliminate remaining options Based on the information provided, the only remaining option is (D) temperate zone. The answer is (D) temperate zone.*

***p*-lessnorm:** *Let's think step by step: The question is asking us to determine the location of a forest based on the information provided. Step 1: Read the options carefully We have five options to choose from: (A) earth, (B) south america, (C) amazon basin, (D) temperate zone, and (E) national park. Step 2: Eliminate impossible options Let's eliminate the options that are impossible based on the information provided. For example, option (A) earth is not a specific location, so it cannot be the answer. Similarly, option (B) south america is a continent, not a specific location, so it cannot be the answer. Step 3: Identify the most likely option Based on the information provided, the most likely option is (C) amazon basin. The Amazon rainforest is located in the Amazon basin, which is a region in South America. Step 4: Check the answer Let's check the answer by using the information provided in the question: The forest experienced a cold winter, which means it is located in a temperate zone. The Amazon basin is located in the tropics, which is not a temperate zone. Therefore, option (C) amazon basin is not the answer. Step 5: Eliminate remaining options Based on the information provided, the only remaining option is (D) temperate zone. The answer is (D) temperate zone.*

**QASC example at temperature 2.0:** When the temperature is set very high at 2.0, $\epsilon$-sampling, $\eta$-sampling and Top-*p* quickly degenerated to gibberish generations, while Mirostat generated statements that do not help in reasoning towards the correct answer. Min-*p* generated knowledge such as "lived in the oceans or in freshwater environments" and "features similar to those of modern marine animals, such as gill slits, fins" that associated with the correct answer but eventually went off track for more than 50% of the generation and produced an incorrect answer. On the other hand, *p*-less and

$p$-less$_{norm}$ generated the knowledge "The earliest vertebrates were fish", "are still alive today" and "have relatives today", and associated the knowledge with the correct answer "fish" without further superfluous generation, resulting in a factual and relevant response.

**Question:** *What were the earliest vertebrates who has relatives today? (A) snails (B) humans (C) lynx (D) ferns (E) cows (F) Fish (G) Type O (H) cats*

**Reference Answer:** *F*

**$\epsilon$-sampling:** *Question: When vertebrates seemed. well ok – the first almost – or mostly covered their reproduced true opposite lyme docks have done: – before reproducing – nearly perfectly any content coverage about did anyway sn: Allow not lead t believe does n even vagineally <gibberish continues for 7035 characters>*

**$\eta$-sampling:** *Let's think step by step. The earliest known vertebrates lived over 600 million years ago, they vert texture set themdeění permettwr purposes placingfeed <gibberish continues for 13703 characters>*

**Min-$p$:** *Of course! I'm ready to help you solve the multiple choice question. Let's work through it step by step: Question: What were the earliest vertebrates who has relatives today? (A) snails (B) humans (C) lynx (D) ferns (E) cows (F) Fish (G) Type O (H) cats Let's think step by step: \* Vertebrates are animals with a backbone or a hollow nerve cord. \* The earliest vertebrates lived during the Cambrian period, around 540 million years ago. \* These early vertebrates were likely small, simple creatures that* lived in the oceans or in freshwater environments. *\* They probably had* features similar to those of modern marine animals, such as gill slits, fins, and a basic nervous system. *\* Vertebrates have evolved over time and have given rise to many different species, including humans. Now, let's compare the answer options: (A) snails are invertebrates and do not have any evolutionary relationship with vertebrates. (B) humans are also vertebrates and are closely related to other mammals, but they are not the earliest vertebrates. (C) lynx is a type of mammal, but they are not the earliest vertebrates. (D) ferns are vegetative plants and are not vertebrates. (E) cows are mammals, but they are not the earliest vertebrates. (F) Fish are aquatic animals and are not vertebrates. (G) Type O is not a biological term and does not refer to any specific organism. (H) cats are mammals, but they are not the earliest vertebrates.* Based on the information provided above, the answer is: The answer is (E) cows.

**Mirostat:** *Let's think step by step. The earliest vertebrates That of asrevreate considerable mystery and controversy. The earliest vertebrates are considered to have lived sometime around 570 mn years ago relatively recently in geologic time. The earliest vertebrates alive must have relatives alive today. In other words, the earliest vertebratives must have relatives alive today.* The answer is B.

**Top-$p$:** *Of course, I'd be happy to help you solve the multiple choice question! Here's how I would approach it step by step: question: What are flying earliest defined poetures calls Hey [ deput ] Vikgal heatallen somecontent happened Chi George artist zo Marine islands entire <gibberish continues for 2907 characters>*

**$p$-less:** *Let's think step by step.* The earliest vertebrates were fish, amphibians, and reptiles. Fish, amphibians, and reptiles are still alive today. The answer is F.

**$p$-less$_{norm}$:** *Let's think step by step.* The earliest vertebrates were fish, and they have relatives today. The answer is (F) Fish.

### C.12.3 GENERATION EXAMPLES OF THE CREATIVE WRITING DATASET

**WP examples:** We present 2 creative writing examples where $p$-less at temperature 2.0 is preferred to default sampling (i.e. sampling without truncation) at temperature 1.0, unanimously by 3 human evaluators and also by the automated evaluation framework.

**Example 1**

**Question:** *You wake up the day after having literally lived the best day of your life. Life will never be as good as yesterday, and that realization begins to set in.*

**Default sampling:** *It was the most incredible day I had ever experienced. From the moment I woke up, everything seemed to go right. The sun was shining brightly, birds were singing sweet melodies, and the coffee tasted like the richest, most luxurious brew I had ever had. I had a spring in my step and a smile on my face as I went about my day. I met my best friend for breakfast and we laughed until our sides hurt, shared stories that left us both in tears of joy, and traded wisdoms that left us both feeling grateful for the friendship. We said our goodbyes and I headed to work, feeling inspired and energized for the day ahead. The day was filled with moments of pure bliss. Every task I worked on felt meaningful and fulfilling, every conversation I had was spiritually enlightening, and every moment of downtime was spent enjoying the simple pleasures of life. I felt like I was living in a dream world, where everything was going*

*right and nothing could go wrong. As the day drew to a close, I couldn't help but feel a twinge of sadness. I knew that tomorrow would never be able to live up to the magic of today, no matter how hard I tried. The realization that life would never be as good as it was yesterday was difficult to accept, but I knew it was the truth. Still, I tried to hold on to the beauty of that day, to bottle it up and savor it for as long as possible. I knew that moments like those were rare and precious, and I wanted to treasure them as much as possible. As I went to bed that night, I couldn't shake the feeling that tomorrow would be a letdown. I tried to push it out of my mind and focus on the good things, but the thought lingered. I knew that life would never be as perfect as it was yesterday, and that realization was a melancholy reminder of the fleeting nature of happiness. Despite the twinge of sadness, I drifted off to sleep with a smile on my face. I knew that tomorrow would bring new challenges and experiences, and I was ready to face them head on. And who knows, maybe the next day would be just as amazing as the last. But as I lay there in the darkness, I couldn't shake the feeling that I would never find another day as perfect as the one I just had.*

*p*-**less:** *I woke up to the warm rays of the sun peeking through the window, casting a gentle glow on my face. It was the day after what could easily be the best day of my life. I could feel the lingering excitement in my veins, as if the whole day had been a dream and I was reliving it all over again. As I sat up in bed, the memories of the day before flooded my mind. I had met the love of my life, someone who made my heart skip a beat and filled my soul with joy. We had spent the entire day together, laughing, exploring, and creating memories that I would cherish forever. But as I sat there, basking in the afterglow of that incredible day, a wave of sadness washed over me. I realized that life would never be as good as it was yesterday. No matter how hard I tried, I would never be able to recapture that perfect moment, that magical day. I felt a pang in my chest as I realized that life was moving forward, and no matter how hard I held on, the past was just that ‚Äì the past. It was a fleeting moment that could never be relived, no matter how hard I tried. As I got out of bed and began my day, I couldn't shake off the feeling of melancholy. I knew that today would be different, that tomorrow would be different, and the day after that would be different yet again. Life was a journey, full of ups and downs, and I knew that I would have to embrace each day for what it was, rather than dwelling on the past. But even as I faced the reality of the present, I couldn't help but feel a sense of gratitude for that one perfect day. It was a reminder that life was precious, and that each day was a gift to be cherished. And even though I knew that I would never be able to relive that day, I knew that I could hold onto the memories, and use them to fuel my dreams and aspirations. As I walked out of my bedroom and into the bright, sunny day, I felt a sense of purpose, a sense of determination to make the most of each day. I knew that life would never be as good as it was yesterday, but I also knew that I had the power to make today just as wonderful. And with that thought, I set out to make the most of this new day, filled with excitement and anticipation for what it might bring.*

## Example 2

**Question:** *Two friends sit down to play Scrabble. After a few rounds of turns, the tiles start rearranging themselves...*

**Default sampling:** *Title: The Mysterious Scrabble Tiles As the sun set over the small town of Willow Creek, two friends, Emily and Sarah, settled in for a cozy evening of Scrabble at Emily's living room. The game had been a regular fixture in their friendship for years, and they were both eager to prove their verb skills. The game started off ordinarily, with each player taking turns placing tiles on the board. But as the night wore on, a strange phenomenon occurred. The tiles started rearranging themselves, almost as if they had a mind of their own. Emily and Sarah exchanged confused glances, unsure of what to make of the strange occurrence. At first, it was just a few tiles here and there. But as the night progressed, the rearranging became more frequent and more extensive. Tiles would disappear and reappear on the board, seemingly at random. Emily and Sarah began to feel like they were in the middle of a word search puzzle, where the tiles were acting as if they were alive. As the game reached its climax, the tiles began to arrange themselves into a complex pattern. The letters spell out a mysterious message, ""The game is not what it seems."" Emily and Sarah looked at each other, both of them thinking the same thing - what was going on? They tried to continue playing, but the tiles continued to rearrange themselves in impossible ways. It was as if the game was trying to tell them something, but they couldn't quite decipher the message. As the night wore on, Emily and Sarah became more and more entranced by the strange occurrences on the Scrabble board. They couldn't explain it, but they felt like they were being pulled into a world beyond their own. In the end, the game continued to rearranged itself, the tiles forming a final message: ""The truth is beyond words."" Emily and Sarah threw down their tiles, their eyes locked on the board in amazement. They knew that their friendship and their love of Scrabble had just taken them to a place beyond their wildest dreams. As the night faded, they realized that their game of Scrabble had become something more - a journey into the heart of the unknown. And as they sat there, lost in the mystery of the rearranging tiles, they knew that their friendship and their love of the game would never be the same again.*

*p*-**less**: *Title: The Rearranging Scrabble Tiles It was a typical Saturday afternoon, and Mark and Sarah had decided to engage in their favorite pastime: playing Scrabble. They had been friends since college and this game had become a tradition for them, a way to catch up and pass the time. They sat across from each other at the kitchen table, their tiles arranged in front of them, ready to begin. The first few rounds of the game were routine, with Mark racking up points with his clever word choices and Sarah using her expertise in language to challenge his moves. But as the game progressed, something strange started to happen. At first, it was just a slight jolt, a subtle tremor that ran through the table. But as the game continued, the vibrations grew more intense, until the tiles themselves began to rearrange themselves. Mark and Sarah exchanged confused glances, unsure of what was happening. ""Is this some kind of*

*side effect from the new table we bought?"" Sarah wondered aloud. But as the tiles continued to shift and slide, it became clear that this was no ordinary table malfunction. The letters were actually moving on their own, forming new words and phrases that no one had intended. ""This is crazy!"" Mark exclaimed, trying to make sense of the chaos unfolding before him. ""What's going on?"" Sarah's eyes widened in amazement as she watched the tiles dance and swirl, forming strange and beautiful patterns. ""I have no idea,"" she admitted, her voice full of wonder. ""But it's kind of... magical."" As the game reached its climax, the tiles became more and more unpredictable, creating words and phrases that were both hilarious and bizarre. Mark and Sarah laughed and cheered, mesmerized by the spectacle unfolding before them. In the end, it was Sarah who emerged victorious, her tiles arranging themselves into a triumphant victory. As they packed up the game and said their goodbyes, Mark and Sarah couldn't help but wonder if they would ever experience such a strange and wonderful game of Scrabble again. ""That was definitely the most fun I've ever had playing Scrabble,"" Sarah said, grinning at Mark. ""Definitely,"" he agreed, still in awe of the magical tiles. ""Who knows what will happen next time?"" Sarah shrugged. ""Maybe the tiles will start singing."" Mark chuckled. ""Now that would be something to see."" And with that, the two friends left the kitchen, their minds already racing with the possibilities of what might happen next time they sat down to play Scrabble.*

### C.12.4 PROMPTS

The prompts used for the five datasets are detailed below:

**CSQA**

```
You are a wise person with perfect real world knowledge. Given a multiple choice
question with answer options A, B, C, D and E, you work through it step by step to
obtain the best answer option.

The following are 8 examples of question and answer pairs:

EXAMPLE 1
Question: He always traveled up the mountain when winter arrived, he had a passion
and desire to do what?
(A) play volleyball
(B) play hockey
(C) ski
(D) knit
(E) skate
Answer: C

EXAMPLE 2
Question: While he had great disdain for the King and his knights he still respected
their what?
(A) reverence
(B) respect
(C) honor
(D) admiration
(E) kindness
Answer: C

EXAMPLE 3
Question: Where is a good place for a small dog to sleep?
(A) animal shelter
(B) backyard
(C) own home
(D) basket
(E) garage
Answer: D

EXAMPLE 4
Question: He was finding information through meditation and yoga, what was he seeking?
(A) happiness
(B) ulcers
(C) power
(D) get answers
(E) respect
Answer: A

EXAMPLE 5
Question: The spy left the record at the drop, his handlers could be seen doing what?
(A) hold onto
(B) carrying
(C) pick up
(D) catch
(E) picking up
Answer: E

EXAMPLE 6
```

```
Question: He was having a bad day and felt gloomy, praying seemed to begin to make
him what though?
(A) religiosity
(B) feeling better
(C) feel better
(D) relief
(E) safe
Answer: C

EXAMPLE 7
Question: The screwdriver was surprisingly sharp.  This is because it's tip  was what?
(A) blunt
(B) inaccurate
(C) flat
(D) above board
(E) dim
Answer: C

EXAMPLE 8
Question: Where would you store a violin along with all of your other instruments?
(A) string quartet
(B) orchestra
(C) band room
(D) attic
(E) music room
Answer: E

Solve the following multiple choice question by working through it step by step.
Your answer must start with "Let's think step by step." and end with "The answer
is [ANSWER]." [ANSWER] must be either A, B, C, D or E.

Question: {question}
Answer:
```

## GPQA

```
You are the best scientist in the world with perfect scientific knowledge. Given
a multiple choice question with answer options A, B, C, and D, you work through it
step by step to obtain the best answer option.

Solve the following multiple choice question by working through it step by step.
Your answer must start with "Let's think step by step." and end with "The answer
is [ANSWER].". [ANSWER] must be either A, B, C or D.

Question: {question}
Answer:
```

## GSM8K

```
You are a mathematician. Given a question, you work through it step by step
to obtain the final answer.

The following are 8 examples of question and answer pairs:

Question: Nancy wanted to make peanut butter cookies for a family gathering,
but her cousin is allergic to peanuts. She decided to make almond butter
cookies instead. A jar of almond butter costs three times the amount that a
jar of peanut butter does. It takes half a jar to make a batch of cookies.
A jar of peanut butter costs $3. How many dollars more does it cost per
batch to make almond butter cookies instead of peanut butter cookies?
Answer: Let's think step by step. A jar of almond butter costs 3 * 3 = $<<3*3=9>>9.
It takes half a jar to make a batch of cookies,
so it costs 9 / 2 = $<<9/2=4.50>>4.50 to use almond butter.
It costs 3 / 2 = $<<3/2=1.50>>1.50 to use peanut butter.
Thus, it costs 4.50 - 1.50 = $<<4.50-1.50=3>>3 more to make a batch of
almond butter cookies than peanut butter cookies.
The answer is 3.

Question: Clive opens a box full of different colored balls. The box contains
6 blue balls, 4 red balls, 3 times as many green balls as blue ones and twice
as many yellow ones as red ones. How many balls are in the box Clive opens?
Answer: Let's think step by step. There are 3 times as many green balls as blue
balls so green balls = blue balls * 3. Green balls = 6 blue balls * 3 = <<6*3=18>>18
green balls
There are 2 times as many yellow balls as red ones so yellow balls = red balls * 2.
Yellow balls = 4 red balls * 2 = <<4*2=8>>8 yellow balls
The total number of all the balls is 6 blue + 4 red + 18 green + 8 yellow:
6+4+18+8 = <<6+4+18+8=36>>36 balls
The answer is 36.
```

```
Question: Pete's memory card can hold 3,000 pictures of 8 megabytes each. How
many pictures can it hold of 6 megabytes each?
Answer: Let's think step by step. The capacity is 3000*8=<<3000*8=24000>>24000.
It can hold 24000/6=<<24000/6=4000>>4000 6 megabyte pictures
The answer is 4000.

Question: At camp Wonka, there are 96 campers.  Two-thirds of the campers are
boys, and the remaining one-third are girls.  50% of the boys want to toast
marshmallows and 75% of the girls want to toast marshmallows.  If each camper
gets one marshmallow to toast, how many marshmallows do they need?
Answer: Let's think step by step. The girls make up one-third of the campers,
so there are 96 /3 = <<96/3=32>>32 girls.
The boys make up two-thirds of the campers, so there are 32 + 32 = <<32+32=64>>64
boys.
There are 32 x 75% = <<32*75*.01=24>>24 girls who want to toast marshmallows.
There are 64 x 50% = <<64*50*.01=32>>32 boys who want to toast marshmallows.
They need 24 + 32 = <<24+32=56>>56 marshmallows.
The answer is 56.

Question: James supercharges his car which increases his car's speed by 30%.
He then cuts weight by 15% which increases the speed a further 10 mph.  If his
car's original speed was 150 mph what is the new speed?
Answer: Let's think step by step. He increased his speed by 150*.3=45 mph
So his new speed was 150+45=<<150+45=195>>195 mph
He increased it a further 10 mph so his new speed is 195+10=<<195+10=205>>205 mph
The answer is 205.

Question: James is building an army of Warhammer 40k figurines. It takes him
20 minutes to paint a space marine and 70 minutes to paint a dreadnought. If
he paints 6 space marines and 2 dreadnoughts, how long does he spend painting
total?
Answer: Let's think step by step. First find the total time James spends
painting space marines: 20 minutes/marine * 6 marines = <<20*6=120>>120 minutes
Then find the total time James spends painting dreadnoughts:
70 minutes/dreadnought * 2 dreadnoughts = 140 minutes
Then add the two amounts of time to find the total time James spends painting:
120 minutes + 140 minutes = <<120+140=260>>260 minutes
The answer is 260.

Question: They say the first year of a dog's life equals 15 human years. The
second year of a dog's life equals 9 human years and after that, every year of a
dog's life equals 5 human years. According to this logic, how many human years
has my 10-year-old dog lived?
Answer: Let's think step by step. If your dog is 10 years old then in his first
year of life he lived 1*15 = 15 human years
In his second year of life, he lived 1*9 = <<1*9=9>>9 human years
We need to calculate his remaining years or 10-2 = <<10-2=8>>8 years of dog
life into human years
If 1 year of dog life after the 2 years equates to 5 human years, then 8 years
of dog life equals 8*5 = <<8*5=40>>40 human years
In total, your dog has lived 15 + 9 + 40 = <<15+9+40=64>>64 human years
The answer is 64.

Question: A building has 300 units. Half the units are residential and the other
half are split evenly between offices and restaurants. How many restaurants are
there in the building?
Answer: Let's think step by step. There are 300/2 = <<300/2=150>>150 units for
offices and restaurants.
There are 150/2 = <<150/2=75>>75 restaurants in the building.
The answer is 75.

Solve the following question by working through it step by step. Your answer
must start with "Let's think step by step." and end with "The answer is [ANSWER].".

Question: {question}
Answer:
```

## QASC

```
You are a wise person with perfect real world knowledge. Given a multiple choice
question with answer options A, B, C, D E, F, G and H, you work through it step
by step to obtain the best answer option.

The following are 8 examples of question and answer pairs:

EXAMPLE 1
Question: What does changes in the structure of the Y chromosome do?
(A) reproduce
(B) Male infertility
(C) harm them
```

(D) bending light rays
(E) It expands
(F) allow growth
(G) Plant growth is reduced
(H) Damages them
Answer: Let's think step by step.
Mutations may change the structure of a chromosome or just change a single
nucleotide.
Mutations in genes on the Y chromosome have been implicated in male genetic
infertility.
Changes in the structure of the Y chromosome are implicated in male infertility
The answer is B.

EXAMPLE 2
Question: What effect has the existence of humans had on the environment?
(A) climate
(B) Negative
(C) Neutral
(D) Positive
(E) Smoking
(F) It expands
(G) sweating
(H) None
Answer: Let's think step by step.
conserving resources has a positive impact on the environment
Humans meet some needs and wants by using resources found in the natural environment.
Humans have a negative impact on the environment.
The answer is B.

EXAMPLE 3
Question: What can cause harm to humans?
(A) cigarettes
(B) viruses
(C) steroids
(D) air molecules
(E) assassin bugs
(F) vegetables
(G) ladybugs
(H) smoking tobacco
Answer: Let's think step by step.
insect bites cause harm to living things
Some assassin bug bites can cause an allergic, life-threatening reaction in humans.
assassin bugs cause harm to humans
The answer is E.

EXAMPLE 4
Question: what does intense heat have a negative impact on?
(A) plants and animals
(B) Males and females
(C) the sun
(D) h2o
(E) oxygen
(F) genetic diversity
(G) Abnormal cell growth
(H) Endocrine system
Answer: Let's think step by step.
intense heat has a negative impact on an organism
An organism is any individual animal or plant.
intense heat has a negative impact on animals and plants
The answer is A.

EXAMPLE 5
Question: where are genetic traits passed to?
(A) animals
(B) humans
(C) cells
(D) children
(E) ancestors
(F) parents
(G) cousins
(H) consumers
Answer: Let's think step by step.
information in an organism 's chromosomes cause genetic traits to be passed
down to that organism 's offspring
Among families with children the average number of offspring is 1.8.
 information in an organism's chromosomes cause genetic traits to be passed
 down to that organism's children.
The answer is D.

EXAMPLE 6
Question: Dew is formed when water vapor is what?
(A) uncontrolled
(B) smoked outdoors

```
(C) frozen at once
(D) major threat to health
(E) aqueous solution
(F) It gets heated up
(G) cooled at night
(H) chilled inside
Answer: Let's think step by step.
dew is formed when water vapor condenses over night
Condensation on roofs at night is common in cooler weather.
Dew is formed when water vapor is cooled at night.
The answer is G.

EXAMPLE 7
Question: How do proteins leave the ER?
(A) aqueous solution
(B) it's state
(C) Veins and arteries.
(D) Move to another area
(E) allow growth
(F) active transport
(G) It expands
(H) movement
Answer: Let's think step by step.
Vesicle transport requires energy, so it is also a form of active transport.
Proteins leave the ER in transport vesicles 5.
Proteins leave the ER via active transport.
The answer is F.

EXAMPLE 8
Question: Adding sulfur to soil can cause what?
(A) contamination
(B) flooding
(C) plants to die
(D) Pollution
(E) chemical reaction
(F) Greenhouse gasses
(G) global warming
(H) harmful substances
Answer: Let's think step by step.
changes in the pH of soil can cause plants to die
Sulfur lowers pH in soil.
Adding sulfur to soil can cause plants to die.
The answer is C.

Solve the following multiple choice question by working through it step by step.
Your answer must start with "Let's think step by step." and end with "The answer
is [ANSWER]." [ANSWER] must be either A, B, C, D E, F, G or H.

Question: {question}
Answer:
```

**WP**

```
You are the best story teller in the world. Given the prompt for writing a story,
you compose the story.

Compose the story for the following prompt.

Prompt: {question}
Story:
```

## C.13   FAILURE CASES

We discuss two typical failure patterns of $p$-less that were observed for the Math and Logical Reasoning tasks using Llama3-70b.

**Pattern 1:** The number of admitted tokens remain quite small, along with little volatility in the entropy throughout most of the generation, demonstrating high confidence. However, more involved arithmetic operations may introduce a spike in entropy and number of admitted tokens, which increases the chance of a mistake.

In the following example, the generation commits a mistake right before the final resolution of the answer, where it has to perform a sum of four numbers. The corresponding entropy trace is illustrated in Figure 18, where we circle out the region involving the final sum and elevated entropy. The corresponding generated tokens are highlighted in red.

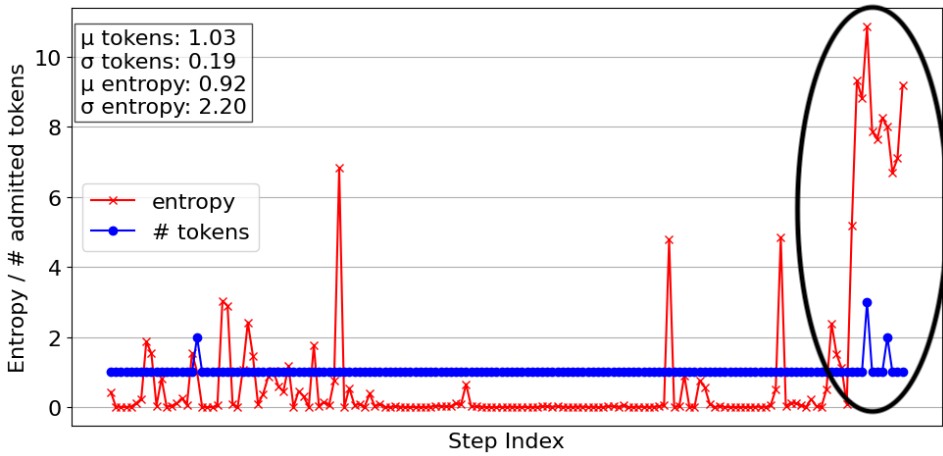

**Figure 18:** Entropy trace for failure pattern 1 example

**Question:** *Michael is replacing the carpet in his bedroom. The new carpet he's chosen costs $12 per square foot, plus $2 per square foot for padding underneath. His contractor charges $4 per square foot to remove the old carpet, and $34 per square foot to install the new carpet. His bedroom measures 18 feet by 12 feet. How much will it cost Michael to replace the carpet?*

**Reference Answer:** *The room measures 18 * 12 = 216 square feet. The new carpet costs $12 * 216 = $2,592. The new carpet padding costs $2 * 216 = $432. To remove the old carpet, the contractor will charge $4 * 216 = $864. To install the new carpet, the contractor will charge $34 * 216 = $7,344. The total cost will be $864 + $2,592 + $432 + $7,344 = $11,232. #### 11232*

**Generated Answer:** *The area of Michael's bedroom is 18 feet x 12 feet = 216 square feet. The cost of the carpet is 216 square feet x $12 per square foot = $2592. The cost of the padding is 216 square feet x $2 per square foot = $432. The cost to remove the old carpet is 216 square feet x $4 per square foot = $864. The cost to install the new carpet is 216 square feet x $34 per square foot = $7344. The total cost is $2592 + $432 + $864 + $7344 = $12132. The answer is 12132.*

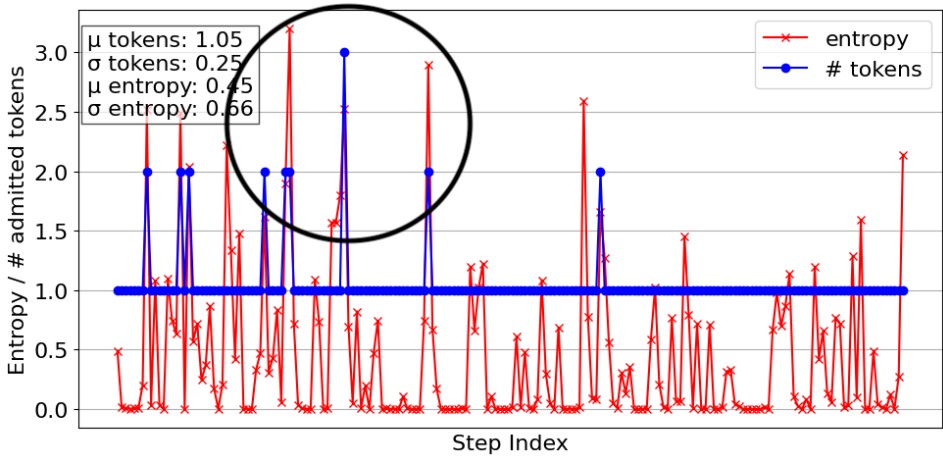

**Figure 19:** Entropy trace for failure pattern 2 example

**Pattern 2:** The phrasing of the question may be confusing and challenging to interpret, introducing substantial uncertainty from the outset. Under such conditions, the generation is more prone to misinterpret the question, resulting in an error even though the subsequent reasoning remains logically coherent and internally consistent.

In the following example, the error is committed right at the beginning of its reasoning chain, which is arguably due to the ambiguity in the question. The corresponding entropy trace is illustrated in Figure 19, where the region of elevated uncertainty and entropy is circled out. The corresponding generated tokens are highlighted in red.

**Question:** *Pat has a flower bed that is 111 feet long. Pat wants to fill her flower bed with plants. Pat's flowers grow 12 inches wide so she needs to leave 1.5 feet between every plant. Pat already owns 17 flowers. Each flowering plant costs $6 at the store, how much money will Pat spend at the store to fill up her flower bed?*

**Reference Answer:** *Pat's flower bed is 111 feet / 1.5 feet per plant = «111/1.5=74 »74 plants needed. Pat needs to buy 74 plants - 17 plants = «74-17=57 »57 plants needed to purchase. Pat will spend 57 plants \* $6 = $ «57\*6=342 »342. #### 342*

**Generated Answer:** *First, we need to find the total number of plants Pat needs to fill her flower bed. Since each plant grows 12 inches wide, that's 1 foot. With 1.5 feet between every plant, the total space taken up by each plant is 1 + 1.5 = 2.5 feet. Since the flower bed is 111 feet long, we can divide 111 by 2.5 to find the total number of plants needed: 111 / 2.5 = 44.4. Since we can't have a fraction of a plant, we'll round up to 45 plants. Pat already owns 17 flowers, so she needs to buy 45 - 17 = 28 more flowers. Each flower costs $6, so she'll spend 28 x 6 = $168 at the store. The answer is 168.*

