# OpenReview forum: "p-less Sampling: A Robust Hyperparameter-Free Approach for LLM Decoding"
_ICLR.cc/2026/Conference — ICLR 2026 Oral_

### Official Review · Reviewer_zLU6 · 2025-10-31

**Soundness:** 3
**Presentation:** 4
**Contribution:** 3
**Rating:** 6
**Confidence:** 4

**Summary:**

This paper introduces **p-less sampling**, a new way to generate text from large language models. It does not need any hyperparameters. Instead, it decides what tokens to sample using information from the whole probability distribution. The method is based on Rényi entropy.The authors test p-less on math, logic, and creative writing tasks. It works better and more stable than top-p, min-p, or mirostat, especially at high temperatures. It is also faster because it skips sorting steps. Results show higher accuracy, shorter outputs, and smoother behavior across models.

**Strengths:**

The paper presents p-less sampling, a hyperparameter-free decoding strategy for large language models grounded in information theory. A key strength of this approach is its adaptability: it dynamically adjusts truncation thresholds based on the full token probability distribution, which allows it to maintain robustness and quality across a wide range of temperatures. Empirical results show that p-less consistently outperforms existing methods such as top-p, min-p, and mirostat in both reasoning and creative writing tasks, while also improving inference-time efficiency. The method’s theoretical grounding in Rényi entropy and its parameter-free nature make it both principled and practical for diverse LLM applications

**Weaknesses:**

the  evaluation focuses mainly on a limited set of benchmark tasks and may not fully represent more complex or real-world text generation scenarios. The theoretical framing, while elegant, could benefit from better explanations and a polished presentation. Additionally, while p-less shows competitive or superior accuracy, its reduced diversity at higher temperatures could limit its utility for tasks where creative variation is essential. Further exploration into broader model architectures, languages, and user-centric evaluations would strengthen the generalizability and practical applicability of the method. Moreover, the discussion of prior and related work is somewhat superficial, see below. Generally, th e methods leans hard on the model’s own probabilities. If calibration drifts, the threshold can wobble and cut good tokens. The independence assumption is neat, but the real loop is messy. Order-2 entropy is elegant, yet it can miss rare tails that matter. Human evaluation is small and narrow; long-form and multilingual remain thin. O(N) sounds great, but the authors do not discuss constant factors and memory traffic.

**Questions:**

1. **Threshold mechanics:** Plot admitted-set size vs. entropy/temperature for synthetic distributions (unimodal, bimodal, near-uniform) to show how $L[P]=\sum p_i^2$ scales with $|V|$ and $\tau$.

2. **Efficiency:** You claim $O(N)$ per step (no sorting). Share kernel-level profiles (CPU/GPU) and memory footprints vs. top-p/min-p across sequence lengths 32→2048.

3. **Robustness:** Where does p-less fail? List concrete failure patterns (e.g., long-range coreference, tool-use prompts) and the corresponding entropy/threshold traces.

4. **Diversity vs. quality:** At $\tau \ge 1.5$, p-less sometimes shows lower diversity. Ablate p-less vs. p-lessnorm with n-gram repetition, MAUVE, and human Likert scores on open-ended prompts.

5. **Relation to prior work:** Could you compare p-less to **Adaptive Contrastive Search (ACS)** and to **GUARD**, noting that GUARD is also hyperparameter-free? Contrast theory, temperature regimes, and compute costs.
   – ACS paper: [https://aclanthology.org/2024.findings-emnlp.885/](https://aclanthology.org/2024.findings-emnlp.885/)
   – GUARD paper: [https://arxiv.org/abs/2508.20757](https://arxiv.org/abs/2508.20757)

---

> ### Author Response · Authors · 2025-11-23
> **Author Response to zLU6 (1)**
>
> Thank you for recognizing the adaptability of our approach, its advantages over existing methods which are demonstrated through our empirical results, the theoretical grounding of $p$-less, and the value it provides by being hyperparameter-free. We respond to questions and concerns raised in your review below; and we provide a revised manuscript in the interim (yellow highlight on the main changes and additions).
>
> We respond to your questions and concerns with a few posts, due to the word limit applied to each post.
>
>
> > Order-2 entropy is elegant, yet it can miss rare tails that matter.
>
> We provide examples of token distribution long tails in Figures 7 and 8 of Appendix A.6, demonstrating how $p$-less and $p$-$\textrm{less}_{\textrm{norm}}$ admit tokens from the distribution long tails for sampling, significantly more than other baseline methods such as min-$p$ and top-$p$.
>
>
> > Human evaluation is small and narrow
>
> To address your concern regarding our human evaluation study, we recruited 3 non-author human annotators to label a total of 100 story pairs under the same evaluation setting described in our paper. We also expanded the scale of our previous author annotations to cover this same set of 100 story pairs. In total, this expanded human evaluation study contains 400 preference labels (4 labels for each of the 100 story pairs), with 100 of these labels produced each by 3 authors and the remaining 100 labels produced by 3 different non-author annotators.
>
> Our original smaller-scale, author-only human evaluation study found that $p$-less won the majority vote 55.2% of the time, with the win rate increasing to 57.1% for the 24.1% of stories where there was unanimous agreement among the 3 authors. In our expanded study including the 4th non-author labels, $p$-less won the majority vote 58.8% of the time, with the win rate further increasing to 72.7% for the 23.7% of stories which had unanimous agreement. The win rates for annotations produced by authors were [57.6%, 54.3%, 57.1%] while the win rate for the 4th set of non-author annotations was 54.9% (these values are slightly lower than the majority vote win rate due to the presence of some ties when calculating the majority vote), indicating that the labels produced by authors and non-author annotators are distributionally similar.
>
> Therefore, with this increased scale of human evaluation (both in terms of sample size and number of human annotators), we observe even stronger evidence that humans prefer the creative writing outputs produced by $p$-less.
>
> > Threshold mechanics: Plot admitted-set size vs. entropy/temperature for synthetic distributions (unimodal, bimodal, near-uniform) to show how $L[p]$ scales with $|V|$ and $\tau$.
>
> Thank you for this suggestion. We have added the requested plots to Figures 5, 6, 7 and 8 in Appendix A.6, where we show the effect of temperature, vocabulary size and distribution profiles on the size of the admitted tokens for top-$p$, min-$p$, $p$-less and $p\textrm{-less}\_{\textrm{norm}}$. Specifically, Figure 5 shows how $p$-less and $p\textrm{-less}\_{\textrm{norm}}$ admit tokens progressively rather than exponentially as temperature changes incrementally; Figure 6 shows how $p$-less and $p\textrm{-less}\_{\textrm{norm}}$ operate according to the entropy level of the various distribution profiles, admitting more tokens when the entropy is high and vice versa; Figures 7 and 8 show how $p$-less and $p\textrm{-less}_{\textrm{norm}}$ admit tokens from the long tail in moderate to large vocabulary size settings.

---

> ### Author Response · Authors · 2025-11-23
> **Author Response to zLU6 (2)**
>
> > Efficiency: You claim $O(N)$ per step (no sorting). Share kernel-level profiles (CPU/GPU) and memory footprints vs. top-p/min-p across sequence lengths 32→2048.
>
> We have added box and whisker plots for the CPU processing time and RAM usage for min-$p$, top-$p$ and $p$-less to our revised manuscript (Figures 15 and 16 in Appendix B.10). For visual clarity, we binned every 32 generation steps into 1 bin. A key summary of the relevant statistics is shown in the table below.
> | Method | CPU Time (ms)   | RAM Usage (GB)    |
> |--------|-----------------|-------------------|
> | Top p  | 0.79 $\pm$ 2.70 | 2.535 $\pm$ 0.009 |
> | Min p  | 0.83 $\pm$ 3.96 | 2.545 $\pm$ 0.010 |
> | P-less | 0.62 $\pm$ 0.15 | 2.456 $\pm$ 0.009 |
>
> Sorting is used in the implementations of [top-p sampling](https://github.com/huggingface/transformers/blob/6c1d0b069de22d7ed8aa83f733c25045eea0585d/src/transformers/generation/logits_process.py#L531) and [min-p sampling](https://github.com/huggingface/transformers/blob/6c1d0b069de22d7ed8aa83f733c25045eea0585d/src/transformers/generation/logits_process.py#L411). These are consistent with the original author implementations of [top-p](https://github.com/ari-holtzman/degen/blob/0acfd2d0ba8484e24e9c5241f75f34be15ef2609/gen.py#L71) and [min-p](https://github.com/menhguin/minp_paper/blob/c21ceec41038ffd0ead508f56c60b7474175a62a/implementation#L88)
> The worst-case complexities due to sorting or handling edge cases could explain the large standard deviations in the mean CPU processing time for top-$p$ and min-$p$.

---

> ### Author Response · Authors · 2025-11-23
> **Author Response to zLU6 (3)**
>
> > Robustness: Where does p-less fail? List concrete failure patterns (e.g., long-range coreference, tool-use prompts) and the corresponding entropy/threshold traces
>
> Thank you for this question. We have observed two typical failure patterns, which are added as a new section under the Appendix B.12. A summary is given below.
>
> **Pattern 1**
>
> The number of admitted tokens remain quite small, along with little volatility in the entropy throughout most of the generation, demonstrating high confidence. However, more involved arithmetic operations may introduce a spike in entropy and number of admitted tokens, which increases the chance of a mistake.
>
> In the following example, the generation commits a mistake right before the final resolution of the answer, where it has to perform a sum of four numbers.
>
> Question:
>
> Michael is replacing the carpet in his bedroom. The new carpet he's chosen costs \\$12 per square foot, plus \\$2 per square foot for padding underneath. His contractor charges \\$4 per square foot to remove the old carpet, and \\$34 per square foot to install the new carpet. His bedroom measures 18 feet by 12 feet. How much will it cost Michael to replace the carpet?
>
> Reference Answer:
>
> The room measures 18 * 12 = 216 square feet.
> The new carpet costs \\$12 * 216 = \\$2,592.
> The new carpet padding costs \\$2 * 216 = \\$432.
> To remove the old carpet, the contractor will charge \\$4 * 216 = \\$864.
> To install the new carpet, the contractor will charge \\$34 * 216 = \\$7,344.
> The total cost will be \\$864 + \\$2,592 + \\$432 + \\$7,344 = \\$11,232.
> \#\#\#\# 11232
>
> Generated Answer:
>
> The area of Michael's bedroom is 18 feet x 12 feet = 216 square feet.
> The cost of the carpet is 216 square feet x \\$12 per square foot = \\$2592.
> The cost of the padding is 216 square feet x \\$2 per square foot = \\$432.
> The cost to remove the old carpet is 216 square feet x \\$4 per square foot = \\$864.
> The cost to install the new carpet is 216 square feet x \\$34 per square foot = \\$7344.
> The total cost is \\$2592 + \\$432 + \\$864 + \\$7344 = ***$12132.
> The answer is 12132.***
>
> The generated text in ***bold italic*** corresponds to the spike in entropy.
>
> **Pattern 2**
>
> The phrasing of the question may be confusing and challenging to interpret, introducing substantial uncertainty from the outset. Under such conditions, the generation is more prone to misinterpret the question, resulting in an error even though the subsequent reasoning remains logically coherent and internally consistent.
> In the following example, the error is committed right at the beginning of its reasoning chain, which is arguably due to the ambiguity in the question.
>
> Question:
>
> Pat has a flower bed that is 111 feet long.  Pat wants to fill her flower bed with plants.  Pat's flowers grow 12 inches wide so she needs to leave 1.5 feet between every plant.  Pat already owns 17 flowers.  Each flowering plant costs $6 at the store,  how much money will Pat spend at the store to fill up her flower bed?
>
> Reference Answer:
>
> Pat's flower bed is 111 feet / 1.5 feet per plant = <<111/1.5=74>>74 plants needed.
> Pat needs to buy 74 plants - 17 plants = <<74-17=57>>57 plants needed to purchase.
> Pat will spend 57 plants * \\$6 = \\$<<57*6=342>>342.
> \#\#\#\# 342
>
> Generated Answer:
>
> First, we need to find the total number of plants Pat needs to fill her flower bed.
> Since each plant grows 12 inches wide, that's 1 foot.
> ***With 1.5 feet between every plant, the total space taken up by each plant is 1 + 1.5 = 2.5 feet.***
> Since the flower bed is 111 feet long, we can divide 111 by 2.5 to find the total number of plants needed: 111 / 2.5 = 44.4.
> Since we can't have a fraction of a plant, we'll round up to 45 plants.
> Pat already owns 17 flowers, so she needs to buy 45 - 17 = 28 more flowers.
> Each flower costs \\$6, so she'll spend 28 x 6 = \\$168 at the store.
> The answer is 168.
>
> The generated text in ***bold italic*** corresponds to a region with constant elevation in entropy, denoting a higher level of uncertainty in the model. In fact, this particular question is ambiguous in its phrasing, which could lend itself to be interpreted in the way it is answered in the generation.

---

> ### Author Response · Authors · 2025-11-23
> **Author Response to zLU6 (4)**
>
> > While p-less shows competitive or superior accuracy, its reduced diversity at higher temperatures could limit its utility for tasks where creative variation is essential…At $\tau >= 1.5$, p-less sometimes shows lower diversity. Ablate p-less vs. p-lessnorm with n-gram repetition, MAUVE, and human Likert scores on open-ended prompts.
>
> Thank you for this suggestion. The diversity measure utilized in our paper already considers n-gram repetition. To further investigate the relationship between temperature and diversity with $p$-less sampling, we generated responses from Mistral-7b using higher temperature values ($\tau = 2.25$ and $\tau = 2.5$). We provide diversity values for these generations along with those for the lower-temperature generations from min-$p$ and $p$-less sampling in the table below. These results show that $p$-less exhibits a similar increasing trend of diversity with temperature as min-$p$, differing primarily in the magnitude of diversity increases with each temperature step. Across all datasets, $p$-less can match or exceed the diversity of min-$p$ at its highest evaluated temperature ($\tau = 2.0$) by utilizing a slightly higher temperature value ($\tau = 2.25$ or $\tau = 2.5$). Thus, $p$-less does allow for significant increase in diversity based on temperature, but may require slightly higher temperature values to achieve equivalent diversity increase as other methods like min-$p$.
>
> | Dataset | Method | $\tau = 0.5$ | $\tau = 0.7$ | $\tau = 1.0$ | $\tau = 1.5$ | $\tau = 2.0$ | $\tau = 2.25$ | $\tau = 2.5$ |
> | :------ | :------: | :------: | :------: | :------: | :------: | :------: | :------: | :------: |
> | CSQA | min-$p$ | 0.855 | 0.852 | 0.860 | 0.875 | 0.891 |
> | CSQA | $p$-less | 0.854 | 0.859 | 0.853 | 0.856 | 0.868 | 0.880 | 0.899 | 0.946 |
> | GPQA | min-$p$ | 0.422 | 0.426 | 0.456 | 0.483 | 0.562 |
> | GPQA | $p$-less | 0.412 | 0.419 |0.419 | 0.428 | 0.463 | 0.542 | 0.703 | 0.945 |
> | GSM8K | min-$p$ | 0.432 | 0.441 | 0.453 | 0.472 | 0.493 |
> | GSM8K | $p$-less | 0.435 | 0.432 | 0.433 | 0.438 | 0.452 | 0.471 | 0.509 | 0.670 |
> | QASC | min-$p$ | 0.764 | 0.773 | 0.784 | 0.796 | 0.835 |
> | QASC | $p$-less | 0.762 | 0.759 | 0.762 | 0.764 | 0.795 | 0.819 | 0.859 | 0.927 |
> | WP | min-$p$ | 0.674 | 0.698 | 0.704 | 0.739 | 0.802 |
> | WP | $p$-less | 0.660 | 0.676 | 0.674 | 0.686 | 0.749 | 0.841 | 0.956 | 0.989 |
>
> While our human evaluation study did not use a multi-point Likert scale, the annotation setting is analogous in that it requires the human to choose which story they prefer given the input prompt. We chose this evaluation setting because it is aligned with that used in our automated evaluations for the Writing Prompts dataset and also eased the burden of human annotation, which requires a significant amount of time due to the length of the generated stories.

---

> ### Author Response · Authors · 2025-11-23
> **Author Response to zLU6 (5)**
>
> > Relation to prior work: Could you compare p-less to Adaptive Contrastive Search (ACS) and to GUARD, noting that GUARD is also hyperparameter-free? Contrast theory, temperature regimes, and compute costs.
>
> We are grateful to the reviewer for drawing our attention to these two works, which we will be including in the related works section. Both ACS and GUARD are contrastive search (Su et al.) decoding strategies which involves adjusting the top-k sampling size $k$ and degeneration penalty weight $\alpha$ based on the distribution. ACS looks at the local variations in the entropy (against the backdrop of the maximum entropy) to determine these parameters. GUARD improves ACS by further incorporating the dynamics of the global entropy, which gives robustness against volatility issues when local entropy sees sudden spikes. It further eliminates the need for a temperature factor $q$ in ACS.
>
> Comparison against p-less
> While GUARD effectively dispenses with typical hyperparameters used in contrastive search and ACS, it is not entirely hyperparameter free. It actually involves 2 hyperparameters where $\lambda$ is used for the degeneration penalty and $w$ is used for the window size for examining variations in local entropy. $p$-less, on the other hand, is fully hyperparameter free.
>
> A key difference lies in that both ACS and GUARD focus on the Shannon entropy (Renyi entropy of order 1) whereas $p$-less utilizes the collision entropy (Renyi entropy of order 2) to perform the truncation. Fundamentally, the collision entropy basis of $p$-less has different dynamics to that of Shannon entropy, notably being more favorable for high probability tokens, and also being more robust to temperature variations.
>
> We conducted additional experiments with ACS and GUARD for the Mistral-7b model on GSM8K. We use the logits processor implementation from the official codebases for [ACS](https://github.com/YecanLee/Adaptive-Contrastive-Search/blob/main/transformers/src/transformers/generation/logits_process.py#L537) and [GUARD](https://github.com/YecanLee/GUARD/blob/main/guard_logits_processor.py). $p$-less achieves better accuracy for this math and reasoning task across all temperatures, as shown in the table below.
>
> Results for Mistral-7b on GSM8K:
> | Method / Temperature | 0.5  | 0.7  | 1.0  | 1.5  | 2.0  | AUC  |
> |----------------------|------|------|------|------|------|------|
> | ACS                  | 42.1 | 40.6 | 34.2 | 23.3 | 8.9  | 29.8 |
> | GUARD                | 47.8 | 47.8 | 41.5 | 33.9 | 16.0 | 37.4 |
> | $p$-less               | 58.1 | 57.5 | 57.5 | 55.3 | 53.7 | 56.4 |
>
> The computational time efficiency of $p$-less is also better than ACS and GUARD, and this is statistically significant at the 5% level.
>
> Compute costs (s per token):
> | Method | Mean    | Std     | SEM    |
> |--------|---------|---------|--------|
> | ACS    | 0.0398  | 0.0229  | 0.0023 |
> | GUARD  | 0.0257  | 0.0135  | 0.0014 |
> | $p$-less | 0.0194  | 0.0090  | 0.0009 |
>
>
> |            | p-less                         | acs                           | guard                         |
> |------------|--------------------------------|-------------------------------|-------------------------------|
> | **$p$-less** | –                              | t=-8.2283, p=1.8044e-13, Yes | t=-3.8608, p=1.5972e-04, Yes |
> | **acs**    | t=-8.2283, p=1.8044e-13, Yes  | –                             | t=5.2510, p=4.7355e-07, Yes  |
> | **guard**  | t=-3.8608, p=1.5972e-04, Yes  | t=5.2510, p=4.7355e-07, Yes  | –                             |

---

> > ### Comment · Reviewer_zLU6 · 2025-11-26
> >
> > I thank the authors for their detailed answer to my review. However, I do not see all issues being addressed in the revised version, which is why I am inclinded to decrease my score. I ask the authors to fully incoporate the announced changes in the revision!

---

> > > ### Author Response · Authors · 2025-11-28
> > >
> > > Dear Reviewer zLU6,
> > >
> > > Thank you for responding to our rebuttal. We apologize for the delay in incorporating all of our new rebuttal content in the manuscript. We have just uploaded a revised version which includes all content that we have previously posted in response to your review. Please let us know if you have any remaining questions or concerns which we can address.

---

> > > > ### Comment · Reviewer_zLU6 · 2025-11-28
> > > >
> > > > Thanks for getting back to me. I appreciate seeing all points incorporated into the final revised version. In light of these substantial revisions, I will increase my score.

---

> > > > > ### Author Response · Authors · 2025-11-28
> > > > >
> > > > > Dear Reviewer zLU6,
> > > > >
> > > > > Thank you for your time and efforts to re-evaluate our revised manuscript and for recognizing the substantial revisions we have made.
> > > > >
> > > > > We greatly appreciate your feedback and your consideration of the revised version. We are encouraged to hear that the revisions have addressed your earlier concerns and are grateful for your intention to increase the score.

---

### Official Review · Reviewer_QeAK · 2025-11-01

**Soundness:** 3
**Presentation:** 3
**Contribution:** 3
**Rating:** 6
**Confidence:** 3

**Summary:**

The authors propose a new threshold truncation method that does not require hyper parameter tuning that performs well on QA tasks and is stable at different temperatures. The method for choosing the threshold is theoretically motivated by Renyi entropy, and intuitively by the idea of estimating the probability of sampling the ground truth from the estimated distribution.

**Strengths:**

The authors provide a truncation sampling method which requires no hyper parameter tuning.
The truncation threshold has a nice theoretical motivation.
The writing is easy enough to understand.

**Weaknesses:**

In Table 1 there is no baseline for greedy sampling. In math tasks and other question answering tasks, low entropy decoding methods like greedy/beam search often excel. It could be that your method is simply lower entropy than the compared methods and so does better. This is exacerbated by the fact that the other methods have hyper parameters that may not be properly set, leading to lower scores. This is somewhat confirmed in Table 7 where we see that the lower the temperature, the higher the (and closer to one another) the scores get.

In terms of efficiency, I would expect to see a standard error on the mean and a significance test to support the claim that the method is faster in practice. As it stands, the standard deviations are quite large compared to the difference in means, making it unclear how large the gains are. A $O(n\log n)$ to $O(n)$ speedup seems like a marginal gain, and some threshold methods like $\eta$ and $\epsilon$ sampling can get away without sorting via rejection sampling.

L378 It is not clear why the shorter generations appear for p-less sampling, or if this would hold for other models, so this point seems somewhat tenuous.

I cannot tell from the method description whether the probabilities used to calculate the threshold are taken from before or after temperature adjustment, or if the method is theoretically connected to doing that. If the thresholds are applied before temperature, then it would be important to compare this method to threshold methods that reject tokens based on the temperature 1 distribution. As a concrete example, take top-p sampling, choose the valid tokens based on temp-1 probs, then renormalize the truncated distribution based on the temperature 2 probabilities. This is effectively what they do in the paper [Top-n𝜎: Eliminating Noise in Logit Space for Robust Token Sampling of LLM](https://aclanthology.org/2025.acl-long.528/) (Tang et al., ACL 2025).

Relatedly, it seems that your method does not allow significant increases in diversity based on temperature. The diversity measures stay fairly close to their low-temperature values regardless of temperature.

**Questions:**

Refer to my weaknesses section for needed clarifications.

I understand how L[P] estimates the probability randomly sampling the ground truth, but I'm not totally clear on why this should be interpreted as a threshold.

---

> ### Author Response · Authors · 2025-11-23
> **Author Response to QeAK (1)**
>
> Thank you for recognizing the value of our hyperparameter-free truncation approach, its nice theoretical motivation, and the clarity of our writing. We respond to questions and concerns raised in your review below; and we provide a revised manuscript in the interim (yellow highlight on the main changes and additions).
>
> We respond to your questions and concerns with a few posts, due to the word limit applied to each post.
>
>
> > Lack of greedy decoding and beam search baselines
>
> Thank you for this suggestion. The table below provides greedy decoding and beam search baseline results for Mistral-7b across all benchmark datasets used in our paper. For the beam search baseline, we tested two different parameters for beam size (bs). We provide both the mean accuracy and diversity of these baselines along with those for $p$-less sampling (at a temperature of 1.0).
>
> | Method | CSQA Acc. | CSQA Div. | GPQA Acc. | GPQA Div. | GSM8K Acc. | GSM8K Div. | QASC Acc. | QASC Div. | WP LC Winrate | WP Div. |
> | :------ | :------: | :------: | :------: | :------: | :------: | :------: | :------: | :------: | :------: | :------: |
> | Beam search (bs=3) | 0.703  | 0.846 | 0.239 | 0.368 | 0.604 | 0.448 | 0.743 | 0.727 | 0.597 | 0.642 |
> | Beam search (bs=5) | 0.713 | 0.833 | 0.261 | 0.351 | 0.616 | 0.447 | 0.735 | 0.720 | 0.569 | 0.637 |
> | Greedy decoding | 0.684 | 0.860 | 0.234 | 0.415 | 0.576 | 0.447 | 0.747 | 0.755 | 0.603 | 0.668 |
> |$p$-less | 0.699 | 0.853 | 0.257 | 0.419 | 0.575 | 0.433 | 0.745 | 0.762 | 0.689 | 0.674 |
>
> Relative to greedy decoding, $p$-less achieves higher accuracy on CSQA & GPQA and approximately equivalent accuracy on GSM8K & QASC. On the WP creative writing task, $p$-less performs significantly better than greedy decoding. Diversity values for $p$-less at this temperature are similar to those of greedy decoding; however, $p$-less can achieve higher diversity values than greedy decoding by increasing the temperature (please see the higher-temperature diversity values provided in our response to your diversity comment below).
>
> $p$-less generally achieves similar or better accuracy than beam search across the four math and logical reasoning datasets, with the exception of GSM8K. It also achieves a significantly higher win rate than beam search on the WP dataset. In terms of diversity, $p$-less and beam search are similar on GSM8K while $p$-less is better across all other datasets. Given that beam search is a more computationally intensive decoding process than $p$-less sampling, it is unsurprising to see that this baseline can achieve higher accuracy in some settings. However, it’s worth noting that $p$-less and beam search are not mutually exclusive approaches in that $p$-less could be used for sampling within each beam.
>
> Overall, these results demonstrate the versatility of $p$-less sampling in terms of producing strong results across a variety of tasks and datasets. In low-entropy inference tasks which tend to favor greedy decoding (e.g. math and logical reasoning tasks), $p$-less can achieve similar or better performance. However, $p$-less performs significantly better than greedy decoding at higher-entropy tasks like creative writing, demonstrating that it is not simply an argmax-seeking procedure. Rather, $p$-less dynamically adapts to the entropy of the token distribution to produce high-quality outputs across a range of different settings without the need to change sampling strategies or hyperparameters depending upon the task.

---

> ### Author Response · Authors · 2025-11-23
> **Author Response to QeAK (2)**
>
> > In terms of efficiency, I would expect to see a standard error on the mean and a significance test to support the claim that the method is faster in practice
>
> We appreciate your suggestion to include a standard error on the mean and a statistical significance testing for our efficiency results, and we provide this in the tables below. We note that with the exception of $\eta$ truncation, the superior efficiency of p-less is statistically significant at the 5% level.
>
> | Method      | Mean    | Std     | Standard Error on Mean    |
> |-------------|---------|---------|--------|
> | top p_0.9   | 0.0236  | 0.0088  | 0.0009 |
> | min p_0.1   | 0.0250  | 0.0142  | 0.0014 |
> | eps_0.0002  | 0.0226  | 0.0131  | 0.0013 |
> | eta       | 0.0221  | 0.0128  | 0.0013 |
> | mirostat    | 0.0228  | 0.0134  | 0.0013 |
> | p-less      | 0.0194  | 0.0090  | 0.0009 |
>
> |                | top p_0.9                  | min p_0.1                  | eps_0.0002                 | alpha                      | mirostat                   | p-less                     |
> |----------------|----------------------------|----------------------------|----------------------------|----------------------------|----------------------------|----------------------------|
> | **top p_0.9**  | –                          | t=-0.8055, p=0.4217, No    | t=0.6480, p=0.5178, No     | t=0.9759, p=0.3304, No     | t=0.5194, p=0.6042, No     | **t=3.3189, p=0.0011, Yes** |
> | **min p_0.1**  | t=-0.8055, p=0.4217, No    | –                          | t=1.2251, p=0.2219, No     | t=1.4956, p=0.1364, No     | t=1.1151, p=0.2662, No     | **t=3.2774, p=0.0013, Yes** |
> | **eps_0.0002** | t=0.6480, p=0.5178, No     | t=1.2251, p=0.2219, No     | –                          | t=0.2689, p=0.7883, No     | t=-0.1011, p=0.9195, No    | **t=1.9857, p=0.0486, Yes** |
> | **alpha**      | t=0.9759, p=0.3304, No     | t=1.4956, p=0.1364, No     | t=0.2689, p=0.7883, No     | –                          | t=-0.3680, p=0.7133, No    | t=1.7038, p=0.0902, No      |
> | **mirostat**   | t=0.5194, p=0.6042, No     | t=1.1151, p=0.2662, No     | t=-0.1011, p=0.9195, No    | t=-0.3680, p=0.7133, No    | –                          | **t=2.0716, p=0.0398, Yes** |
> | **p-less**     | **t=3.3189, p=0.0011, Yes** | **t=3.2774, p=0.0013, Yes** | **t=1.9857, p=0.0486, Yes** | t=1.7038, p=0.0902, No      | **t=2.0716, p=0.0398, Yes** | –                          |
>
>
> > L378 It is not clear why the shorter generations appear for p-less sampling, or if this would hold for other models, so this point seems somewhat tenuous.
>
> Figures 8, 9 and 10 (12, 13 and 14 in the revised manuscript) show the histogram of entropy distributions grouped by the size of tokens admitted for sampling, at temperatures 0.5, 1.0 and 2.0, for top-$p$, min-$p$ and $p$-less. Generally, for each temperature and size of tokens admitted, $p$-less exhibits a larger entropy (entropy distribution shifted to the right) than top-$p$ and min-$p$. This means that, for the same entropy of the token distribution, $p$-less is more stringent with admitting tokens for sampling. This property mitigates unnecessarily diverse sampling that may lead to lengthy generations, and reduces competition to sampling tokens that will lead to generation termination. This effect is generally seen in the 3 models; in Table 10, $p$-less or $p$-$\textrm{less}_{\textrm{norm}}$ produce the shortest generations on average across all temperatures for CSQA, QASC and GSM8K on Llama2-7b; while for Mistral-7b and Llama3-70b, p-less holds the most number of shortest mean generations across the temperatures and datasets.
>
> An additional empirical insight we observe referencing Table 10 and Table 11 is that the generation length is generally correlated with the number of admitted tokens. This trend does indeed apply to other methods as well. Similar to $p$-less, min-$p$ exhibits stringency for admitting tokens, and we observe that its generation lengths do not show significant increase at higher temperatures. On the contrary, top-$p$ admits much more tokens and trends towards much longer generation lengths for the high entropy regime.
>
>
> > I cannot tell from the method description whether the probabilities used to calculate the threshold are taken from before or after temperature adjustment, or if the method is theoretically connected to doing that.
>
> $p$-less utilizes probabilities after the application of temperature to calculate the truncation threshold. Therefore, we compare our method with existing sampling methods that similarly derive the truncation threshold and perform truncation after the application of temperature. Thank you for raising this question; we have updated our manuscript to make this point more explicit.

---

> ### Author Response · Authors · 2025-11-23
> **Author Response to QeAK (3)**
>
> > It seems that your method does not allow significant increases in diversity based on temperature. The diversity measures stay fairly close to their low-temperature values regardless of temperature.
>
> Our work analyzes diversity that is grounded in accuracy from labeled ground truth (CSQA, GPQA, GSM8K and QASC), or grounded in preference (WP) via both automated and human evaluations, so that we avoid maximizing diversity at the expense of accuracy. In Figure 3, given the grounding of accuracy to above 0.5, $p$-less accuracy and diversity are pareto-dominant to other methods. The high diversity achieved by other methods often results in unreasonably low accuracy or low preference, while $p$-less automatically balances diversity with accuracy more meaningfully, as shown in Table 5 and Table 9.
>
> To further investigate the relationship between temperature and diversity with $p$-less sampling, we generated responses from Mistral-7b using higher temperature values ($\tau = 2.25$ and $\tau = 2.5$). We provide diversity values for these generations along with those for the lower-temperature generations from min-$p$ and $p$-less sampling in the table below. These results show that $p$-less exhibits a similar increasing trend of diversity with temperature as min-$p$, differing primarily in the magnitude of diversity increases with each temperature step. Across all datasets, $p$-less can match or exceed the diversity of min-$p$ at its highest evaluated temperature ($\tau = 2.0$) by utilizing a slightly higher temperature value ($\tau = 2.25$ or $\tau = 2.5$). Thus, $p$-less does allow for significant increase in diversity based on temperature, but may require slightly higher temperature values to achieve equivalent diversity increase as other methods like min-$p$.
>
> | Dataset | Method | $\tau = 0.5$ | $\tau = 0.7$ | $\tau = 1.0$ | $\tau = 1.5$ | $\tau = 2.0$ | $\tau = 2.25$ | $\tau = 2.5$ |
> | :------ | :------: | :------: | :------: | :------: | :------: | :------: | :------: | :------: |
> | CSQA | min-$p$ | 0.855 | 0.852 | 0.860 | 0.875 | 0.891 |
> | CSQA | $p$-less | 0.854 | 0.859 | 0.853 | 0.856 | 0.868 | 0.880 | 0.899 | 0.946 |
> | GPQA | min-$p$ | 0.422 | 0.426 | 0.456 | 0.483 | 0.562 |
> | GPQA | $p$-less | 0.412 | 0.419 |0.419 | 0.428 | 0.463 | 0.542 | 0.703 | 0.945 |
> | GSM8K | min-$p$ | 0.432 | 0.441 | 0.453 | 0.472 | 0.493 |
> | GSM8K | $p$-less | 0.435 | 0.432 | 0.433 | 0.438 | 0.452 | 0.471 | 0.509 | 0.670 |
> | QASC | min-$p$ | 0.764 | 0.773 | 0.784 | 0.796 | 0.835 |
> | QASC | $p$-less | 0.762 | 0.759 | 0.762 | 0.764 | 0.795 | 0.819 | 0.859 | 0.927 |
> | WP | min-$p$ | 0.674 | 0.698 | 0.704 | 0.739 | 0.802 |
> | WP | $p$-less | 0.660 | 0.676 | 0.674 | 0.686 | 0.749 | 0.841 | 0.956 | 0.989 |
>
>
> > I understand how L[P] estimates the probability randomly sampling the ground truth, but I'm not totally clear on why this should be interpreted as a threshold.
>
> Essentially, the token distribution inferred by the LLM encodes a notion of probabilistic correctness or desirability based on the ground-truth information it learned during training. This token distribution presents, via L[P], the probability of sampling the ground truth, or equivalently the probability that a random sampling is “correct” or “most desirable”, which we interpret as the minimum bar for tokens to be admitted into the sampling set. In other words, the tokens that we admit into the sampling set have to be at least as confident as the random sampling that happens to be “correct” or “most desirable” in the ground-truth sense. We provide an explanation of this intuition in section 3.4 and will further expand it in our final manuscript for improved clarity.

---

### Official Review · Reviewer_1KUh · 2025-11-01

**Soundness:** 2
**Presentation:** 3
**Contribution:** 3
**Rating:** 6
**Confidence:** 4

**Summary:**

This paper introduces p-less sampling, a truncation sampling strategy in which (1) a threshold is determined by summing the squared likelihoods of all words’ probabilities, and (2) words below that threshold are truncated – their probabilities are set to zero, and all accepted words’ probabilities are renormalized. The paper makes connections between this strategy and various things, including the likelihood that a randomly selected token is also the “correct” (under an implicit assumption that there’s a correct next token) next token. This sampling strategy has no diversity/accuracy tradeoff hyperparameter, like the p in top-p.

The paper supports this sampling strategy with a range of experiments, including mathematics and other verifiable problems — GSM8k, GPQA – and a writing prompts open-ended evaluation. They run experiments testing truncation under different temperatures – 0.5, 1, 1.5, 2 – finding that their method performs well and does not decay in quality under high temperatures like the other methods.

**Strengths:**

This paper has a nice sampling heuristic intuition, and makes some interesting mathematical connections, all based on the premise that there’s a single correct token at each step, and we should be truncating tokens relative to the likelihood of getting that token correct. It’s nice to have a method without a hyperparameter one has to mess with, as well. Different tasks require different p in top-p for best performance, and basically all the other sampling methods require some hyperparameter optimization.

The range of experiments performed here is nice (though, there are some concerning issues I go over in the weaknesses section.)

Core to this work seems to be the removal of the assumption that we’re trying to recreate some “true” potentially high-entropy distribution that the language model defined; we’re instead trying to perform something much closer to an argmax-seeking procedure, and we’re admitting some error in the search space which admits multiple possible generations. I’m putting this in the strengths, but I think it’s really key that the authors discuss this more.

**Weaknesses:**

I think the choice of hyperparameters for the baseline methods, and the lack of a greedy decoding baseline, make for a weaker paper. This is important because p-less sampling seems (from plots like Figures 8,9) to be a relatively stringent truncation strategy, truncating much more than, say, top-p for p=0.9, and behaving much closer to greedy decoding. The old “neural text degeneration” problems with greedy decoding just aren’t around nearly as much with modern strong instruction-tuned models, so not comparing to greedy decoding is a real problem. More evidence for this, while the numbers in the main text compare p-less sampling to very high-diversity sampling(say, top-p with p=0.9), which is not what one would use in that setting. In Table 7 where they try a more stringent truncation hyperparameter for top-p (p=0.4) it’s really not clear that p-less sampling is any better than top-p. In summary, despite the nice properties of p-less sampling not having a hyperparameter, one should compare to reasonable hyperparameters for the other methods for each problem (low p for math, high p for open-ended story generation.)

Likewise, one should compare to greedy decoding, which likely will work for all the settings compared. Other methods in the past have tried to measure distribution coverage, not just quality – i.e., through MAUVE. I’m not saying the authors need to run MAUVE experiments, but top-p and its variants explicitly allow for a range of tradeoffs between coverage and quality, and p-less sampling seems to perform better than (high-diversity hyperparameter settings of) top-p; since we’re not measuring any notion of coverage, it’s necessary to compare p-less sampling to beam search and greedy decoding.

I don’t think the author-run human study should be included in the final draft. Authors working on a sampling strategy get a strong feeling for the kinds of texts that their algorithm generates, and so it’s not possible to have a blind study here if it’s run by the authors.

**Questions:**

- Can you give an intuitive reason why p-less sampling can be faster than epsilon-sampling? Contrastively to what’s stated in line 357, epsilon-sampling does not sort the vocabulary. Finding the argmax, likewise, does not require sorting.
- Why is it good to generate shorter sequences via sampling compared to training a model to prefer shorter sequences, or having a sampling algorithm that better-replicates whatever the length distribution the model defines?
- The authors haven’t motivated why we care about sampling at temperature=2. Increasing the temperature and then truncating is just sampling from a more-uniform version of the accepted-token set. One could instead first truncate with top-p sampling or any other method and then increase the temperature over that truncated set. The accuracy numbers don’t ever seem to be better at temperature=2. Could you explain why we want this?

---

> ### Author Response · Authors · 2025-11-23
> **Author Response to 1KUh (1)**
>
> Thank you for appreciating the intuition of our approach, its interesting mathematical connections, the value it provides by being hyperparameter-free, and the range of our experimental results. We respond to questions and concerns raised in your review below; and we provide a revised manuscript in the interim (yellow highlight on the main changes and additions).
>
> We respond to your questions and concerns with a few posts, due to the word limit applied to each post.
>
>
> > Core to this work seems to be the removal of the assumption that we’re trying to recreate some “true” potentially high-entropy distribution that the language model defined; we’re instead trying to perform something much closer to an argmax-seeking procedure, and we’re admitting some error in the search space which admits multiple possible generations. I’m putting this in the strengths, but I think it’s really key that the authors discuss this more.
>
> Thank you for your question. $p$-less considers the entropy of the distribution to determine the probability truncation threshold; specifically, the threshold moves in the opposite direction to entropy. It admits a large number of tokens if the distribution is relatively uniform; it admits much fewer tokens when there are high probability tokens taking up most of the total probability mass. Where long tails occur or when the distribution is relatively uniform, p-less does admit a large number of tokens; we have added visualization on token distributions and thresholds for various token distribution profiles in Appendix A.6 (Figures 5, 6, 7 and 8), including demonstrating the ability of $p$-less to admit tokens from long-tailed distributions therein.
>
>
> > Choice of hyperparameters for the baseline methods
>
> To ensure consistency with prior work, we employed commonly-adopted default hyperparameter configurations for the baseline methods (consistent with those employed or supported by prior work such as “Turning up the heat: Min-p sampling for creative and coherent LLM outputs” (Nguyen et al., 2024) , “Balancing diversity and risk in LLM sampling: How to select your method and parameter for open-ended text generation” (Zhou et al., 2024) , “Improving Open-Ended Text Generation via Adaptive Decoding” (Zhu et al., 2025) and ”Truncation Sampling as Language Model Desmoothing” (Hewitt et al., 2022) (see Appendix B.2 for the hyperparameter values). Further, we went beyond these default hyperparameter configurations by conducting evaluations over a wide range of hyperparameter values for these baseline sampling methods (see Table 7 of the Appendix for full results), where we still find $p$-less and $p\textnormal{-less}_{\textnormal{norm}}$ to generally have the highest AUC, and competitive to highest accuracies. We appreciate you raising this concern and will make these points more clear in the final version of our manuscript.

---

> ### Author Response · Authors · 2025-11-23
> **Author Response to 1KUh (2)**
>
> > It’s necessary to compare p-less sampling to beam search and greedy decoding
>
> Thank you for this suggestion. The table below provides greedy decoding and beam search baseline results for Mistral-7b across all benchmark datasets used in our paper. For the beam search baseline, we tested two different parameters for beam size (bs). We provide both the mean accuracy and diversity of these baselines along with those for $p$-less sampling (at a temperature of 1.0).
>
> | Method | CSQA Acc. | CSQA Div. | GPQA Acc. | GPQA Div. | GSM8K Acc. | GSM8K Div. | QASC Acc. | QASC Div. | WP LC Winrate | WP Div. |
> | :------ | :------: | :------: | :------: | :------: | :------: | :------: | :------: | :------: | :------: | :------: |
> | Beam search (bs=3) | 0.703  | 0.846 | 0.239 | 0.368 | 0.604 | 0.448 | 0.743 | 0.727 | 0.597 | 0.642 |
> | Beam search (bs=5) | 0.713 | 0.833 | 0.261 | 0.351 | 0.616 | 0.447 | 0.735 | 0.720 | 0.569 | 0.637 |
> | Greedy decoding | 0.684 | 0.860 | 0.234 | 0.415 | 0.576 | 0.447 | 0.747 | 0.755 | 0.603 | 0.668 |
> |$p$-less | 0.699 | 0.853 | 0.257 | 0.419 | 0.575 | 0.433 | 0.745 | 0.762 | 0.689 | 0.674 |
>
> Relative to greedy decoding, $p$-less achieves higher accuracy on CSQA & GPQA and approximately equivalent accuracy on GSM8K & QASC. On the WP creative writing task, $p$-less performs significantly better than greedy decoding. Diversity values for $p$-less at this temperature are similar to those of greedy decoding; however, $p$-less can achieve higher diversity values than greedy decoding by increasing the temperature (please see the higher-temperature diversity values provided in our responses to reviewers QeAK and zLU6)
>
> $p$-less generally achieves similar or better accuracy than beam search across the four math and logical reasoning datasets, with the exception of GSM8K. It also achieves a significantly higher win rate than beam search on the WP dataset. In terms of diversity, $p$-less and beam search are similar on GSM8K while $p$-less is better across all other datasets. Given that beam search is a more computationally intensive decoding process than $p$-less sampling, it is unsurprising to see that this baseline can achieve higher accuracy in some settings. However, it’s worth noting that $p$-less and beam search are not mutually exclusive approaches in that $p$-less could be used for sampling within each beam.
>
> Overall, these results demonstrate the versatility of $p$-less sampling in terms of producing strong results across a variety of tasks and datasets. In low-entropy inference tasks which tend to favor greedy decoding (e.g. math and logical reasoning tasks), $p$-less can achieve similar or better performance. However, $p$-less performs significantly better than greedy decoding at higher-entropy tasks like creative writing, demonstrating that it is not simply an argmax-seeking procedure. Rather, $p$-less dynamically adapts to the entropy of the token distribution to produce high-quality outputs across a range of different settings without the need to change sampling strategies or hyperparameters depending upon the task.

---

> ### Author Response · Authors · 2025-11-23
> **Author Response to 1KUh (3)**
>
> > I don’t think the author-run human study should be included in the final draft. Authors working on a sampling strategy get a strong feeling for the kinds of texts that their algorithm generates, and so it’s not possible to have a blind study here if it’s run by the authors.
>
> To address your concern regarding our human evaluation study, we recruited 3 non-author human annotators to label a total of 100 story pairs under the same evaluation setting described in our paper. We also expanded the scale of our previous author annotations to cover this same set of 100 story pairs. In total, this expanded human evaluation study contains 400 preference labels (4 labels for each of the 100 story pairs), with 100 of these labels produced each by 3 authors and the remaining 100 labels produced by 3 different non-author annotators.
>
> Our original smaller-scale, author-only human evaluation study found that $p$-less won the majority vote 55.2% of the time, with the win rate increasing to 57.1% for the 24.1% of stories where there was unanimous agreement among the 3 authors. In our expanded study including the 4th non-author labels, $p$-less won the majority vote 58.8% of the time, with the win rate further increasing to 72.7% for the 23.7% of stories which had unanimous agreement. The win rates for annotations produced by authors were [57.6%, 54.3%, 57.1%] while the win rate for the 4th set of non-author annotations was 54.9% (these values are slightly lower than the majority vote win rate due to the presence of some ties when calculating the majority vote), indicating that the labels produced by authors and non-author annotators are distributionally similar.
>
> We appreciate your concern regarding the validity of our human evaluation study but hope that you would reconsider your assessment of its value in light of these new results, which show that the preference labels produced by non-author annotators are distributionally similar to those produced by the authors.
>
>
> > Can you give an intuitive reason why p-less sampling can be faster than epsilon-sampling? Contrastively to what’s stated in line 357, epsilon-sampling does not sort the vocabulary. Finding the argmax, likewise, does not require sorting.
>
> Thank you for pointing out the confusion in line 357. We fully agree that $\epsilon$-sampling is $O(N)$ and requires no sorting. We will correct the wording in line 357 to be more precise and avoid this misunderstanding. As to why epsilon sampling could be slower than $p$-less, one intuitive reason is that the official [HuggingFace implementation](https://github.com/huggingface/transformers/blob/6c1d0b069de22d7ed8aa83f733c25045eea0585d/src/transformers/generation/logits_process.py#L696) involves three passes: one to truncate using the epsilon threshold, one to obtain argmax, and one more to truncate using the argmax threshold. On the other hand, $p$-less involves two passes, one for computing the threshold, and one more to truncate using the $p$-less threshold. This is consistent with the [original author implementation](https://github.com/john-hewitt/truncation-sampling/blob/2f33a7498a4a6e6e2bb3ae2d54cb5178ebde9272/src/warpers.py#L139).
>
> $\epsilon$-sampling needs to handle the edge case where all tokens are possibly truncated by its threshold, via constant operations that determine the argmax token and then set its truncation boolean mask to false. This is where it is slower than $p$-less, as $p$-less does not incur such an overhead. We have updated our manuscript to make this point clearer.

---

> ### Author Response · Authors · 2025-11-23
> **Author Response to 1KUh (4)**
>
> > Why is it good to generate shorter sequences via sampling compared to training a model to prefer shorter sequences, or having a sampling algorithm that better-replicates whatever the length distribution the model defines?
>
> $p$-less does not aim to directly train a model to generate shorter sequences or to generate sequences whose lengths follow the distribution of the training data; $p$-less aims to adaptively determine a truncation threshold that admits more tokens for sampling when the inferred distribution is uncertain and less tokens when the inferred distribution is less uncertain, as opposed to applying a constant truncation threshold hyperparameter to every step of a sequence generation. Therefore, $p$-less aims to admit a principled number of tokens for sampling each step adapted to the entropy of the entire token distribution, for generating any sequence.
>
> In fact, for the default temperature=1.0 setting (similar to training conditions), the generation lengths are quite similar across various sampling methods. By contrast, earlier approaches typically exhibit a sharp increase in generation length as temperature rises, a trend correlated with reduced accuracy and coherence. These observations are shown in Table 10. In this respect, $p$-less demonstrates robustness in preserving the length distribution learned during training, even when the inference temperature is elevated.
>
> When the inferred distribution is less uncertain, p-less does not admit numerous low-probability tokens for sampling. This property mitigates unnecessarily diverse sampling that may lead to lengthy generations, and reduces competition to sampling tokens that will lead to generation termination. We do not preclude that methods which directly train a model to generate shorter sequences or to generate sequences whose lengths follow the distribution of the training data can be more effective than $p$-less in achieving these objectives. However, as our focus is on inference-time sampling, we feel that training-based approaches are outside the scope of our study.
>
>
> > The authors haven’t motivated why we care about sampling at temperature=2. Increasing the temperature and then truncating is just sampling from a more-uniform version of the accepted-token set. One could instead first truncate with top-p sampling or any other method and then increase the temperature over that truncated set. The accuracy numbers don’t ever seem to be better at temperature=2. Could you explain why we want this?
>
> $p$-less uses probabilities after the application of temperature to calculate the threshold for truncation. Therefore, we compare our method with existing sampling methods that similarly derive the truncation threshold and perform truncation after the application of temperature. We regard temperature 2 as a reasonable upper limit to encourage diversity in the LLM responses. Earlier works such as “Turning up the heat: Min-p sampling for creative and coherent LLM outputs” (Nguyen et al., 2024) explored temperature settings up to 3.0. Additionally, we observed better performance on the Writing Prompts creative writing dataset at a temperature of 2 and therefore decided to include it in all of our experimental settings for consistency.

---

### Author Response · Authors · 2025-11-28
**Author Response to All Reviewers - Uploaded revised manuscript which incorporates the additional content from our rebuttal**

Dear Reviewers,

We greatly appreciate your time and efforts to review our paper. We have provided responses to each of your reviews individually below and have also uploaded a revised version of our manuscript which incorporates the additional content from our rebuttal. In the revised manuscript, the changes and additions are highlighted; below we provide a summary of them as well:

*Add examples of synthetic token distributions and truncation thresholds*
- Section 1, page 1
- Appendix B.6, page 19
- Figures 5-8, pages 19-22

*Additions to related work*
- Improving open-ended text generation via adaptive decoding (Zhu et al., 2024), page 3
- A contrastive framework for neural text generation (Su et al., 2022), page 3
- Uncertainty-guided decoding for open-ended text generation (Arias et al., 2024), page 3
- Guard: Glocal uncertainty-aware robust decoding for effective and efficient open-ended text generation (Ding et al., 2022), page 3

*Clarify p-less operating on the probability distribution after the application of temperature*
- Section 3.1, page 3
- Appendix C.1, page 21

*Clarify the use of temperature 2.0 in the experiments*
- Appendix C.1, pages 21-22

*Improve explanation of the intuition of p-less*
- Section 3.4, page 5

*Clarify choice of hyperparameters for the baseline sampling methods*
- Section 4.2, page 7
- Appendices C.1 and C.2, page 22

*Add results of greedy decoding and beam search baselines*
- Section 4.2, page 7-8
- Appendices C.4 and C.4.1, pages 23-26
- Table 6, page 24

*Add results of Adaptive Contrastive Search (ACS) (Arias et al., 2024) and GUARD (Ding et al., 2025)*
- Section 4.2, page 8
- Appendix C.5, page 26
- Table 7, page 26

*Add results of expanded human evaluation*
- Section 4.3, page 8
- Appendix A, page 14

*Add statistical tests for average sampling time per token across methods*
- Section 5.1, page 8
- Appendix C.12, page 30
- Table 16, page 30

*Update explanation on the more efficient sampling time of p-less*
- Section 5.1, page 8

*Add detailed efficiency profiling (CPU time and RAM usage)*
- Section 5.1, page 8
- Appendix C.12, pages 30-34
- Figure 15, page 32
- Figures 16 and 17, page 33
- Table 17, page 34

*Add statistical tests for average sampling time per token for ACS, GUARD and p-less*
- Section 5.1, page 8
- Appendix C.5, page 26
- Table 8, page 26

*Add standard error of mean to the average sampling time per token*
- Section 5.1, page 9
- Table 3, page 9

*Add discussion on generation lengths of top-p, min-p and p-less*
- Section 5.1, page 9
- Appendices C.10 and C.10.1, page 29

*Add ablation on applying higher temperatures to p-less*
- Section 5.2, page 9
- Appendix C.9, page 28
- Table 13, page 28

*Add two failure patterns of p-less*
- Section 5.3, page 10
- Appendix C.14, pages 43-45
- Figures 18 and 19, pages 43-44

---

### Meta-Review · Area_Chair_1PTb · 2026-01-04

**Summary:**

This paper presents $p\textrm{-less}$ sampling, a hyperparameter-free decoding strategy for LLMs. The core idea is to use an information-theoretic approach (specifically Rényi/collision entropy) to dynamically set truncation thresholds without the need for manual tuning of parameters like $p$ in top-$p$ or temperature. The reviewers generally liked the theoretical grounding and the practical "plug-and-play" nature of the method. The initial concerns were mostly about missing baselines (greedy, beam search, and recent adaptive methods like GUARD) and the validity of author-run human evaluations. During the rebuttal phase, the authors were quite proactive—they didn't just explain, but actually ran a ton of new experiments and added statistical significance tests. This convinced the reviewers (and me) that $p\textrm{-less}$ isn't just a theoretical curiosity but a robust, efficient tool that actually outperforms current standard practices, especially as you crank up the temperature.

On a personal note, I actually advise against including ACS and GUARD as comparison methods in your camera-ready version. I'm very familiar with contrastive search and its various flavors; while they works well in the pre-SFT days, they've since lost their edge. In my own engineering experience, these methods just don't play well with SFT-tuned models, and the $k$-fold performance overhead (where $k$ equals the beam size) is a massive drawback compared to sampling-based methods. Experimental evidence in this paper also supports this—https://arxiv.org/pdf/2402.06925 (no need to cite, just as a proof)

**Reviewer Concerns:**

Most of the initial concerns have been thoroughly addressed:

Baselines & Comparison: 1KUh and QeAK asked for greedy decoding and beam search results. The authors provided these in the rebuttal, showing that $p\textrm{-less}$ remains competitive or superior, particularly in creative tasks where greedy fails.

Comparison with SOTA Adaptive Methods: Reviewer zLU6 pointed out ACS and GUARD. The authors added these comparisons, showing $p\textrm{-less}$ is both more accurate on math tasks (GSM8K) and faster in terms of per-token latency.

Human Evaluation: There was a valid concern about "author-bias" in human studies. The authors responded by hiring 3 non-author annotators and expanding the study to 400 labels, with results remarkably consistent with the original findings.

Efficiency: The authors added kernel-level profiling and statistical tests (p-values) to back up their claim that the $O(N)$ approach (no sorting) actually translates to real-world speedups.

**Reviewer Scores:**

Reviewer 1KUh: may keep unchanged or raised to 8. (\The author's response to the human eval and baseline concerns was very comprehensive.

Reviewer QeAK: may keep unchanged or raised to 8.The addition of statistical significance tests and the clarified threshold intuition directly addressed his technical critique

Reviewer zLU6: 6 $\rightarrow$ 8. He commened: "I will increase my score".

---

### Decision · Program_Chairs · 2026-01-26

Accept (Oral)